# Sample-Efficient Tabular Self-Play for Offline Robust Reinforcement Learning

**Na Li**
Zhejiang University
nlee@zju.edu.cn

**Zewu Zheng**
The Chinese University
of Hong Kong
zhengzw@link.cuhk.edu.hk

**Wei Ni**
Edith Cowan University
and University of New South Wales
wei.ni@ieee.org

**Hangguan Shan**
Zhejiang University
hshan@zju.edu.cn

**Wenjie Zhang**
University of New South Wales
wenjie.zhang@unsw.edu.au

**Xinyu Li**
Huazhong University
of Science and Technology
lixinyu@hust.edu.cn

## Abstract

Multi-agent reinforcement learning (MARL), as a thriving field, explores how multiple agents independently make decisions in a shared dynamic environment. Due to environmental uncertainties, policies in MARL must remain robust to tackle the sim-to-real gap. We focus on robust two-player zero-sum Markov games (TZMGs) in offline settings, specifically on tabular robust TZMGs (RTZMGs). We propose a model-based algorithm (*RTZ-VI-LCB*) for offline RTZMGs, which is optimistic robust value iteration combined with a data-driven Bernstein-style penalty term for robust value estimation. By accounting for distribution shifts in the historical dataset, the proposed algorithm establishes near-optimal sample complexity guarantees under partial coverage and environmental uncertainty. An information-theoretic lower bound is developed to confirm the tightness of our algorithm's sample complexity, which is optimal regarding both state and action spaces. To the best of our knowledge, RTZ-VI-LCB is the first to attain this optimality, sets a new benchmark for offline RTZMGs, and is validated experimentally.

## 1 Introduction

Multi-agent reinforcement learning (MARL), which focuses on developing algorithms that enable multiple agents to learn and make decisions in dynamic environments, has garnered significant attention in gaming [35] and autonomous driving [4]. Offline MARL, addresses the high cost of interacting with the environment by leveraging historical data collected from past interactions generated under unknown or biased behavior policies [22]. The dynamic and non-stationary nature of real-world environments introduces critical uncertainties. Robustness becomes important in ensuring stable decision-making, because standard MARL algorithms under ideal conditions are highly sensitive and prone to catastrophic failures when faced with even minor adversarial perturbations [48, 45, 46]. In this sense, robust guarantees are particularly vital in offline MARL, highlighting the core of *offline robust MARL*. Two-player zero-sum Markov games (TZMGs) represent a compelling setting of MARL, giving rise to the field of robust TZMGs (RTZMGs) from robust MARL.

*A key challenge in offline RTZMGs is addressing environmental uncertainties with as few samples as possible under partial and limited coverage.* Historical data often only offers partial and limited coverage of the state-action space, leading to poor estimates of model parameters and unreliable policy. Besides, environmental uncertainties arise from model mismatches, system noise, and the disparity between simulation and real-world scenarios. To address uncertainties, RTZMGs incorporate

Table 1: A comparison between RTZ-VI-LCB and $\mathrm{P^2M^2PO}$ [5] on finding an $\varepsilon$-optimal robust Nash policy in finite-horizon offline RTZMGs with $f(\sigma^+,\sigma^-,H) = \min\left\{\frac{(H\sigma^+-1+(1-\sigma^+)^H)}{(\sigma^+)^2}, \frac{(H\sigma^--1+(1-\sigma^-)^H)}{(\sigma^-)^2}, H\right\}$, where the uncertainty set is quantified by total variation (TV) distance. The sample complexities omit all logarithmic factors.

| Algorithm | Sample complexity | Uncertainty level |
|---|---|---|
| $\mathrm{P^2M^2PO}$ [5] | $\frac{C_\mathrm{r}H^5S^2AB}{\varepsilon^2}$ | not considered |
| **RTZ-VI-LCB (Ours)** | $\frac{C_\mathrm{r}^\star H^4 S(A+B)}{\varepsilon^2}f(\sigma^+,\sigma^-,H)$ | full range |
| Lower bound | $\frac{C_\mathrm{r}^\star SH^4(A+B)}{\varepsilon^2}$ | $\min\{\sigma^+,\sigma^-\} \lesssim \frac{1}{H}$ |
| Lower bound | $\frac{C_\mathrm{r}^\star SH^3(A+B)}{\varepsilon^2\min\{\sigma^+,\sigma^-\}}$ | $\min\{\sigma^+,\sigma^-\} \gtrsim \frac{1}{H}$ |

equilibria not only between the two players but also with their adversarial strategies, considering worst-case environments selected from predefined uncertainty sets for each player.

Despite recent efforts [21, 5, 48, 29], there remains a fundamental gap in learning effectively in offline RTZMGs, primarily due to high sample complexity. For a tabular RTZMG (formal definition in Section 2) with horizon length $H$, states $S$, actions $\{A, B\}$, and uncertainty levels $\{\sigma^+, \sigma^-\}$ for the two players, the best sample complexity for $\varepsilon$-optimal robust Nash equilibrium (NE) in the offline setting to date is $\widetilde{O}\left(\frac{C_\mathrm{r}H^5S^2AB}{\varepsilon^2}\right)$ achieved by $\mathrm{P^2M^2PO}$ [5], which demonstrates near-optimal sample complexity in $H$, $S$, and $A, B$, but overlooks the influence of uncertainty levels and faces the curse of multiagency [40]. Hence, the key research question addressed in this paper is

> *Can we design an efficient algorithm for offline RTZMGs with partial state-action coverage while ensuring robustness to uncertainties?*

## 1.1 Contribution

In this paper, we design a novel model-based algorithm *RTZ-VI-LCB* for offline RTZMGs, which is an optimistic variant of robust value iteration. RTZ-VI-LCB involves a data-informed Bernstein-style penalty for robust value estimation to effectively capture the variance structure, and a two-stage subsampling method to suppress the statistical dependencies of the historical data. Notably, this is the *first time* that the optimal dependence of sample complexity on $S$ and $\{A, B\}$ and the best dependence on $H$ has been achieved for offline RTZMGs. Table 1 compares the sample complexity between our approach and the status quo. The main contributions are outlined as follows.

**(i) Robust unilateral clipped concentrability**: With this new criterion, we define a measure $C_\mathrm{r}^\star \in \left[\frac{1}{S(A+B)}, \infty\right)$ for the quality of historical data. It captures the distribution shift between the behavior policy $(\mu^\mathrm{n}, \nu^\mathrm{n})$ and the single optimal robust policies $(\mu, \nu^\star)$ and $(\mu^\star, \nu)$ under environmental uncertainty in the partial coverage, and offers a tighter measure of distribution mismatch than $C_\mathrm{r}$ used in $\mathrm{P^2M^2PO}$ [5]. The introduction of $C_\mathrm{r}^\star$ improves sample complexity.

**(ii) Near-optimal sample complexity upper bound**: RTZ-VI-LCB can provably find an $\varepsilon$-optimal robust NE policy as long as the sample size exceeds $\widetilde{O}\left(\frac{C_\mathrm{r}^\star H^4 S(A+B)}{\varepsilon^2}f(\sigma^+,\sigma^-,H)\right)$ with an $\varepsilon$-independent burn-in cost. This significantly improves upon the prior art [5] on state $S$ and action $\{A, B\}$, and further delineates the impact of the uncertainty levels $\{\sigma^+, \sigma^-\}$.

**(iii) Information-theoretic sample complexity lower bound**: We establish a tight lower bound for RTZMGs, revealing at least $\Omega\left(C_\mathrm{r}^\star SH^4(A+B)/\varepsilon^2\right)$ samples are needed for an $\varepsilon$-optimal robust NE policy under the uncertainty level $\min\{\sigma^+, \sigma^-\} \lesssim \frac{1}{H}$, and at least $\Omega\left(C_\mathrm{r}^\star SH^3(A+B)/(\varepsilon^2\min\{\sigma^+,\sigma^-\})\right)$ samples are needed under $\min\{\sigma^+, \sigma^-\} \gtrsim \frac{1}{H}$. Comparing these upper and lower bounds confirms the optimality of RTZ-VI-LCB w.r.t. state $S$ and actions $\{A, B\}$ in sample complexity across uncertainty levels.

**(iv) Extension to multi-agent RL**: We generalize RTZ-VI-LCB to *Multi-RTZ-VI-LCB* for robust multi-player general-sum Markov games, and achieve an $\varepsilon$-optimal robust NE policy for $\widetilde{O}\left(C_\mathrm{r}^\star H^4 S\sum_{i=1}^m A_i\min\left\{\{(H\sigma_i-1+(1-\sigma_i)^H)/(\sigma_i)^2\}_{i=1}^m, H\right\}/\varepsilon^2\right)$ samples with $M$ players, $A_i$ actions, and uncertainty level $\sigma_i$ per player.

## 1.2 Related Work

This section reviews a curated selection of related research focusing on provably tabular RL.

**Finite-sample studies of standard TZMGs.** Markov games (MGs), or stochastic games, were proposed in the early 1950s [32]. Then, extensive research has been conducted, and MARL has gained significant attention [30], particularly around Nash equilibrium [27, 23]. Numerous MARL algorithms with provable convergence and asymptotic guarantees have been developed [31]. More recent work has focused on creating algorithms for standard MARL with non-asymptotic guarantees through finite-sample analysis. In this area, most efforts to compute Nash equilibria are focused on TZMGs. The studies in [2] and [41] were the first to provide non-asymptotic sample complexity guarantees for model-based (e.g., VI-Explore and VI-ULCB) and model-free algorithms (e.g., OMNI-VI). Further improvements in sample complexity have been explored [10, 8, 28, 13, 26].

**Robustness in MARL.** Although progress has been made in MARL, existing algorithms may struggle when faced with environmental uncertainties, leading to significantly deviated equilibria. MARL robustness against uncertainties has drawn attention in different parts of MGs [38], including state [49], environment (reward and transition dynamics), agent types [47], and other agents' policies [20]. A typical method to address robustness against uncertainties of the environment is distributionally robust optimization (DRO), which is a method predominantly explored in supervised learning [3, 14, 6]. The application of DRO to manage model uncertainty in single-agent RL [17] has attracted considerable attention. However, when extended to MARL, researchers formulated the problem as robust MGs armed with DRO and developed a relatively understudied field with only a few proved algorithms [5, 21, 29, 48, 34]. Thus, relevant algorithms based on partial coverage of datasets while considering the uncertainty level are lacking.

**Single-agent robust offline RL.** In single-agent offline RL, addressing uncertainties of environments using DRO—such as robust Markov decision processes (MDPs) and distributionally robust dynamic programming—has attracted considerable interest in both theoretical research and practical applications [16]. Recent work has focused on the finite-sample performance of provable robust offline RL algorithms, exploring different divergence functions for uncertainty sets, various sampling mechanisms, and related challenges [5]. It has been shown that addressing robust MDPs does not demand more samples compared with those needed for standard MDPs [33]. However, RTZMGs present additional complexities beyond those in robust single-agent offline RL.

## 2 Problem Formulation

This paper focuses on offline RTZMGs, which is a robust version of standard offline TZMGs by taking environmental uncertainties into consideration. RTZMGs form a broader class than standard TZMGs, accommodating various prescribed environmental uncertainty sets. In RTZMGs, the dynamics extend standard TZMGs by incorporating two players, as well as their respective situations that determine the worst-case transitions. We investigate an efficient algorithm to achieve robustness and optimal sample complexity on state $S$ and actions $\{A, B\}$ under partial coverage of the state-action space.

An RTZMG under the finite-horizon setting can be defined as $\mathcal{MG}_r = \left\{\mathcal{S}, \mathcal{A}, \mathcal{B}, \mathcal{U}_\rho^{\sigma^+}\left(P^0\right), \mathcal{U}_\rho^{\sigma^-}\left(P^0\right), r, H\right\}$, where $\mathcal{S} := \{1, \cdots, S\}$ is the state space of size $S$; $(\mathcal{A} := \{1, \cdots, A\}, \mathcal{B} := \{1, \cdots, B\})$ denotes the action spaces of the max-player and the min-player with sizes $A$ and $B$, respectively; $H$ is the horizon length; $r = \{r_h\}_{h=1}^H$ represents the immediate reward obtained at time step $h$. Specifically, $r_h(s, a, b)$ is assumed to be deterministic on a state-action pair $(s, a, b)$ and falls within the range $[0, 1]$. This reward can represent both the gain of the max-player and the loss of the min-player. Here, $\mathcal{U}_\rho^{\sigma^+}(P^0)$ and $\mathcal{U}_\rho^{\sigma^-}(P^0)$ represent the uncertainty sets for the max-player and min-player, respectively.

Unlike standard TZMGs that assume a fixed transition kernel, these uncertainty sets account for bounded perturbations in the transition kernel and enable modeling of environmental uncertainties. These uncertainty sets are centered on a nominal kernel $P^0 : \mathcal{S} \times \mathcal{A} \times \mathcal{B} \mapsto \Delta(\mathcal{S})$, with their size and shape defined by a distance metric $\rho$ and radius parameters $\sigma^+ > 0$ and $\sigma^- > 0$. Considering players' individual properties, both players can independently define their uncertainty sets $\mathcal{U}_\rho^{\sigma^+}(P^0)$

and $\mathcal{U}_\rho^{\sigma^-}(P^0)$, by specifying different sizes ($\sigma^+ > 0$ and $\sigma^- > 0$) and potentially using distinct divergence functions ($\rho$) to shape these sets. For illustration convenience, we consider the same divergence function for both players in this paper.

**Uncertainty set with *two-player* $(s, a, b)$-*rectangularity*.** According to the transition kernel uncertainty sets $\mathcal{U}_\rho^{\sigma^+}(P^0)$ and $\mathcal{U}_\rho^{\sigma^-}(P^0)$ defined above, we adapt the *rectangularity* condition to a two-player setting inspired by [33, 17], termed *two-player-wise* $(s, a, b)$-*rectangularity*. The adaptation enhances computational tractability and facilitates the robust version of Bellman recursions. It permits each player to select its uncertainty set independently, which can be decomposed for each state-action pair into a product of subsets. Thus, the uncertainty sets $\mathcal{U}_\rho^{\sigma^+}(P^0)$ and $\mathcal{U}_\rho^{\sigma^-}(P^0)$ for the two players, adhering to *two-player-wise* $(s, a, b)$-*rectangularity*, are mathematically defined as

$$\mathcal{U}_\rho^{\sigma^+}\left(P^0\right) := \otimes\, \mathcal{U}_\rho^{\sigma^+}\left(P^0_{h,s,a,b}\right), \quad \mathcal{U}_\rho^{\sigma^-}\left(P^0\right) := \otimes\, \mathcal{U}_\rho^{\sigma^-}\left(P^0_{h,s,a,b}\right), \tag{1}$$

where $\mathcal{U}_\rho^{\sigma^+}\left(P^0_{h,s,a,b}\right) := \left\{P_{h,s,a,b} \in \Delta(\mathcal{S}) : \rho\left(P_{h,s,a,b}, P^0_{h,s,a,b}\right) \leq \sigma^+\right\}$, $\otimes$ represents the Cartesian product, and $\mathcal{U}_\rho^{\sigma^-}\left(P^0_{h,s,a,b}\right)$ can be defined similarly. We define a vector of the transition kernel $P$ or $P^0$ at any state-action pair $(s, a, b)$ as

$$P_{h,s,a,b} := P_h(\cdot \,|\, s, a, b) \in \mathbb{R}^{1 \times S}, \quad P^0_{h,s,a,b} := P^0_h(\cdot \,|\, s, a, b) \in \mathbb{R}^{1 \times S}. \tag{2}$$

Here, the distance function $\rho$ for each player's uncertainty set can be selected from various options that quantify differences between probability vectors. These include $f$-divergences (e.g., KL divergence, TV distance, and chi-square) [44], the Wasserstein distance [42], and $\ell_q$ norms [9].

**Offline dataset.** Let $\mathcal{D}$ be a dataset consisting of $K$ episodes under independence, with each episode produced by implementing a behavior policy $\{\mu_h^{\mathsf{n}}, \nu_h^{\mathsf{n}}\}_{h=1}^H$ in a nominal MDP $\mathcal{M}^0 = \left(\mathcal{S}, \mathcal{A}, \mathcal{B}, H, P^0 := \{P_h^0\}_{h=1}^H, \{r_h\}_{h=1}^H\right)$. For $1 \leq k \leq K$, the $k$-th episode $\left(s_1^k, a_1^k, b_1^k, \ldots, s_H^k, a_H^k, b_H^k, s_{H+1}^k\right)$ is generated as follows:

$$s_1^k \sim \varrho^{\mathsf{n}}, \quad a_h^k \sim \mu_h^{\mathsf{n}}(\cdot \,|\, s_h^k), \quad b_h^k \sim \nu_h^{\mathsf{n}}(\cdot \,|\, s_h^k), \quad s_{h+1}^k \sim P_h^0(\cdot \,|\, s_h^k, a_h^k, b_h^k), \quad 1 \leq h \leq H. \tag{3}$$

Throughout this paper, $\varrho^{\mathsf{n}}$ denotes the initial distribution related to a historical dataset. We use the short-hand notation for the occupancy distribution with respect to (w.r.t.) the behavior policy $(\mu^{\mathsf{n}}, \nu^{\mathsf{n}})$ as: $\forall (h, s, a, b) \in [H] \times \mathcal{S} \times \mathcal{A} \times \mathcal{B}$,

$$d_h^{\mu^{\mathsf{n}}, \nu^{\mathsf{n}}, P^0}(s) := \mathbb{P}(s_h = s | s_1 \sim \varrho^{\mathsf{n}}, \mu^{\mathsf{n}}, \nu^{\mathsf{n}}, P^0); \ d_h^{\mu^{\mathsf{n}}, \nu^{\mathsf{n}}, P^0}(s, a, b) := d_h^{\mu^{\mathsf{n}}, \nu^{\mathsf{n}}, P^0}(s) \mu_h^{\mathsf{n}}(a \,|\, s)\, \nu_h^{\mathsf{n}}(b \,|\, s), \tag{4}$$

which are simplified to $d_h^{\mathsf{n}, P^0}(s) = d_h^{\mu^{\mathsf{n}}, \nu^{\mathsf{n}}, P^0}(s)$ and $d_h^{\mathsf{n}, P^0}(s, a, b) = d_h^{\mu^{\mathsf{n}}, \nu^{\mathsf{n}}, P^0}(s, a, b)$. Similarly, for any product policy $(\mu, \nu)$, we define: $\forall (h, s, a, b) \in [H] \times \mathcal{S} \times \mathcal{A} \times \mathcal{B}$

$$d_h^{\mu, \nu, P}(s) := \mathbb{P}(s_h = s \,|\, s_1 \sim \varrho, \mu, \nu, P); \ d_h^{\mu, \nu, P}(s, a, b) := d_h^{\mu, \nu, P}(s) \mu_h(a \,|\, s)\, \nu_h(b \,|\, s). \tag{5}$$

**Robust value functions.** In RTZMGs, players seek to optimize their worst-case performance across all possible transition kernels within their respective uncertainty sets $\mathcal{U}_\rho^{\sigma^+}\left(P^0\right)$ and $\mathcal{U}_\rho^{\sigma^-}\left(P^0\right)$. For any product policy $(\mu \times \nu) \in \Delta(\mathcal{A} \times \mathcal{B})$, the max-player's worst-case performance at time step $h$ is measured with the *robust value function* $V_h^{\mu, \nu, \sigma^+}$ and the *robust Q-function* $Q_h^{\mu, \nu, \sigma^+}$, $\forall (h, s, a, b) \in [H] \times \mathcal{S} \times \mathcal{A} \times \mathcal{B}$, as given by

$$V_h^{\mu, \nu, \sigma^+}(s) := \inf_{P \in \mathcal{U}_\rho^{\sigma^+}(P^0)} V_h^{\mu, \nu, P}(s), \quad Q_h^{\mu, \nu, \sigma^+}(s, a, b) := \inf_{P \in \mathcal{U}_\rho^{\sigma^+}(P^0)} Q_h^{\mu, \nu, P}, \tag{6a}$$

$$V_h^{\mu, \nu, \sigma^-}(s) := \sup_{P \in \mathcal{U}_\rho^{\sigma^-}(P^0)} V_h^{\mu, \nu, P}(s), \quad Q_h^{\mu, \nu, \sigma^-}(s, a, b) := \sup_{P \in \mathcal{U}_\rho^{\sigma^-}(P^0)} Q_h^{\mu, \nu, P}, \tag{6b}$$

where

$$V_h^{\mu, \nu, P}(s) := \mathbb{E}_{\mu, \nu, P}\left[\sum_{t=h}^H r_t(s_t, a_t, b_t) \,|\, s_h = s\right];$$

$$Q_h^{\mu, \nu, P}(s, a, b) := \mathbb{E}_{\mu, \nu, P}\left[\sum_{t=h}^H r_t(s_t, a_t, b_t) \,|\, s_h = s, a_h = a, b_h = b\right].$$

**Robust Bellman equations.** Based on the robust value functions in (6), RTZMGs include a robust version of the Bellman equation, dubbed *robust Bellman equation*. The robust value functions $V_h^{\mu,\nu,\sigma^+}(s)$ for the max-player with any product policy $(\mu,\nu)$ satisfy: $\forall (h,s) \in [H] \times \mathcal{S}$,

$$V_h^{\mu,\nu,\sigma^+}(s) = \mathbb{E}_{(a,b)\sim(\mu_h(a),\nu_h(a))}\left[r_h(s,a,b) + \inf_{P \in \mathcal{U}_\rho^{\sigma^+}(P_{h,s,a,b}^0)} PV_{h+1}^{\mu,\nu,\sigma^+}\right]. \tag{7}$$

Likewise, $V_h^{\mu,\nu,\sigma^-}(s)$ can be obtained for the min-player. Note that the robust Bellman equations are intrinsically connected to the *two-player-wise $(s,a,b)$-rectangularity* condition (see (1)) applied to the uncertainty set. This condition separates the dependencies of uncertainty subsets among different time steps, the players, and state-action pairs, thus leading to the Bellman recursion.

**Optimal robust policy.** Again, based on (6), we define the maximum robust value function of the max-player under the fixed opponent policy as: $\forall (h,s) \in [H] \times \mathcal{S}$,

$$V_h^{\star,\nu,\sigma^+}(s) := \max_{\mu:\mathcal{S}\times[H]\mapsto\Delta(\mathcal{A})} V_h^{\mu,\nu,\sigma^+}(s) = \max_{\mu:\mathcal{S}\times[H]\mapsto\Delta(\mathcal{A})} \inf_{P\in\mathcal{U}_\rho^{\sigma^+}(P^0)} V_h^{\mu,\nu,P}(s).$$

The maximum robust value function for the min-player can be obtained similarly.

As proved in [5], there is at least one policy, denoted by $\mu_h^\star(s) : \mathcal{S} \times [H] \mapsto \Delta(\mathcal{A})$ (for the max-player) and $\nu_h^\star(s) : \mathcal{S} \times [H] \mapsto \Delta(\mathcal{B})$ (for the min-player), corresponding to the *robust best-response policy*. These policies can simultaneously achieve $V_h^{\star,\nu,\sigma^+}(s)$ (for the max-player) and $V_h^{\mu,\star,\sigma^-}(s)$ (for the min-player) for all $s \in \mathcal{S}$ and $h \in [H]$.

**Robust Nash equilibrium.** We introduce the robust variant of standard solution concepts—robust NE for RTZMGs. A product policy $(\mu,\nu)$ is considered a *robust NE* if: $\forall(s) \in \mathcal{S}$

$$V_h^{\star,\nu,\sigma^+}(s) = V_h^{\star,\sigma^+}(s); \quad V_h^{\mu,\star,\sigma^-}(s) = V_h^{\star,\sigma^-}(s). \tag{8}$$

A robust NE signifies that given the product policy $(\mu,\nu)$ of the opponents, no player can enhance their outcome by deviating from their current policy unilaterally when each player accounts for the worst-case scenario within their uncertainty set $\mathcal{U}_\rho^{\sigma^+}(P^0)$ or $\mathcal{U}_\rho^{\sigma^-}(P^0)$.

Since finding exact robust equilibria can be complex and may not always be feasible, practitioners often seek approximate equilibria. In this context, a product policy $(\mu \times \nu) \in \Delta(\mathcal{A} \times \mathcal{B})$ can be termed an $\varepsilon$-*robust NE* if

$$\text{Gap}(\mu,\nu) := \max\{V_1^{\star,\nu,\sigma^+}(\varrho) - V_1^{\star,\sigma^+}(\varrho), \quad V_1^{\star,\sigma^-}(\varrho) - V_1^{\mu,\star,\sigma^-}(\varrho)\} \leq \varepsilon, \tag{9}$$

where

$$V_1^{\star,\nu,\sigma^+}(\varrho) = \mathbb{E}_{s\sim\varrho}V_1^{\star,\nu,\sigma^+}(s), \quad V_1^{\star,\sigma^+}(\varrho) = \mathbb{E}_{s\sim\varrho}V_1^{\star,\sigma^+}(s).$$

The definitions of $V_1^{\mu,\star,\sigma^-}(\varrho)$ and $V_1^{\star,\sigma^-}(\varrho)$ can be obtained similarly. The existence of a robust NE has been proved for general divergence functions in the uncertainty set in [5].

**Our Goal** With a dataset collected from the nominal environment, our objective is to find a solution yielding an $\varepsilon$-robust NE for RTZMG w.r.t. a specified uncertainty set $\mathcal{U}(P^0)$ around the nominal kernel, minimizing the number of samples required under partial coverage of the state-action space.

## 3 Algorithm Design

In this section, we propose an efficient model-based algorithm, RTZ-VI-LCB, to learn a robust NE policy. Notably, RTZ-VI-LCB achieves a near-optimal sample complexity with robustness, which is designed for offline RTZMGs within the finite-horizon setting.

### 3.1 Building an Empirical Nominal MDP

According to the empirical frequencies of state transitions, we can construct an empirical estimate $\widehat{P}^0 = \{\widehat{P}_h^0\}_{h=1}^H$ of $P^0$, where $\forall(h,s,a,b,s') \in [H] \times \mathcal{S} \times \mathcal{A} \times \mathcal{B} \times \mathcal{S}$

$$\widehat{P}_h^0(s' \mid s,a,b) = \begin{cases} \frac{\sum_{i=1}^N \mathbb{1}\{(s_i,a_i,b_i,s_i')=(s,a,b,s')\}}{N_h(s,a,b)}, & \text{if } N_h(s,a,b) > 0; \\ 1/S, & \text{if } N_h(s,a,b) = 0, \end{cases} \tag{10}$$

$$\widehat{r}_h(s,a,b) = \begin{cases} r_h(s,a,b), & \text{if } N_h(s,a,b) > 0; \\ 0, & \text{if } N_h(s,a,b) = 0, \end{cases} \qquad (11)$$

where $N_h(s,a,b)$ represents the total number of sample transitions from $(s,a,b)$ at step $h$, and

$$N_h(s,a,b) := \sum_{i=1}^{N} \mathbb{1}\{(s_i, a_i, b_i) = (s,a,b)\}. \qquad (12)$$

Although it is feasible to decompose the historical dataset $\mathcal{D}$ into sample transitions, the dependencies between transitions within the same episode introduce significant complexities to the analysis.

---

**Algorithm 1** Two-stage subsampling for RTZ-VI-LCB.

---

**input** Dataset $\mathcal{D}$, probability $\delta$.

1: **Step 1: Data Partitioning.** Split $\mathcal{D}$ into two equal-sized subsets, $\mathcal{D}^{\mathsf{m}}$ and $\mathcal{D}^{\mathsf{a}}$, each containing $K/2$ trajectories.

2: **Step 2: Defining Transition Bounds.** For step $h$ and state $s$, denote the number of transitions from $\mathcal{D}^{\mathsf{m}}$ (resp. $\mathcal{D}^{\mathsf{a}}$) as $N_h^{\mathsf{m}}(s)$ (resp. $N_h^{\mathsf{a}}(s)$). Construct the trimmed count as:

$$N_h^{\mathsf{t}}(s) := \max\left\{ N_h^{\mathsf{a}}(s) - 10\sqrt{N_h^{\mathsf{a}}(s)\log\frac{HS}{\delta}}, 0 \right\}; \qquad (13)$$

3: **Step 3: Generating Subsampled Dataset.** Randomly sample transitions (quadruples of the form $(s,a,b,h,s')$) from $\mathcal{D}^{\mathsf{m}}$ uniformly. For each $(s,h) \in \mathcal{S} \times [H]$, include $\min\{N_h^{\mathsf{t}}(s), N_h^{\mathsf{m}}(s)\}$ transitions in the new dataset $\mathcal{D}^{\mathsf{t}}$.

**output** Set $\mathcal{D}_0 = \mathcal{D}^{\mathsf{t}}$.

---

Inspired by [24], we design a new two-stage subsampling method for RTZMGs, as shown in Algorithm 1. This method effectively reduces statistical dependencies, resulting in a distributionally equivalent dataset $\mathcal{D}_0$ composed of independent samples. The property of $\mathcal{D}_0$ is summarized in the following lemma and formally proved in Appendix C.

**Lemma 3.1.** *The dataset produced by the two-stage subsampling method is distributionally identical to $\mathcal{D}_0$ with probability of at least $1 - 8\delta$, where $\{N_h(s,a,b)\}$ are independent of the sample transitions in $\mathcal{D}^0$ and obey:* $\forall (h,s,a,b) \in [H] \times \mathcal{S} \times \mathcal{A} \times \mathcal{B}$,

$$N_h(s,a,b) \geq \frac{K d_h^{\mathsf{n}}(s,a,b)}{8} - 5\sqrt{K d_h^{\mathsf{n}}(s,a,b)\log\frac{KH}{\delta}}. \qquad (14)$$

By applying the two-fold sampling method, we can treat the dataset $\mathcal{D}_0$ as having independent samples, simplifying the analysis significantly as supported by Lemma 3.1.

## 3.2 Optimistic Variant of Robust Value Iteration with Lower Confidence Bounds

We propose a model-based algorithm, RTZ-VI-LCB, for solving RTZMGs using an approximate $\widehat{P}^0$ for $P^0$, which is the nominal transition kernel. Specifically, we introduce value iteration with lower confidence bounds for RTZMGs to compute a robust NE for two players, along with a data-informed penalty, as summarized in Algorithm 2.

Our algorithm begins at the final time step $h = H$ and proceeds backward through $h = H - 1, H - 2, \ldots, 1$. Following from single-agent offline RL algorithms [24, 19], we design an optimistic robust Q-value for all $(h,s,a,b) \in [H] \times \mathcal{S} \times \mathcal{A} \times \mathcal{B}$ as

$$\widehat{Q}_h^+(s,a,b) = \min\left\{ \widehat{r}_h(s,a,b) + \inf_{P \in \mathcal{U}^{\sigma^+}\left(\widehat{P}_{h,s,a,b}^0\right)} P\widehat{V}_{h+1}^+ + \beta_h\left(s,a,b,\widehat{V}_{h+1}^+\right), H \right\}; \quad (15a)$$

$$\widehat{Q}_h^-(s,a,b) = \max\left\{ \widehat{r}_h(s,a,b) + \sup_{P \in \mathcal{U}^{\sigma^-}\left(\widehat{P}_{h,s,a,b}^0\right)} P\widehat{V}_{h+1}^- - \beta_h\left(s,a,b,\widehat{V}_{h+1}^-\right), 0 \right\}, \quad (15b)$$

to estimate the robust Q-function at time step $h \in [H]$ as $\widehat{Q}_h^+$ and $\widehat{Q}_h^-$.

**Dual problem.** Solving (15) directly is computationally intensive because it requires optimizing over an $S$-dimensional probability simplex, which becomes exponentially more difficult as the state

space size $S$ increases. Fortunately, strong duality for TV distance allows us to tackle this problem by solving its dual [17] as

$$\inf_{P \in \mathcal{U}^{\sigma^+}\left(\widehat{P}^0_{h,s,a,b}\right)} P\widehat{V}^+_{h+1} = \max_{\alpha \in [\min_s \widehat{V}^+_{h+1}, \max_s \widehat{V}^+_{h+1}]} \left\{ \widehat{P}^0_{h,s,a,b} \left[\widehat{V}^+_{h+1}\right]_\alpha - \sigma^+ \left(\alpha - \min_{s'} \left[\widehat{V}^+_{h+1}\right]_\alpha (s')\right)\right\}, \quad (16)$$

where $\left[\widehat{V}^+_{h+1}\right]_\alpha$ denotes the clipped versions of $\widehat{V}^+_{h+1} \in \mathbb{R}^S$ based on some level $\alpha \geq 0$, as follows.

$$\left[\widehat{V}^+_{h+1}\right]_\alpha (s) := \begin{cases} \alpha, & \text{if } \widehat{V}^+_{h+1}(s) > \alpha; \\ \widehat{V}^+_{h+1}(s), & \text{otherwise.} \end{cases} \quad (17)$$

Moreover, $\sup_{P \in \mathcal{U}^{\sigma^-}\left(\widehat{P}^0_{h,s,a,b}\right)} P\widehat{V}^-_{h+1}$ can be defined similarly. See Appendix B for details.

**Penalty term.** We design a data-driven penalty term, $\beta_h(s, a, b, \widehat{V})$, to account for uncertainty in value estimates, resulting in an optimistic robust $Q$-function estimate. To achieve this, we utilize a Bernstein-style penalty, which effectively captures the variance structure over time [24]. In particular, for any $(s, a, b, h) \in \mathcal{S} \times \mathcal{A} \times \mathcal{B} \times [H]$ and $\delta \in (0, 1)$, the penalty term $\beta_h(s, a, b, \widehat{V})$ is defined as

$$\beta_h \left(s, a, b, \widehat{V}\right) = \min \left\{ \max \left\{ \sqrt{\frac{C_\mathsf{n} \log \frac{KH}{\delta}}{N_h (s, a, b)} \mathsf{Var}_{\widehat{P}^0_{h,s,a,b}}(\widehat{V})}, \frac{2C_\mathsf{n} H \log \frac{KH}{\delta}}{N_h (s, a, b)} \right\}, H \right\}, \quad (18)$$

where $C_\mathsf{n}$ is some universal constant, and

$$\mathsf{Var}_{\widehat{P}^0_{h,s,a,b}} \left(\widehat{V}\right) := \widehat{P}^0_{h,s,a,b} \widehat{V}^2 - (\widehat{P}^0_{h,s,a,b} \widehat{V})^2. \quad (19)$$

Note that we choose $\widehat{P}^0$ in the variance term $\mathsf{Var}_{\widehat{P}^0_{h,s,a,b}}(\widehat{V})$, as opposed to $P^0$, since we have no access to the true transition kernel $P^0$. The penalty term $\beta_h \left(s, a, b, \widehat{V}\right)$ is crafted to address the unique structure of RTZMGs, distinguishing it from the penalty terms used in standard offline TZMGs [10, 24]. In particular, it provides a tight upper bound on statistical uncertainty, considers the non-linear and implicit dependency introduced by the uncertainty set $\mathcal{U}(P^0)$, and addresses challenges not present in standard MDPs.

**Policy estimation.** We update the policies using the estimated $Q$-functions with uncertainty, as described in line 5 of Algorithm 2. For any matrix $\mathbf{N} \in \mathbb{R}^{A \times B}$, the function $\mathsf{ComputNash}(\mathbf{N})$ returns a solution $(\widehat{w}, \widehat{z})$ to the minimax problem $\max_{w \in \Delta(\mathcal{A})} \min_{z \in \Delta(\mathcal{B})} w^\top \mathbf{N} z$ [11]. In other words, for each $s \in \mathcal{S}$, we compute the NE policies $\left(\mu^+_h(s), \nu^+_h(s)\right)$ and $\left(\mu^-_h(s), \nu^-_h(s)\right) \in \Delta(\mathcal{A}) \times \Delta(\mathcal{B})$ for the robust zero-sum matrix games with payoff matrices $\widehat{Q}^+_h(s, \cdot, \cdot)$ and $\widehat{Q}^-_h(s, \cdot, \cdot)$, respectively. Solving robust zero-sum matrix games is generally PPAD-hard as the players can potentially choose different worst-case transition kernels.

## 4 Performance Guarantees

We provide the theoretical guarantee of RTZ-VI-LCB, including the upper and lower bounds of sample complexity. **We also validate the theoretical guarantee through numerical experiments; see Appendix A.**

**Robust unilateral clipped concentrability.** For offline RTZMGs, it is essential to measure the distributional discrepancy between the historical data and the target data. Drawing on the *single-policy clipped concentrability* in the single-agent RL [24], we propose a novel criterion, *robust unilateral clipped concentrability*, to measure the distributional discrepancy for RTZMGs:

---
**Algorithm 2** Value iteration with lower confidence bounds for RTZMGs (RTZ-VI-LCB).
---
1: **Initialization**: Set uncertainty levels $\sigma^-$ and $\sigma^+$; set $\widehat{V}_h^-(s) = 0$ and $\widehat{V}_h^+(s) = H$ for all $(s,h) \in \mathcal{S} \times [H+1]$; set $\widehat{Q}_h^-(s,a,b) = 0$ and $\widehat{Q}_h^+(s,a,b) = H$ for all $(s,a,b,h) \in \mathcal{S} \times \mathcal{A} \times \mathcal{B} \times [H+1]$.
2: **Compute** the empirical reward function $\widehat{r}$ with (11) and empirical transition kernel $\widehat{P}_0$ with (10).
3: **for** $h = H, H-1, \ldots, 1$ **do**
4:     **Update** the robust Q-value estimate as (15) with $\beta_h(s,a,b,V)$ defined in (18).
5:     **Compute** Nash policy for each $s \in \mathcal{S}$ as

$$\left(\mu_h^+(s), \nu_h^+(s)\right) = \mathsf{ComputNash}\left(\widehat{Q}_h^+(s,\cdot,\cdot)\right);$$

$$\left(\mu_h^-(s), \nu_h^-(s)\right) = \mathsf{ComputNash}\left(\widehat{Q}_h^-(s,\cdot,\cdot)\right).$$

6:     **Update** the robust value estimate for each $s \in \mathcal{S}$ as

$$\widehat{V}_h^-(s) = \mathbb{E}_{a\sim\mu_h^-(s),b\sim\nu_h^-(s)}\left[\widehat{Q}_h^-(s,a,b)\right]; \ \widehat{V}_h^+(s) = \mathbb{E}_{a\sim\mu_h^+(s),b\sim\nu_h^+(s)}\left[\widehat{Q}_h^+(s,a,b)\right].$$

7: **end for**
**output** The policy pair $(\widehat{\mu}, \widehat{\nu})$, where $\widehat{\mu} = \{\mu_h^-\}_{h=1}^H$ and $\widehat{\nu} = \{\nu_h^+\}_{h=1}^H$.
---

**Definition 4.1** (Robust unilateral clipped concentrability). *We define $C_{\mathrm{r}}^\star \in \left[\frac{1}{S(A+B)}, \infty\right]$ as the smallest value that satisfies*

$$\max\left\{\sup_{(\mu,s,a,b,h,P)\in\Delta(\mathcal{A})\times\mathcal{S}\times\mathcal{A}\times\mathcal{B}\times[H]\times\mathcal{U}^{\sigma^-}(P^0)} \frac{\min\left\{d_h^{\mu,\nu^\star,P}(s,a,b), \frac{1}{S(A+B)}\right\}}{d_h^{\mathsf{n},P^0}(s,a,b)},\right.$$
$$\left.\sup_{(\nu,s,a,b,h,P)\in\Delta(\mathcal{B})\times\mathcal{S}\times\mathcal{A}\times\mathcal{B}\times[H]\times\mathcal{U}^{\sigma^+}(P^0)} \frac{\min\left\{d_h^{\mu^\star,\nu,P}(s,a,b), \frac{1}{S(A+B)}\right\}}{d_h^{\mathsf{n},P^0}(s,a,b)}\right\} \le C_{\mathrm{r}}^\star \quad (20)$$

*for the behavior policies of the historical dataset $\mathcal{D}$ satisfies, and refer to it as the robust unilateral clipped concentrability coefficient. For consistency, we define the convention "$0/0 = 0$".*

Notably, if $d_h^{\mu,\nu^\star,P}(s,a,b)$ or $d_h^{\mu^\star,\nu,P}(s,a,b)$ is larger than $1/(S(A+B))$, the robust unilateral clipped concentrability assumption does not require the data distribution $d_h^{\mathsf{n},P^0}(s,a,b)$ to scale with $d_h^{\mu,\nu^\star,P}(s,a,b)$ or $d_h^{\mu^\star,\nu,P}(s,a,b)$ proportionally. Estimating the concentrability coefficient $C_{\mathrm{r}}^\star$ from offline data is generally information-theoretically impossible, as demonstrated even in the single-agent case by the example construction in Section 3.4 of [24]. Fortunately, this does not affect the execution of our algorithm.

Next, we outline the principal theoretical findings concerning the sample complexity of learning robust NE in RTZMGs, including an upper bound for RTZ-VI-LCB (Algorithm 2) and an information-theoretic lower bound. We start with the finite-sample guarantee for RTZ-VI-LCB, with the proof provided in Appendix D.

**Theorem 4.2** (Upper bound for RTZ-VI-LCB). *Under the TV uncertainty set $\mathcal{U}^{\sigma^+}(\cdot)$ and $\mathcal{U}^{\sigma^-}(\cdot)$ defined in (2) with $\sigma^+$, $\sigma^- \in (0,1]$, define $d_{\mathsf{m}}^{\mathsf{n}} = \min_{h,s,a,b}\{d_h^{\mathsf{n}}(s,a,b) : d_h^{\mathsf{n}}(s,a,b) > 0\}$, and let $f(\sigma^+,\sigma^-) = \min\left\{(H\sigma^+-1+(1-\sigma^+)^H)/(\sigma^+)^2, (H\sigma^--1+(1-\sigma^-)^H)/(\sigma^-)^2, H\right\}$. Consider any $\delta \in (0,1)$ and any RTZMG $\mathcal{MG}_{\mathrm{r}} = \left\{\mathcal{S}, \mathcal{A}, \mathcal{B}, \mathcal{U}^{\sigma^+}(P^0), \mathcal{U}^{\sigma^-}(P^0), r, H\right\}$. For sufficient large constants $c_0, c_1 > 0$, with probability of at least $1-\delta$, we can achieve*

$$\mathrm{Gap}(\widehat{\mu},\widehat{\nu}) \le c_1\sqrt{\frac{C_{\mathrm{r}}^\star H^3 S(A+B)\log\frac{KH}{\delta}}{K} f(\sigma^+,\sigma^-,H)} \quad (21)$$

*with the total number of samples $T$ exceeding*

$$T = KH \ge c_0 \frac{H^2 S(A+B)}{d_{\mathsf{m}}^{\mathsf{n}}} \log\frac{KH}{\delta} f(\sigma^+,\sigma^-,H). \quad (22)$$

*Remark* 4.3. Under a generative model with uniform sampling, the visitation distribution becomes explicit; that is, $d_h^{n,P^0}(s,a,b) = \frac{1}{SAB}$. In this case, it is sufficient to choose $C_r^\star = \min\{A,B\}$, which can be readily verified to satisfy Eq. (20). This choice leads to the sample complexity $\widetilde{O}\left(\frac{H^4 SAB}{\varepsilon^2} \cdot f(\sigma^+, \sigma^-, H)\right)$, which matches the bound established in [34] and confirms the efficiency of our approach in the generative model setting.

We derive an information-theoretic lower bound of sample complexity, with its proof in Appendix E.

**Theorem 4.4** (Lower bound for RTZMGs). *Consider any tuple $\mathcal{MG}_r = \left\{\mathcal{S}, \mathcal{A}, \mathcal{B}, \mathcal{U}^{\sigma^+}(P^0), \mathcal{U}^{\sigma^-}(P^0), r, H\right\}$ obeying $H > 16\log 2$ and $\sigma^+, \sigma^- \in (0, 1 - c_0]$ with any small efficiently positive constant $0 < c_0 \leq \frac{1}{4}$. Let*

$$\varepsilon \leq \begin{cases} \frac{c_2}{H}, & \text{if } \max\{\sigma^+, \sigma^-\} \leq \frac{c_2}{2H}; \\ 1, & \text{otherwise,} \end{cases} \tag{23}$$

*for any $c_2 \leq \frac{1}{4}$. With an initial state distribution $\varrho$, we can construct a set of RTZMGs $\left\{\mathcal{M}_f^\phi | f \in \mathcal{F} = \{0, 1, \cdots, SA - 1\}, \phi = [\phi_h]_{1 \leq h \leq H} \in \Phi \subseteq \{0, 1\}^H\right\}$. For any dataset with $K$ independent sample trajectories of length $H$ per trajectory satisfying $C \leq C_r^\star \leq 2C$, we have*

$$\inf_{\widehat{\mu}, \widehat{\nu}} \max_{(f, \phi) \in \mathcal{F} \times \Phi} \left\{\mathbb{P}_\phi\left(\text{Gap}(\widehat{\mu}, \widehat{\nu}) > \varepsilon\right)\right\} \geq 1/8, \tag{24}$$

*provided that*

$$T = KH \leq \frac{cC_r^\star H^3 S(A + B) \min\left\{\frac{1}{\min\{\sigma^+, \sigma^-\}}, H\right\}}{\varepsilon^2}. \tag{25}$$

*Here, $c$ is an efficiently small constant. The infimum is obtained over all estimators $(\widehat{\mu}, \widehat{\nu})$.*

These theorems offer the following key implications:

(i) **Theorem 4.2** demonstrates that the proposed RTZ-VI-LCB algorithm can attain an $\varepsilon$-robust NE solution when sample size exceeds $\widetilde{O}\left(\frac{C_r^\star H^4 S(A+B)}{\varepsilon^2} f(\sigma^+, \sigma^-, H)\right)$, suggesting that the sample efficiency for robust offline TZMGs is strongly influenced by the dataset quality (quantified by $C_r^\star$) and problem structure of RTZMGs (reflected in the occupancy distributions $d_m^n$). If $C_r^\star$ is as small as $\frac{1}{S(A+B)}$, the upper bound of the sample complexity exhibits a weaker dependency on actions $\{A, B\}$ and state $S$. Combining this upper bound with the lower bound in Theorem 4.4 shows that RTZ-VI-LCB's sample complexity is optimal w.r.t. key factors $S$, $A$, $B$ and $\varepsilon$. This is the first optimal sample complexity upper bound for offline RTZMGs, regarding state $S$ and actions $\{A, B\}$.

(ii) **Theorem 4.4** conveys two important points. When the uncertainty level is small (i.e., $\min\{\sigma^+, \sigma^-\} \lesssim \frac{1}{H}$), no algorithm can find an $\varepsilon$-optimal robust policy with fewer than $\Omega\left(\frac{C_r^\star S H^4(A+B)}{\varepsilon^2}\right)$ samples for all offline RTZMGs, matching the complexity requirement for non-robust offline TZMGs [18]. This implies that robust TZMGs are at least as challenging as standard TZMGs for low uncertainty. When the uncertainty level satisfies $\min\{\sigma^+, \sigma^-\} \gtrsim \frac{1}{H}$, no algorithm can find an $\varepsilon$-optimal robust policy with the numbers of samples fewer than $\Omega\left(\frac{C_r^\star S H^3(A+B)}{\varepsilon^2 \min\{\sigma^+, \sigma^-\}}\right)$. To this end, RTZ-VI-LCB is the first provably optimal algorithm on $S$ and $\{A, B\}$ for RTZMGs without requiring full coverage assumptions.

Moreover, our algorithm can be extended to multi-player general-sum MGs with $m$ players and $A_i$ actions and uncertainty level $\sigma_i$ per player; see Appendix F. We can obtain the following theoretical guarantee for this extended algorithm, named Multi-RTZ-VI-LCB:

**Theorem 4.5** (Upper bound for Multi-RTZ-VI-LCB). *Consider any $\delta \in (0, 1)$ and any robust multi-player general-sum MGs $\mathcal{MG}_r = \mathcal{M}(\mathcal{S}, \{\mathcal{A}_i\}_{i=1}^m, H, \{\mathcal{U}_\rho^{\sigma_i}(P^0)\}_{i=1}^m, \{r_i\}_{i=1}^m)$. Under the TV uncertainty set $\mathcal{U}^{\sigma_i}(\cdot)$ defined in (2) with $\sigma_i \in (0, 1]$ for $i = 1, 2, \cdots, m$. Define $d_m^n = \min_{h,s,\boldsymbol{a}} \{d_h^n(s, \boldsymbol{a}) : d_h^n(s, \boldsymbol{a}) > 0\}$, and $f(\{\sigma_i\}_{i=1}^m, H) = \min\left\{\left\{\frac{(H\sigma_i - 1 + (1-\sigma_i)^H)}{(\sigma_i)^2}\right\}_{i=1}^m, H\right\}$. For sufficient large constants $c_0, c_1 > 0$, with probability of at least $1 - \delta$, we can achieve*

$$\text{Gap}(\widehat{\pi}) \leq c_1 \sqrt{\frac{C_r^\star H^3 S \sum_{i=1}^m A_i}{K} \log \frac{KH}{\delta} f(\{\sigma_i\}_{i=1}^m, H)},$$

*with the total number of samples $T$ exceeding*

$$T = KH \geq c_0 \frac{H^2 S \sum_{i=1}^{m} A_i}{d_{\mathsf{m}}^{\mathsf{n}}} \log \frac{KH}{\delta} f(\{\sigma_i\}_{i=1}^m, H). \tag{26}$$

Theorem 4.5 demonstrates that Multi-RTZ-VI-LCB can attain an $\varepsilon$-robust NE solution when the sample size exceeds $\widetilde{O}\left( \frac{C_r^\star H^4 S \sum_{i=1}^m A_i}{\varepsilon^2} f(\{\sigma_i\}_{i=1}^m, H) \right)$, breaking the curse of multiagency.

## 5  Numerical Experiments

To effectively evaluate our algorithm, we have conducted numerical experiments on randomly generated transition kernels, following the code proposed by [39]. In particular, we adopt the parameter setting as $S = 50$, $A = B = 2$, and $H = 100$, averaged over 100 seeds. Experiments are conducted on PyTorch 2.0.0 with a single NVIDIA RTX 4090 24GB GPU. In our experiments, the robust NE at each state and timestep is computed using standard NE solvers, i.e., the Python package nashpy. Our algorithm is compatible with any exact or approximate NE solver, including computational relaxations or sampling-based methods.

As shown in Figure 1(a), the case of $K = 148 \approx e^5$ demonstrates that our proposed algorithm consistently outperforms the baseline value iteration for robust TZMGs (RTZ-VI) across all states and all sample sizes. This trend remains consistent across other values of $K$ as well. Moreover, we have plotted the sub-optimality performance gap of RTZ-VI-LCB w.r.t. the sample size on a log-log scale to corroborate the scaling of the sample size on the performance gap. Fitting using linear regression leads to a slope estimate of $-0.4877$. This nicely matches the finding of our theoretical guarantee.

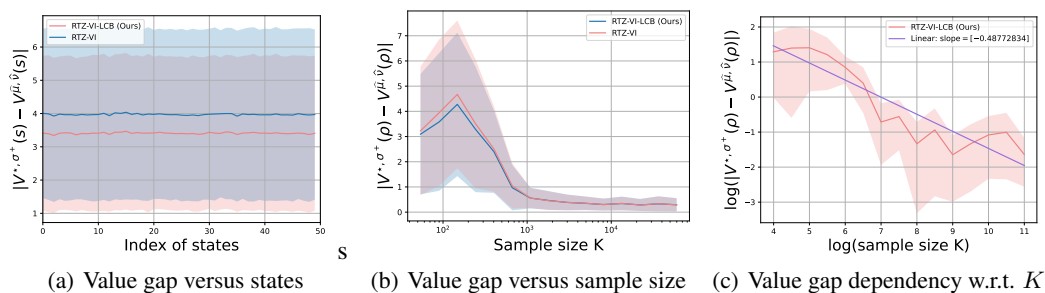

| (a) Value gap versus states | (b) Value gap versus sample size | (c) Value gap dependency w.r.t. $K$ |

Figure 1: The performances of RTZ-VI-LCB and RTZ-VI in the stochastic TZMG problem.

## 6  Conclusion

In this paper, we design an efficient robust model-based algorithm for offline RTZMGs, which is value iteration with lower confidence bounds for RTZMGs. Our algorithm integrates robust value iteration with the principle of pessimism. By imposing a tailored assumption (robust unilateral clipped concentrability) on the historical dataset to account for the distribution shift, we address robustness in the worse-case scenario of the shared environment, analyze the finite-sample complexity of the proposed RTZ-VI-LCB algorithm, and establish an information-theoretic lower bound to evaluate its optimality across various uncertainty levels. To the best of our knowledge, this is the first provably optimal algorithm for offline RTZMGs that addresses the dependency on states $S$ and actions $\{A, B\}$, while accounting for model perturbations and partial coverage. Furthermore, we extend RTZ-VI-LCB to multi-agent general-sum MGs, demonstrating a breakthrough in breaking the curse of multiagency.

**Broader Impacts**  This paper presents work whose goal is to advance the field of Reinforcement Learning. There are many potential societal consequences of our work, none of which we feel must be specifically highlighted here.

## Acknowledgements

The work was supported in part by the National Natural Science Foundation of China (NSFC) under Grants U21B2029, 62531019, and U21A20456; in part by the Zhejiang Provincial Natural Science Foundation of China under Grant LR23F010006; and in part by a Nokia Donation Project. The work of Na Li was supported by the China Scholarship Council.

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

# Contents

# A  Numerical Experiments

| $\log(KH)$ | 6 | 7 | 8 | 9 |
|---|---|---|---|---|
| RTZ-VI-LCB | **3.1699** | **2.1498** | **0.2324** | **0.0819** |
| P$^2$M$^2$PO | 3.5970 | 2.2503 | 0.2404 | 0.0890 |

These results highlight the superiority of our method across varying dataset sizes and validate the theoretical claims under controlled experimental conditions. While we focus on tabular settings in this work, our algorithm can serve as a theoretical foundation for scalable extensions based on linear or kernel-based function approximation.

# B  Preliminaries

In this appendix, we introduce the dual equivalence of robust Bellman operators and key facts about RTZMGs and their empirical counterparts, laying the foundation for our theoretical analysis.

## B.1  Dual equivalence of robust Bellman

We can compute the robust Bellman operator by solving its dual formulation rather than the original form, as long as the predefined uncertainty set is in a benign form (e.g., utilizing TV distance as the divergence function) [17, 33]. Taking TV distance as an example, we describe the equivalence under strong duality between the robust Bellman operator and its dual form as Lemma B.1.

**Lemma B.1.** *Consider any TV uncertainty set $\mathcal{U}^{\sigma^+}(P)$ and $\mathcal{U}^{\sigma^-}(P)$ associated with fixed uncertainty levels $\sigma^+, \sigma^- \in (0,1]$ and any probability vector $P \in \Delta(\mathcal{S})$. For any vector $V \in \mathbb{R}^S$ obeying $V \geq 0$, one has*

$$\inf_{P \in \mathcal{U}^{\sigma^+}(P)} PV = \max_{\alpha \in [\min_s V(s), \max_s V(s)]} \left\{ P[V]_\alpha - \sigma^+ \left( \alpha - \min_{s'} [V]_\alpha(s') \right) \right\}; \tag{27a}$$

$$\sup_{P \in \mathcal{U}^{\sigma^-}(P)} PV = \min_{\alpha \in [\min_s V(s), \max_s V(s)]} \left\{ P[V]_\alpha - \sigma^- \left( \alpha - \max_{s'} [V]_\alpha(s') \right) \right\}, \tag{27b}$$

*where $[V]_\alpha$ is defined in (17)*

Lemma B.1 can be proved similarly to Lemma 4.3 in [17]. Compared the standard Bellman operator, this lemma guarantees that no additional computing cost is required when applying the robust Bellman operator.

## B.2  Facts of RTZMGs and empirical RTZMGs

Recall the definition of any RTZMG $\mathcal{MG}_r = \left\{ \mathcal{S}, \mathcal{A}, \mathcal{B}, \mathcal{U}_\rho^{\sigma^+}(P^0), \mathcal{U}_\rho^{\sigma^-}(P^0), r, H \right\}$. According to robust Bellman equations in (7), one has: for any product policy $(\mu, \nu)$ and any $(h, s, a, b) \in [H] \times \mathcal{S} \times \mathcal{A} \times \mathcal{B}$,

$$Q_h^{\mu,\nu,\sigma^+}(s,a,b) = r_h(s,a,b) + \inf_{P \in \mathcal{U}_\rho^{\sigma^+}(P_{h,s,a,b}^0)} PV_{h+1}^{\mu,\nu,\sigma^+}; \tag{28a}$$

$$Q_h^{\mu,\nu,\sigma^-}(s,a,b) = r_h(s,a,b) + \sup_{P \in \mathcal{U}_\rho^{\sigma^-}(P_{h,s,a,b}^0)} PV_{h+1}^{\mu,\nu,\sigma^-}, \tag{28b}$$

where

$$V_h^{\mu,\nu,\sigma^+}(s) = \mathbb{E}_{a\sim\mu_h(s), b\sim\nu_h(s)} \left[ Q_h^{\mu,\nu,\sigma^+}(s,a,b) \right];$$

$$V_h^{\mu,\nu,\sigma^-}(s) = \mathbb{E}_{a\sim\mu_h(s), b\sim\nu_h(s)} \left[ Q_h^{\mu,\nu,\sigma^-}(s,a,b) \right].$$

Considering the offline setting, we use $\widehat{\mathcal{MG}}_r = \left\{ \mathcal{S}, \mathcal{A}, \mathcal{B}, \mathcal{U}_\rho^{\sigma^+}(\widehat{P}^0), \mathcal{U}_\rho^{\sigma^-}(\widehat{P}^0), r, H \right\}$ to represent the empirical RTZMG, which is established along with the estimated nominal distribution $\widehat{P}^0$

in (10). Therefore, for any product policy $(\mu, \nu)$, we define the empirical robust value function (resp. empirical robust Q-function) in $\widehat{\mathcal{MG}}_{\mathsf{r}}$ as $\widehat{V}_h^{\mu,\nu,\sigma^+}$ and $\widehat{V}_h^{\mu,\nu,\sigma^-}$ (resp. $\widehat{Q}_h^{\mu,\nu,\sigma^+}$ and $\widehat{Q}_h^{\mu,\nu,\sigma^-}$), which are analogous to (6). Moreover, we can similarly define the optimal empirical robust value function for both players over $\widehat{\mathcal{MG}}_{\mathsf{r}}$: $\forall s \in \mathcal{S}$,

$$\widehat{V}_h^{\star,\nu,\sigma^+}(s) = \widehat{V}_h^{\mu^\star,\nu,\sigma^+}(s) := \max_{\mu:\mathcal{S}\times[H]\to\Delta(\mathcal{A})} \widehat{V}_h^{\mu,\nu,\sigma^+}(s) = \max_{\mu:\mathcal{S}\times[H]\to\Delta(\mathcal{A})} \inf_{P\in\mathcal{U}^{\sigma^+}(\widehat{P}^0)} \widehat{V}_h^{\mu,\nu,P}(s); \tag{29a}$$

$$\widehat{V}_h^{\mu,\star,\sigma^-}(s) = \widehat{V}_h^{\mu,\nu^\star,\sigma^-}(s) := \max_{\nu:\mathcal{S}\times[H]\to\Delta(\mathcal{B})} \widehat{V}_h^{\mu,\nu,\sigma^-}(s) = \max_{\nu:\mathcal{S}\times[H]\to\Delta(\mathcal{B})} \inf_{P\in\mathcal{U}^{\sigma^-}(\widehat{P}^0)} \widehat{V}_h^{\mu,\nu,P}(s). \tag{29b}$$

Notably, for all $s \in \mathcal{S}$, there exists at least one *robust best-response* policy that can achieve $\widehat{V}_h^{\star,\nu,\sigma^+}(s)$ and $\widehat{V}_h^{\mu,\star,\sigma^-}(s)$, as proved in [5]. Therefore, we can obtain the empirical robust Bellman equation similar to (7) as: for any product policy $(\mu,\nu)$,

$$\widehat{Q}_h^{\mu,\nu,\sigma^+}(s,a,b) = r_h(s,a,b) + \inf_{P\in\mathcal{U}_\rho^{\sigma^+}(\widehat{P}_{h,s,a,b}^0)} P\widehat{V}_{h+1}^{\mu,\nu,\sigma^+}; \tag{30a}$$

$$\widehat{Q}_h^{\mu,\nu,\sigma^-}(s,a,b) = r_h(s,a,b) + \sup_{P\in\mathcal{U}_\rho^{\sigma^-}(\widehat{P}_{h,s,a,b}^0)} P\widehat{V}_{h+1}^{\mu,\nu,\sigma^-}, \tag{30b}$$

where

$$\widehat{V}_h^{\mu,\nu,\sigma^+}(s) = \mathbb{E}_{a\sim\mu_h(s),b\sim\nu_h(s)}[\widehat{Q}_h^{\mu,\nu,\sigma^+}(s,a,b)];$$

$$\widehat{V}_h^{\mu,\nu,\sigma^-}(s) = \mathbb{E}_{a\sim\mu_h(s),b\sim\nu_h(s)}[\widehat{Q}_h^{\mu,\nu,\sigma^-}(s,a,b)].$$

## C  Proof of Lemma 3.1

### C.1  Independence property

Let us examine two distinct data-generation mechanisms, where a sample transition quadruple $(s,a,b,h,s')$ represents a transition from state $s$ with actions $(a,b)$ to state $s'$ at step $h$.

**Step 1: Creating $\mathcal{D}^{\mathsf{t,a}}$ based on $\mathcal{D}^{\mathsf{t}}$.**   To construct the augmented dataset $\mathcal{D}^{\mathsf{t,a}}$, for each $(s,h) \in \mathcal{S} \times [H]$, we first include all $N_h^{\mathsf{t}}(s)$ sample transitions in $\mathcal{D}^{\mathsf{t}}$ originating from state $s$ at step $h$ in $\mathcal{D}^{\mathsf{t,a}}$. If $N_h^{\mathsf{t}}(s) > N_h^{\mathsf{m}}(s)$, we supplement $\mathcal{D}^{\mathsf{t,a}}$ with additional $[N_h^{\mathsf{t}}(s) - N_h^{\mathsf{m}}(s)]$ independent sample transitions $\{(s, a_{h,s}^{(i)}, b_{h,s}^{(i)}, h, s_{h,s}'^{(i)})\}$, as follows:

$$a_{h,s}^{(i)} \overset{\text{i.i.d.}}{\sim} \mu_h^{\mathsf{b}}(\cdot|s), \quad b_{h,s}^{(i)} \overset{\text{i.i.d.}}{\sim} \nu_h^{\mathsf{b}}(\cdot|s), \quad s_{h,s}'^{(i)} \overset{\text{i.i.d.}}{\sim} P_h(\cdot|s, a_{h,s}^{(i)}, b_{h,s}^{(i)}), \quad N_h^{\mathsf{m}}(s) < i \le N_h^{\mathsf{t}}(s).$$

**Step 2: Constructing $\mathcal{D}^{\text{iid}}$.**   For each $(s,h) \in \mathcal{S} \times [H]$, we generate $N_h^{\mathsf{t}}(s)$ independent sample transitions $\{(s, a_{h,s}^{(i)}, b_{h,s}^{(i)}, h, s_{h,s}'^{(i)})\}$, as follows:

$$a_{h,s}^{(i)} \overset{\text{i.i.d.}}{\sim} \mu_h^{\mathsf{b}}(\cdot|s), \quad b_{h,s}^{(i)} \overset{\text{i.i.d.}}{\sim} \nu_h^{\mathsf{b}}(\cdot|s), \quad s_{h,s}'^{(i)} \overset{\text{i.i.d.}}{\sim} P_h(\cdot|s,a,b), \quad 1 \le i \le N_h^{\mathsf{t}}(s).$$

The resulting dataset is defined as:

$$\mathcal{D}^{\text{iid}} := \left\{ (s, a_{h,s}^{(i)}, b_{h,s}^{(i)}, h, s_{h,s}'^{(i)}) \mid s \in \mathcal{S}, 1 \le h \le H, 1 \le i \le N_h^{\mathsf{t}}(s) \right\}.$$

**Establishing independence property.**   The dataset $\mathcal{D}^{\mathsf{t,a}}$ deviates from $\mathcal{D}^{\mathsf{t}}$ only if $N_h^{\mathsf{t}}(s) > N_h^{\mathsf{m}}(s)$. This augmentation ensures that $\mathcal{D}^{\mathsf{t,a}}$ contains precisely $N_h^{\mathsf{t}}(s)$ sample transitions from state $s$ at step $h$. Both $\mathcal{D}^{\mathsf{t,a}}$ and $\mathcal{D}^{\text{iid}}$ comprise exactly $N_h^{\mathsf{t}}(s)$ sample transitions from state $s$ at step $h$, with $\{N_h^{\mathsf{t}}(s)\}$ being statistically independent of random sample generation. Consequently, given $\{N_h^{\mathsf{t}}(s)\}$, the sample transitions in $\mathcal{D}^{\mathsf{t,a}}$ across different steps are statistically independent. As a result, both $\mathcal{D}^{\mathsf{t}}$ and $\mathcal{D}^{\text{iid}}$ can be regarded as collections of independent samples.

## C.2   Proof of $N_h^{\mathsf{t}}(s) \leq N_h^{\mathsf{m}}(s)$

Since $\mathcal{D}^{\mathsf{a}}$ is generated by half of the sample trajectories in line 2 in Algorithm 1, we have

$$N_h^{\mathsf{a}}(s) = \sum_{k=K/2+1}^{K} \mathbb{1}\{s_h^k = s\}, \forall s \in \mathcal{S}, 1 \leq h \leq H.$$

Thus, we can view $N_h^{\mathsf{a}}(s)$ as the sum of $K/2$ independent Bernoulli random variables with mean $d_h^{\mu^{\mathsf{n}},\nu^{\mathsf{n}}}(s)$. According to the Bernstein inequality and the union bound, we derive

$$\mathbb{P}\left\{\exists (s,h) \in \mathcal{S} \times [H] : \left| N_h^{\mathsf{a}}(s) - \frac{K}{2} d_h^{\mu^{\mathsf{n}},\nu^{\mathsf{n}}}(s) \right| \geq N_0 \right\}$$

$$\leq \sum_{s \in \mathcal{S}, h \in [H]} \mathbb{P}\left\{ \left| N_h^{\mathsf{a}}(s) - \frac{K}{2} d_h^{\mu^{\mathsf{n}},\nu^{\mathsf{n}}}(s) \right| \geq N_0 \right\}$$

$$\leq 2HS \exp\left( -\frac{N_0^2/2}{N_{h,s} + N_0/3} \right), \quad \forall N_0 \geq 0,$$

where

$$N_{h,s} := \frac{K}{2} \mathsf{Var}\big( \mathbb{1}\{s_h^t = s\} \big) = \frac{K d_h^{\mu^{\mathsf{n}},\nu^{\mathsf{n}}}(s)\big(1 - d_h^{\mu^{\mathsf{n}},\nu^{\mathsf{n}}}(s)\big)}{2} \leq \frac{K d_h^{\mu^{\mathsf{n}},\nu^{\mathsf{n}}}(s)}{2}.$$

Therefore, with probability of at least $1 - 2\delta$, we have that: $\forall s \in \mathcal{S}$ and $\forall 1 \leq h \leq H$,

$$\left| N_h^{\mathsf{a}}(s) - \frac{K}{2} d_h^{\mu^{\mathsf{n}},\nu^{\mathsf{n}}}(s) \right| \leq \sqrt{4 N_{h,s} \log \frac{HS}{\delta}} + \frac{2}{3} \log \frac{HS}{\delta} \leq \sqrt{2K d_h^{\mu^{\mathsf{n}},\nu^{\mathsf{n}}}(s) \log \frac{HS}{\delta}} + \log \frac{HS}{\delta}. \tag{31}$$

Since $\mathcal{D}^{\mathsf{m}}$ and $\mathcal{D}^{\mathsf{a}}$ are generated in the same way, we obtain that with probability exceeding $1 - 2\delta$,

$$\left| N_h^{\mathsf{m}}(s) - \frac{K}{2} d_h^{\mu^{\mathsf{n}},\nu^{\mathsf{n}}}(s) \right| \leq \sqrt{2K d_h^{\mu^{\mathsf{n}},\nu^{\mathsf{n}}}(s) \log \frac{HS}{\delta}} + \log \frac{HS}{\delta}, \forall s \in \mathcal{S}, 1 \leq h \leq H. \tag{32}$$

By combining (31) and (32), it follows that

$$|N_h^{\mathsf{m}}(s) - N_h^{\mathsf{a}}(s)| \leq 2\sqrt{2K d_h^{\mu^{\mathsf{n}},\nu^{\mathsf{n}}}(s) \log \frac{HS}{\delta}} + 2 \log \frac{HS}{\delta}, \forall s \in \mathcal{S}, 1 \leq h \leq H. \tag{33}$$

Now, we prove $N_h^{\mathsf{t}}(s) \leq N_h^{\mathsf{m}}(s)$ by considering two cases. *In the first case*, $N_h^{\mathsf{a}}(s) \leq 100 \log \frac{HS}{\delta}$. According to the definition in (13), we obtain

$$N_h^{\mathsf{t}}(s) = \max\left\{ N_h^{\mathsf{a}}(s) - 10\sqrt{N_h^{\mathsf{a}}(s) \log \frac{HS}{\delta}}, 0 \right\} = 0 \leq N_h^{\mathsf{m}}(s). \tag{34}$$

*In the second case*, $N_h^{\mathsf{a}}(s) > 100 \log \frac{HS}{\delta}$. From (31), it follows that

$$\frac{K}{2} d_h^{\mu^{\mathsf{n}},\nu^{\mathsf{n}}}(s) + \sqrt{2K d_h^{\mu^{\mathsf{n}},\nu^{\mathsf{n}}}(s) \log \frac{HS}{\delta}} + \log \frac{HS}{\delta} \geq N_h^{\mathsf{a}}(s),$$

which leads to

$$K d_h^{\mu^{\mathsf{n}},\nu^{\mathsf{n}}}(s) \geq (9\sqrt{2})^2 \log \frac{HS}{\delta} \geq 100 \log \frac{HS}{\delta}. \tag{35}$$

By substituting (35) into (31), we obtain

$$N_h^{\mathsf{a}}(s) \geq \frac{K}{2} d_h^{\mu^{\mathsf{n}},\nu^{\mathsf{n}}}(s) - \sqrt{2K d_h^{\mu^{\mathsf{n}},\nu^{\mathsf{n}}}(s) \log \frac{HS}{\delta}} - \log \frac{HS}{\delta} \geq \frac{K}{4} d_h^{\mu^{\mathsf{n}},\nu^{\mathsf{n}}}(s). \tag{36}$$

Consequently, in the case of $N_h^a(s) > 100 \log \frac{HS}{\delta}$, we have

$$N_h^t(s) = \max \left\{ N_h^a(s) - 10\sqrt{N_h^a(s) \log \frac{HS}{\delta}}, \ 0 \right\}$$

$$= N_h^a(s) - 10\sqrt{N_h^a(s) \log \frac{HS}{\delta}}$$

$$\overset{(i)}{\leq} N_h^a(s) - 5\sqrt{K d_h^{\mu^n, \nu^n}(s) \log \frac{HS}{\delta}}$$

$$\overset{(ii)}{\leq} N_h^a(s) - \left\{ 2\sqrt{2K d_h^{\mu^n, \nu^n}(s) \log \frac{HS}{\delta}} + 2\log \frac{HS}{\delta} \right\} \overset{(iii)}{\leq} N_h^m(s), \tag{37}$$

where (i) holds under condition (36), (ii) holds under the condition (35), and (iii) comes from the inequality (33) with probability of at least $1 - 2\delta$.

Combining (34) and (37), we establish $N_h^t(s) \leq N_h^m(s)$.

As proved in Appendix C.3, we have: $\forall (s, a, b, h) \in \mathcal{S} \times \mathcal{A} \times \mathcal{B} \times [H]$, with probability exceeding $1 - 2\delta$,

$$N_h^t(s, a, b) \geq N_h^t(s)\mu_h^n(a \mid s)\nu_h^n(b \mid s) - \sqrt{4N_h^t(s)\mu_h^n(a \mid s)\nu_h^n(b \mid s) \log \frac{KH}{\delta}} - \log \frac{KH}{\delta}. \tag{38}$$

Employing the fact $N_h^t(s) \leq N_h^m(s)$ and (38), we can prove (14) in two cases, i.e., $K d_h^{\mu^n, \nu^n}(s, a, b) \leq 1600 \log \frac{KH}{\delta}$ and $K d_h^{\mu^n, \nu^n}(s, a, b) > 1600 \log \frac{KH}{\delta}$.

In the case of $K d_h^{\mu^n, \nu^n}(s, a) \leq 1600 \log \frac{KH}{\delta}$, we can readily obtain

$$\frac{K}{8} d_h^{\mu^n, \nu^n}(s, a) - 5\sqrt{K d_h^{\mu^n, \nu^n}(s, a) \log \frac{KH}{\delta}} \leq 0 \leq N_h^t(s, a). \tag{39}$$

In the case of $K d_h^{\mu^n, \nu^n}(s, a, b) = K d_h^{\mu^n, \nu^n}(s)\mu_h^n(a \mid s)\nu_h^n(b \mid s) > 1600 \log \frac{KH}{\delta}$, we obtain

$$N_h^a(s) \geq \frac{K}{4} d_h^{\mu^n, \nu^n}(s) \geq 400 \log \frac{KH}{\delta}, \tag{40}$$

which can be derived following the same line of (36). Then, (40) and the definition of $N_h^t(s)$ together yield

$$N_h^t(s) \geq N_h^a(s) - 10\sqrt{N_h^a(s) \log \frac{KH}{\delta}}$$

$$\geq \frac{K}{4} d_h^{\mu^n, \nu^n}(s) - 10\sqrt{\frac{K}{4} d_h^{\mu^n, \nu^n}(s) \log \frac{KH}{\delta}} \geq \frac{K}{8} d_h^{\mu^n, \nu^n}(s).$$

As a consequent,

$$N_h^t(s)\mu_h^n(a \mid s)\nu_h^n(b \mid s) \geq \frac{K}{8} d_h^{\mu^n, \nu^n}(s)\mu_h^n(a \mid s)\nu_h^n(b \mid s) \tag{41}$$

$$= \frac{K}{8} d_h^{\mu^n, \nu^n}(s, a, b) \geq 200 \log \frac{KH}{\delta}, \tag{42}$$

where the last inequality holds under the assumption of the second case. Combining (41) with (38) yields

$$N_h^t(s, a, b) \geq \frac{K}{8} d_h^{\mu^n, \nu^n}(s, a, b) - \sqrt{\frac{K}{2} d_h^{\mu^n, \nu^n}(s, a, b) \log \frac{KH}{\delta}} - \log \frac{KH}{\delta}$$

$$\geq \frac{K}{8} d_h^{\mu^n, \nu^n}(s, a, b) - 2\sqrt{K d_h^{\mu^n, \nu^n}(s, a, b) \log \frac{KH}{\delta}}.$$

Combining the result above with (39) and referencing (38), we conclude the proof of Lemma 3.1.

## C.3 Proof of (38).

To prove (38), we analyze two cases, i.e., $N_h^{\mathsf{t}}(s)\mu_h^{\mathsf{n}}(a\,|\,s)\nu_h^{\mathsf{n}}(b\,|\,s) \leq 4\log\frac{KH}{\delta}$ and $N_h^{\mathsf{t}}(s)\mu_h^{\mathsf{n}}(a\,|\,s)\nu_h^{\mathsf{n}}(b\,|\,s) > 4\log\frac{KH}{\delta}$.

In the first case of $N_h^{\mathsf{t}}(s)\mu_h^{\mathsf{n}}(a\,|\,s)\nu_h^{\mathsf{n}}(b\,|\,s) \leq 4\log\frac{KH}{\delta}$, we conclude the right-hand side of (38) is negative, leading to (38). In the second case of $N_h^{\mathsf{t}}(s)\mu_h^{\mathsf{n}}(a\,|\,s)\nu_h^{\mathsf{n}}(b\,|\,s) > 4\log\frac{KH}{\delta}$, we compose a special set $\mathcal{D}^{\mathsf{l}}$ as

$$\mathcal{D}^{\mathsf{l}} := \left\{ (s,a,b,h) \in \mathcal{S} \times \mathcal{A} \times \mathcal{B} \times [H] \,\Big|\, N_h^{\mathsf{t}}(s)\mu_h^{\mathsf{n}}(a\,|\,s)\nu_h^{\mathsf{n}}(b\,|\,s) > 4\log\frac{KH}{\delta} \right\}. \quad (43)$$

Given the fact that

$$\sum_{(s,a,b,h)\in\mathcal{S}\times\mathcal{A}\times\mathcal{B}\times[H]} N_h^{\mathsf{t}}(s)\mu_h^{\mathsf{n}}(a\,|\,s)\nu_h^{\mathsf{n}}(b\,|\,s) = \sum_{(s,h)\in\mathcal{S}\times[H]} N_h^{\mathsf{t}}(s) \sum_{(a,b)\in\mathcal{A}\times\mathcal{B}} \mu_h^{\mathsf{n}}(a\,|\,s)\nu_h^{\mathsf{n}}(b\,|\,s)$$

$$= \sum_{(s,h)\in\mathcal{S}\times[H]} N_h^{\mathsf{t}}(s) \leq \sum_{(s,h)\in\mathcal{S}\times[H]} N_h^{\mathsf{a}}(s) = \frac{KH}{2},$$

the cardinality of $\mathcal{D}^{\mathsf{l}}$ can be bounded as:

$$\left| \mathcal{D}^{\mathsf{l}} \right| < \frac{\sum_{(s,a,b,h)} N_h^{\mathsf{t}}(s)\mu_h^{\mathsf{n}}(a\,|\,s)\nu_h^{\mathsf{n}}(b\,|\,s)}{4\log\frac{KH}{\delta}} \leq KH/2. \quad (44)$$

Moreover, we can view $N_h^{\mathsf{t}}(s,a)$ as the sum of $N_h^{\mathsf{t}}(s)$ independent Bernoulli random variables with mean $\mu_h^{\mathsf{n}}(a\,|\,s)\nu_h^{\mathsf{n}}(b\,|\,s)$, due to $N_h^{\mathsf{t}}(s) \leq N_h^{\mathsf{m}}(s)$ with high probability and conditioned on $N_h^{\mathsf{t}}(s)$ and $N_h^{\mathsf{m}}(s)$. Analogous to (31) based on the condition $N_h^{\mathsf{t}}(s) \leq N_h^{\mathsf{m}}(s)$, we can repeat the Bernstein-type argument and obtain that for any fixed triple $(s,a,b,h)$, with probability of at least $1 - 2\delta/(KH)$,

$$N_h^{\mathsf{t}}(s,a) \geq N_h^{\mathsf{t}}(s)\mu_h^{\mathsf{n}}(a\,|\,s)\nu_h^{\mathsf{n}}(b\,|\,s) - \sqrt{4N_h^{\mathsf{t}}(s)\mu_h^{\mathsf{n}}(a\,|\,s)\nu_h^{\mathsf{n}}(b\,|\,s)\log\frac{KH}{\delta}} - \log\frac{KH}{\delta}. \quad (45)$$

With probability exceeding $1-\delta$, (45) holds $\forall(s,a,b,h) \in \mathcal{D}^{\mathsf{l}}$ by utilizing the union bound of (44) over all $(s,a,b,h) \in \mathcal{D}^{\mathsf{l}}$. Combining the two cases, we conclude that (38) holds $\forall(s,a,b,h) \in \mathcal{S} \times \mathcal{A} \times \mathcal{B} \times [H]$ with probability of at least $1-\delta$.

# D  Proof of Theorem 4.2

Theorem 4.2 can be proved in three steps.

## D.1  Step 1: Decoupling statistical dependency

Before bounding $\mathrm{Gap}(\widehat{\mu},\widehat{\nu})$, we introduce an important lemma, quantifying the difference between $\widehat{P}$ and $P$ when projected in the direction of the value function.

**Lemma D.1.** *Instate the assumptions in Theorem 4.2. Consider any vector $V \in \mathbb{R}^S$ with $\|V\|_\infty \leq H$ for all $(h,s,a,b) \in [H] \times \mathcal{S} \times \mathcal{A} \times \mathcal{B}$ satisfying $N_h(s,a,b) > 0$. With probability of at least $1-\delta$, one has*

$$\left| \inf_{P\in\mathcal{U}^{\sigma^+}(\widehat{P}_{h,s,a,b}^0)} PV - \inf_{P\in\mathcal{U}^{\sigma^+}(P_{h,s,a,b}^0)} PV \right| \leq C_4\sqrt{\frac{1}{N_h(s,a,b)}\mathsf{Var}_{\widehat{P}_{h,s,a,b}^0}(V)\log\frac{KH}{\delta}} + C_4\frac{H\log\frac{KH}{\delta}}{N_h(s,a,b)} \quad (46)$$

*for some sufficiently large constant $C_4 > 0$, and*

$$\mathsf{Var}_{\widehat{P}_{h,s,a,b}^0}(V) \leq 2\mathsf{Var}_{P_{h,s,a,b}^0}(V) + O\left(\frac{H^2}{N_h(s,a,b)}\log\frac{KH}{\delta}\right). \quad (47)$$

Proof can be found in Appendix G.1.

In simple terms, (46) provides a Bernstein-type concentration bound, while (47) ensures that the empirical variance estimate (i.e., the plug-in estimate) closely matches the true variance. Notably, Lemma D.1 does not require $V$ to be statistically independent of $\widehat{P}^0_{h,s,a,b}$, which is essential given the complex statistical dependencies in RTZ-VI-LCB. Under the leave-one-out analysis (see, e.g., [1, 7, 24, 25]), we prove Lemma D.1 to decouple statistical dependencies, as illustrated in Appendix G.1.

With Lemma D.1, we now have: for any $(h, s, a, b) \in [H] \times \mathcal{S} \times \mathcal{A} \times \mathcal{B}$ satisfying $N_h(s, a, b) \geq 1$,

$$\left| \inf_{\mathcal{P} \in \mathcal{U}^{\sigma^+}(\widehat{P}^0_{h,s,a,b})} PV - \inf_{\mathcal{P} \in \mathcal{U}^{\sigma^+}(P^0_{h,s,a,b})} PV \right| \leq \beta_h(s, a, b, V). \tag{48}$$

Therefore, we conclude that $\widehat{Q}^+_h(s, a, b)$ is an optimistic estimation of $\widehat{Q}^{\mu,\nu,\sigma^+}_h(s, a, b)$, as summarized below.

**Lemma D.2.** *With probability exceeding $1 - \delta$, it holds that*

$$\widehat{Q}^+_h(s, a, b) \geq Q^{\star,\widehat{\nu},\sigma^+}_h(s, a, b) \qquad and \qquad \widehat{V}^+_h(s) \geq V^{\star,\widehat{\nu},\sigma^+}_h(s); \tag{49}$$

See Appendix G.2 for detailed proofs.

Moreover, we introduce another key lemma highlighting the difference between RTZMGs and standard TZMGs from the same idea of Lemma 3 in [34]. The range of the robust value function narrows as the uncertainty level $\sigma^+$ of its uncertainty set increases, as shown below.

**Lemma D.3.** *Consider the uncertainty set $\mathcal{U}^{\sigma^+}(\cdot)$ with TV distance and any RTZMG $\mathcal{MG}_r = \left\{ \mathcal{S}, \mathcal{A}, \mathcal{B}, \mathcal{U}^{\sigma^+}(P), \mathcal{U}^{\sigma^-}(P), r, H \right\}$. The optimistic robust value function estimate $\widehat{V}^+_h$:*

$$\forall h \in [H]: \quad \max_{s \in \mathcal{S}} \widehat{V}^+_h - \min_{s \in \mathcal{S}} \widehat{V}^+_h \leq \min \left\{ \frac{(H+1)\left(1 - (1 - \sigma^+)^{H-h}\right)}{\sigma^+}, H \right\}.$$

See Appendix G.3 for detail proofs.

## D.2   Step 2: Decomposing the error $\mathrm{Gap}(\widehat{\mu}, \widehat{\nu})$

The goal of RTZ-VI-LCB is to output an $\varepsilon$-robust NE policy $(\widehat{\mu}, \widehat{\nu})$ satisfying $\mathrm{Gap}(\widehat{\mu}, \widehat{\nu})$ in (9), i.e.,

$$\mathrm{Gap}(\widehat{\mu}, \widehat{\nu}) := \max \left\{ V^{\star,\widehat{\nu},\sigma^+}_1(\varrho) - V^{\star,\sigma^+}_1(\varrho), \ V^{\star,\sigma^-}_1(\varrho) - V^{\widehat{\mu},\star,\sigma^-}_1(\varrho) \right\} \leq \varepsilon.$$

Due to the interchangeability between the max-player and the min-player, we assume without loss of generality that $V^{\star,\widehat{\nu},\sigma^+}_1(\varrho) - V^{\star,\sigma^+}_1(\varrho)$ is larger than $V^{\star,\sigma^-}_1(\varrho) - V^{\widehat{\mu},\star,\sigma^-}_1(\varrho)$, leading to $\mathrm{Gap}(\widehat{\mu}, \widehat{\nu}) \leq \left\{ V^{\star,\widehat{\nu},\sigma^+}_1(\varrho) - V^{\star,\sigma^+}_1(\varrho) \right\}$.

According to the relationship in Lemma D.2, we obtain

$$V^{\star,\widehat{\nu},\sigma^+}_h(s) \leq \widehat{V}^+_h(s) = \max_{\mu \in \Delta(\mathcal{A})} \min_{\nu \in \Delta(\mathcal{B})} \mathbb{E}_{(a,b) \sim (\mu(s), \nu(s))} \left[ \widehat{Q}^+_h(s, a, b) \right]$$

$$\leq \max_{\mu \in \Delta(\mathcal{A})} \mathbb{E}_{(a,b) \sim (\mu(s), \nu^\star(s))} \left[ Q^+_h(s, a, b) \right], \tag{50}$$

where the first equality comes from line 5 in Algorithm 2. Therefore, there exists a deterministic policy $\mu^{\mathsf{d}} : \mathcal{S} \leftarrow \Delta(\mathcal{A})$ satisfying that for any $s \in \mathcal{S}$

$$\mu^{\mathsf{d}}(s) := \arg \max_{\mu \in \Delta(\mathcal{A})} \mathbb{E}_{(a,b) \sim (\mu(s), \nu^\star(s))} \left[ Q^+_h(s, a, b) \right]. \tag{51}$$

We start by defining the following notation:

- The state-action space covered by the behavior policy $(\mu^{\mathsf{n}}, \nu^{\mathsf{n}})$ in the nominal transition kernel $P^0$ is denoted as

$$\mathcal{C}^{\mathsf{n}} = \left\{ (h, s, a, b) : d^{\mathsf{n}}_h(s, a, b) > 0 \right\}. \tag{52}$$

- For any time step $h \in [H]$, the sets of potential state occupancy distributions w.r.t. the policy $(\mu^{\mathsf{d}}(s), \nu^{\star}(s))$ and the uncertainty set $P \in \mathcal{U}^{\sigma^+}(P^0)$ are given by

$$\mathcal{D}_h^{\mathsf{p}} := \left\{ \left[ d_h^{\mu^{\mathsf{d}}(s), \nu^{\star}(s), P}(s) \right]_{s \in \mathcal{S}} : P \in \mathcal{U}^{\sigma^+}(P^0) \right\}; \tag{53}$$

$$\mathcal{D}_h^{\mathsf{pa}} := \left\{ \left[ d_h^{\mu^{\mathsf{d}}(s), \nu^{\star}(s), P}(s, a, b) \right]_{(s,a,b) \in \mathcal{S} \times \mathcal{A} \times \mathcal{B}} : P \in \mathcal{U}^{\sigma^+}(P^0) \right\}. \tag{54}$$

- For convenience and without ambiguity, we introduce additional notation for $h \in [H]$ as

$$\beta_h^{\mu^{\mathsf{d}}, \nu^{\star}}(s) = \mathbb{E}_{(a,b) \sim (\mu^{\mathsf{d}}(s), \nu^{\star}(s))} \beta_h \left( s, a, b, \widehat{V}_{h+1}^+ \right).$$

In particular, the vector $\beta_h^{\mu^{\mathsf{d}}, \nu^{\star}} \in \mathbb{R}^S$ is defined with its $s$-th term given by $\beta_h^{\mu^{\mathsf{d}}, \nu^{\star}}(s)$.

- Similarly, we can define the notation related to rewards for $h \in [H]$ as

$$\widehat{r}_h^{\mu^{\mathsf{d}}, \nu^{\star}}(s) = \mathbb{E}_{(a,b) \sim (\mu^{\mathsf{d}}(s), \nu^{\star}(s))} \widehat{r}_h(s, a, b).$$

According to the update rule in line 4 in Algorithm 2 and robust Bellman equality (28), we derive

$$V_h^{\star, \widehat{\nu}, \sigma^+}(s) - V_h^{\star, \sigma^+}(s) \tag{55}$$

$$\leq \widehat{V}_h^+(s) - V_h^{\mu^{\mathsf{d}}, \nu^{\star}, \sigma^+}(s)$$

$$\leq \mathbb{E}_{(a,b) \sim (\mu^{\mathsf{d}}(s), \nu^{\star}(s))} \inf_{P \in \mathcal{U}^{\sigma^+}\left(\widehat{P}_{h,s,a,b}^0\right)} P\widehat{V}_{h+1}^+ + \beta_h^{\mu^{\mathsf{d}}, \nu^{\star}}(s)$$

$$- \mathbb{E}_{(a,b) \sim (\mu^{\mathsf{d}}(s), \nu^{\star}(s))} \inf_{P \in \mathcal{U}^{\sigma^+}\left(P_{h,s,a,b}^0\right)} PV_{h+1}^{\mu^{\mathsf{d}}, \nu^{\star}, \sigma^+}$$

$$\leq \mathbb{E}_{(a,b) \sim (\mu^{\mathsf{d}}(s), \nu^{\star}(s))} \left[ \inf_{P \in \mathcal{U}^{\sigma^+}\left(P_{h,s,a,b}^0\right)} P\widehat{V}_{h+1}^+ - \inf_{P \in \mathcal{U}^{\sigma^+}\left(P_{h,s,a,b}^0\right)} PV_{h+1}^{\mu^{\mathsf{d}}, \nu^{\star}, \sigma^+} \right.$$

$$\left. + \left| \inf_{P \in \mathcal{U}^{\sigma^+}\left(P_{h,s,a,b}^0\right)} P\widehat{V}_{h+1}^+ - \inf_{P \in \mathcal{U}^{\sigma^+}\left(\widehat{P}_{h,s,a,b}^0\right)} P\widehat{V}_{h+1}^+ \right| \right] + \beta_h^{\mu^{\mathsf{d}}, \nu^{\star}}(s)$$

$$\overset{\text{(i)}}{\leq} \mathbb{E}_{(a,b) \sim (\mu^{\mathsf{d}}(s), \nu^{\star}(s))} \left[ \inf_{P \in \mathcal{U}^{\sigma^+}\left(P_{h,s,a,b}^0\right)} P\widehat{V}_{h+1}^+ - \inf_{P \in \mathcal{U}^{\sigma^+}\left(P_{h,s,a,b}^0\right)} PV_{h+1}^{\mu^{\mathsf{d}}, \nu^{\star}, \sigma^+} \right] + 2\beta_h^{\mu^{\mathsf{d}}, \nu^{\star}}(s)$$

$$\overset{\text{(ii)}}{\leq} \mathbb{E}_{(a,b) \sim (\mu^{\mathsf{d}}(s), \nu^{\star}(s))} \left[ P_{h,s,a,b}^{\mathrm{inf}, V} \left( \widehat{V}_{h+1}^+ - V_{h+1}^{\mu^{\mathsf{d}}, \nu^{\star}, \sigma^+} \right) \right] + 2\beta_h^{\mu^{\mathsf{d}}, \nu^{\star}}(s). \tag{56}$$

Here, (i) holds due to (48) in Lemma D.1 for $N_h(s, a, b) > 0$ and

$$\left| \inf_{P \in \mathcal{U}^{\sigma^+}\left(P_{h,s,a,b}^0\right)} P\widehat{V}_{h+1}^+ - \inf_{P \in \mathcal{U}^{\sigma^+}\left(\widehat{P}_{h,s,a,b}^0\right)} P\widehat{V}_{h+1}^+ \right| \leq H = \beta_h^{\mu^{\mathsf{d}}, \nu^{\star}}(s) \text{ for } N_h(s, a, b) = 0. \tag{57}$$

Moreover, (ii) is due to

$$P_{h,s,a,b}^{\mathrm{inf}, V} := \operatorname{argmin}_{P \in \mathcal{U}^{\sigma^+}\left(P_{h,s,a,b}^0\right)} PV_{h+1}^{\mu^{\mathsf{d}}, \nu^{\star}, \sigma^+} \tag{58}$$

and consequently,

$$\inf_{P \in \mathcal{U}^{\sigma^+}\left(P_{h,s,a,b}^0\right)} PV_{h+1}^{\mu^{\mathsf{d}}, \nu^{\star}, \sigma^+} = P_{h,s,a,b}^{\mathrm{inf}, V} V_{h+1}^{\mu^{\mathsf{d}}, \nu^{\star}, \sigma^+}, \text{ and } \inf_{P \in \mathcal{U}^{\sigma^+}\left(P_{h,s,a,b}^0\right)} P\widehat{V}_{h+1}^+ \leq P_{h,s,a,b}^{\mathrm{inf}, V} \widehat{V}_{h+1}^+.$$

For ease of exposure, we introduce notation as $\widetilde{P}_{h,s}^{\mathrm{inf}, V} := \mathbb{E}_{(a,b) \sim (\mu^{\mathsf{d}}(s), \nu^{\star}(s))} P_{h,s,a,b}^{\mathrm{inf}, V}$. Furthermore, we define a sequence of matrices $\widetilde{P}_h^{\mathrm{inf}, V} \in \mathbb{R}^{S \times S}$. We can utilize (56) recursively over the time steps

$h, h+1, \cdots, H$ and derive

$$V_h^{\star,\widehat{\nu},\sigma^+}(s) - V_h^{\star,\sigma^+}(s) \leq \widehat{V}_h^+(s) - V_h^{\mu^{\mathsf{d}},\nu^\star,\sigma^+}(s)$$

$$\leq \widetilde{P}_h^{\mathrm{inf},V}\left(\widehat{V}_{h+1}^+ - V_{h+1}^{\mu^{\mathsf{d}},\nu^\star,\sigma^+}\right) + 2\beta_h^{\mu^{\mathsf{d}},\nu^\star}(s)$$

$$\leq \widetilde{P}_h^{\mathrm{inf},V}\widetilde{P}_{h+1}^{\mathrm{inf},V}\left(\widehat{V}_{h+2}^+ - V_{h+2}^{\mu^{\mathsf{d}},\nu^\star,\sigma^+}\right) + 2\widetilde{P}_h^{\mathrm{inf},V}\beta_{h+1}^{\mu^{\mathsf{d}},\nu^\star} + 2\beta_h^{\mu^{\mathsf{d}},\nu^\star}(s)$$

$$\leq \cdots \leq 2\sum_{i=h+1}^{H}\left(\prod_{j=h}^{i-1}\widetilde{P}_j^{\mathrm{inf},V}\right)\beta_i^{\mu^{\mathsf{d}},\nu^\star} + 2\beta_h^{\mu^{\mathsf{d}},\nu^\star}(s) \tag{59}$$

$$= 2\sum_{i=h}^{H}\left(\prod_{j=h}^{i-1}\widetilde{P}_j^{\mathrm{inf},V}\right)\beta_i^{\mu^{\mathsf{d}},\nu^\star}, \tag{60}$$

For conciseness, we define $\left(\prod_{j=h}^{h-1}\widetilde{P}_j^{\mathrm{inf},V}\right) = I$ to simplify notation.

For any $d_h^{\mu^{\mathsf{d}},\nu^\star} \in \mathcal{D}_h^{\mathsf{p}}$ (cf. (53)), taking inner product with (59) yields

$$\left\langle d_h^{\mu^{\mathsf{d}},\nu^\star}, V_h^{\star,\widehat{\nu},\sigma^+} - V_h^{\star,\sigma^+}\right\rangle \leq \left\langle d_h^{\mu^{\mathsf{d}},\nu^\star}, 2\sum_{i=h}^{H}\left(\prod_{j=h}^{i-1}\widetilde{P}_j^{\mathrm{inf},V}\right)\beta_i^{\mu^{\mathsf{d}},\nu^\star}\right\rangle = 2\sum_{i=h}^{H}\left\langle d_i^{\mathsf{p},\mu^{\mathsf{d}},\nu^\star}, \beta_i^{\mu^{\mathsf{d}},\nu^\star}\right\rangle,$$
$$\tag{61}$$

where

$$d_i^{\mathsf{p},\mu^{\mathsf{d}},\nu^\star} := \left[\left(d_h^{\mu^{\mathsf{d}},\nu^\star}\right)^\top\left(\prod_{j=h}^{i-1}\widetilde{P}_j^{\mathrm{inf},V}\right)\right]^\top \in \mathcal{D}_i^{\mathsf{p}} \tag{62}$$

by the definition of $\mathcal{D}_i^{\mathsf{p}}$ (cf. (53)) for all $i = h+1, \cdots, H$.

Next, we control $\langle d_i^{\mathsf{p},\mu^{\mathsf{d}},\nu^\star}, \beta_i^{\mu^{\mathsf{d}},\nu^\star}\rangle$ by using concentrability. According to (18) in Lemma D.1, we first demonstrate that the pessimistic penalty satisfies

$$\beta_i(s,a,b,\hat{V}) \leq \max\left\{\sqrt{\frac{C_{\mathsf{n}}\log\frac{KH}{\delta}}{N_i(s,a,b)}\mathsf{Var}_{\widehat{P}_{i,s,a,b}^0}(\widehat{V})}, \frac{2C_{\mathsf{n}}H\log\frac{KH}{\delta}}{N_i(s,a,b)}\right\}$$

$$\leq \sqrt{\frac{C_{\mathsf{n}}\log\frac{KH}{\delta}}{N_i(s,a,b)}\mathsf{Var}_{\widehat{P}_{i,s,a,b}^0}(\widehat{V})} + \frac{2C_{\mathsf{n}}H\log\frac{KH}{\delta}}{N_i(s,a,b)}$$

$$\overset{(i)}{\leq} \sqrt{\frac{C_{\mathsf{n}}\log\frac{KH}{\delta}}{N_i(s,a,b)}\left(2\mathsf{Var}_{P_{i,s,a,b}^0}(\widehat{V}) + \frac{C_0H^2}{N_i(s,a,b)}\log\frac{KH}{\delta}\right)} + \frac{2C_{\mathsf{n}}H\log\frac{KH}{\delta}}{N_i(s,a,b)}$$

$$\overset{(ii)}{\leq} \sqrt{\frac{2C_{\mathsf{n}}\log\frac{KH}{\delta}}{N_i(s,a,b)}\mathsf{Var}_{P_{i,s,a,b}^0}(\widehat{V})} + \frac{\left(2C_{\mathsf{n}} + \sqrt{C_{\mathsf{n}}C_0}\right)H\log\frac{KH}{\delta}}{N_i(s,a,b)} \tag{63}$$

where (i) holds by applying (47) for some sufficiently large $C_0$ and (ii) follows from the Cauchy-Schwarz inequality. Therefore, combining the definition of $\beta_i^{\mu^{\mathsf{d}},\nu^\star}(s)$, we obtain

$$\langle d_i^{\mathsf{p},\mu^{\mathsf{d}},\nu^\star}, \beta_i^{\mu^{\mathsf{d}},\nu^\star}\rangle = \sum_{s\in\mathcal{S}} d_i^{\mathsf{p},\mu^{\mathsf{d}},\nu^\star}(s)\beta_i^{\mu^{\mathsf{d}},\nu^\star}(s)$$

$$= \sum_{s\in\mathcal{S}} d_i^{\mathsf{p},\mu^{\mathsf{d}},\nu^\star}(s)\mathbb{E}_{(a,b)\sim(\mu^{\mathsf{d}}(s),\nu^\star(s))}\beta_i(s,a,b,\hat{V})$$

$$= \sum_{(s,a,b)\in\mathcal{S}\times\mathcal{A}\times\mathcal{B}} d_i^{\mathsf{p},\mu^{\mathsf{d}},\nu^\star}(s)\mathbb{1}\{a=\mu^{\mathsf{d}}(s)\}\nu^\star(b|s)\beta_i(s,a,b,\hat{V})$$

$$= \sum_{(s,b)\in\mathcal{S}\times\mathcal{B}} d_i^{\mathsf{p},\mu^{\mathsf{d}},\nu^\star}(s,\mu^{\mathsf{d}}(s),b)\beta_i(s,\mu^{\mathsf{d}}(s),b,\hat{V}), \tag{64}$$

where the last equation holds due to the definition in (5). Then, we observe $d_h^{\mathsf{p},\mu^{\mathsf{d}},\nu^\star}(s,a,b) \in \mathcal{D}_h^{\mathsf{pa}}$; cf. (54). Thereafter, we divide the bound (64) into two cases. *In the first case*, i.e., $s \in S$ where $\max_{P \in \mathcal{U}^{\sigma+}(P^0)} d_i^{\mu^{\mathsf{d}},\nu^\star,P}(s,\mu^{\mathsf{d}}(s),b) = 0$, it follows from the definition (53) that: For any $d_i^{\mathsf{p},\mu^{\mathsf{d}},\nu^\star}(s,\mu^{\mathsf{d}}(s),b) \in \mathcal{D}_i^{\mathsf{pa}}$,

$$d_i^{\mathsf{p},\mu^{\mathsf{d}},\nu^\star}(s,\mu^{\mathsf{d}}(s),b) = 0. \tag{65}$$

*In the second case*, i.e., $s \in S$ where $\max_{P \in \mathcal{U}^{\sigma+}(P^0)} d_i^{\mu^{\mathsf{d}},\nu^\star,P}(s,\mu^{\mathsf{d}}(s),b) > 0$, under the assumption in (20),

$$\max_{P \in \mathcal{U}^{\sigma+}(P^0)} \frac{\min\left\{ d_i^{\mu^{\mathsf{d}},\nu^\star,P}(s,\mu^{\mathsf{d}}(s),b), \frac{1}{S(A+B)} \right\}}{d_i^{\mathsf{n}}(s,\mu^{\mathsf{d}}(s),b)} \le C_{\mathsf{r}}^\star < \infty,$$

which implies that

$$d_i^{\mathsf{n}}(s,\mu^{\mathsf{d}}(s),b) > 0 \quad \text{and} \quad (i,s,\mu^{\mathsf{d}}(s),b) \in \mathcal{C}^{\mathsf{n}}. \tag{66}$$

Lemma 3.1 tells that with probability of at least $1 - 8\delta$,

$$
\begin{aligned}
N_i(s,\mu^{\mathsf{d}}(s),b) &\ge \frac{K d_i^{\mathsf{n}}(s,\mu^{\mathsf{d}}(s),b)}{8} - 5\sqrt{K d_i^{\mathsf{n}}(s,\mu^{\mathsf{d}}(s),b)\log\frac{KH}{\delta}} \\
&\overset{(\mathrm{i})}{\ge} \frac{K d_i^{\mathsf{n}}(s,\mu^{\mathsf{d}}(s),b)}{16} \\
&\overset{(\mathrm{ii})}{\ge} \frac{K \max_{P \in \mathcal{U}^{\sigma}(P^0)} \min\left\{ d_i^{\mu^{\mathsf{d}},\nu^\star,P}(s,\mu^{\mathsf{d}}(s),b), \frac{1}{S(A+B)} \right\}}{16 C_{\mathsf{r}}^\star} \\
&\ge \frac{K \min\left\{ d_i^{\mathsf{p},\mu^{\mathsf{d}},\nu^\star}(s,\mu^{\mathsf{d}}(s),b), \frac{1}{S(A+B)} \right\}}{16 C_{\mathsf{r}}^\star},
\end{aligned}
\tag{67}
$$

Here, (i) holds since, with $f(\sigma^+,\sigma^-,H) = \min\left\{ \frac{H\sigma^+ + 1 - (1-\sigma^+)^H}{(\sigma^+)^2}, \frac{H\sigma^- + 1 - (1-\sigma^-)^H}{(\sigma^-)^2}, H \right\}$,

$$
\begin{aligned}
K d_i^{\mathsf{n}}(s,\mu^{\mathsf{d}}(s),b) &\ge c_0 \frac{HS(A+B)}{d_{\mathsf{m}}^{\mathsf{n}}} \log\frac{KH}{\delta} f(\sigma^+,\sigma^-,H) d_i^{\mathsf{n}}(s,\mu^{\mathsf{d}}(s),b) \\
&\ge c_0 HS(A+B) \log\frac{KH}{\delta} f(\sigma^+,\sigma^-,H) \ge 1600 \log\frac{KH}{\delta}, \tag{68}
\end{aligned}
$$

where the first inequality follows from condition (22), and the second inequality follows from

$$d_{\mathsf{m}}^{\mathsf{n}} = \min_{h,s,\mu^{\mathsf{d}}(s),b}\left\{ d_h^{\mathsf{n}}(s,\mu^{\mathsf{d}}(s),b) : d_h^{\mathsf{n}}(s,\mu^{\mathsf{d}}(s),b) > 0 \right\} \le d_i^{\mathsf{n}}(s,\mu^{\mathsf{d}}(s),b). \tag{69}$$

Moreover, (ii) comes from Assumption 4.1.

Combining (63) and (64), we arrive at

$$
\begin{aligned}
&\langle d_i^{\mathsf{p},\mu^{\mathsf{d}},\nu^\star}, \beta_i^{\mu^{\mathsf{d}},\nu^\star}\rangle \\
&= \sum_{(s,b)in\mathcal{S}\times\mathcal{B}} d_i^{\mathsf{p},\mu^{\mathsf{d}},\nu^\star}(s,\mu^{\mathsf{d}}(s),b)\beta_i(s,\mu^{\mathsf{d}}(s),b,\hat{V}) \\
&\leq \sum_{(s,b)\in\mathcal{S}\times\mathcal{B}} d_i^{\mathsf{p},\mu^{\mathsf{d}},\nu^\star}(s,\mu^{\mathsf{d}}(s),b)\sqrt{\frac{2C_{\mathsf{n}}\log\frac{KH}{\delta}}{N_i(s,\mu^{\mathsf{d}}(s),b)}\mathsf{Var}_{P^0_{i,s,\mu^{\mathsf{d}}(s),b}}(\widehat{V})} \\
&\quad + \sum_{(s,b)\in\mathcal{S}\times\mathcal{B}} d_i^{\mathsf{p},\mu^{\mathsf{d}},\nu^\star}(s,\mu^{\mathsf{d}}(s),b)\frac{\left(2C_{\mathsf{n}}+\sqrt{C_{\mathsf{n}}C_0}\right)H\log\frac{KH}{\delta}}{N_i(s,\mu^{\mathsf{d}}(s),b)} \\
&\overset{(i)}{\leq} \sum_{(s,b)\in\mathcal{S}\times\mathcal{B}} d_i^{\mathsf{p},\mu^{\mathsf{d}},\nu^\star}(s,\mu^{\mathsf{d}}(s),b)\underbrace{\sqrt{\frac{32C_{\mathsf{r}}^\star C_{\mathsf{n}}\log\frac{KH}{\delta}}{K\min\left\{d_i^{\mathsf{p},\mu^{\mathsf{d}},\nu^\star}(s,\mu^{\mathsf{d}}(s),b),\frac{1}{S(A+B)}\right\}}\mathsf{Var}_{P^0_{i,s,\mu^{\mathsf{d}}(s),b}}(\widehat{V})}}_{B_1} \\
&\quad + \sum_{(s,b)\in\mathcal{S}\times\mathcal{B}} d_i^{\mathsf{p},\mu^{\mathsf{d}},\nu^\star}(s,\mu^{\mathsf{d}}(s),b)\underbrace{\frac{16C_{\mathsf{r}}^\star\left(2C_{\mathsf{n}}+\sqrt{C_{\mathsf{n}}C_0}\right)H\log\frac{KH}{\delta}}{K\min\left\{d_i^{\mathsf{p},\mu^{\mathsf{d}},\nu^\star}(s,\mu^{\mathsf{d}}(s),b),\frac{1}{S(A+B)}\right\}}}_{B_2} .
\end{aligned}
\tag{70}
$$

From (61), we need to bound $\sum_{i=1}^H\sum_{(s,b)\in\mathcal{S}\times\mathcal{B}} d_i^{\mathsf{p},\mu^{\mathsf{d}},\nu^\star}(s,\mu^{\mathsf{d}}(s),b)B_1$ and $\sum_{i=1}^H\sum_{(s,b)\in\mathcal{S}\times\mathcal{B}} d_i^{\mathsf{p},\mu^{\mathsf{d}},\nu^\star}(s,\mu^{\mathsf{d}}(s),b)B_2$:

### D.2.1  Bounding $\sum_{i=1}^H\sum_{(s,b)\in\mathcal{S}\times\mathcal{B}} d_i^{\mathsf{p},\mu^{\mathsf{d}},\nu^\star}(s,\mu^{\mathsf{d}}(s),b)B_1$

Combining (68) with $\sum_{i=1}^H\sum_{(s,b)\in\mathcal{S}\times\mathcal{B}} d_i^{\mathsf{p},\mu^{\mathsf{d}},\nu^\star}(s,\mu^{\mathsf{d}}(s),b)B_1$ yields

$$
\begin{aligned}
&\sum_{i=1}^H\sum_{(s,b)\in\mathcal{S}\times\mathcal{B}} d_i^{\mathsf{p},\mu^{\mathsf{d}},\nu^\star}(s,\mu^{\mathsf{d}}(s),b)B_1 \\
&= \sum_{i=1}^H\sum_{(s,b)\in\mathcal{S}\times\mathcal{B}} d_i^{\mathsf{p},\mu^{\mathsf{d}},\nu^\star}(s,\mu^{\mathsf{d}}(s),b)\sqrt{\frac{32C_{\mathsf{r}}^\star C_{\mathsf{n}}\log\frac{KH}{\delta}}{K\min\left\{d_i^{\mathsf{p},\mu^{\mathsf{d}},\nu^\star}(s,\mu^{\mathsf{d}}(s),b),\frac{1}{S(A+B)}\right\}}\mathsf{Var}_{P^0_{i,s,\mu^{\mathsf{d}}(s),b}}(\widehat{V})} \\
&\leq \sum_{i=1}^H\sum_{(s,b)\in\mathcal{S}\times\mathcal{B}} d_i^{\mathsf{p},\mu^{\mathsf{d}},\nu^\star}(s,\mu^{\mathsf{d}}(s),b)\times \\
&\qquad \max\left\{\sqrt{\frac{32C_{\mathsf{r}}^\star C_{\mathsf{n}}\log\frac{KH}{\delta}}{Kd_i^{\mathsf{p},\mu^{\mathsf{d}},\nu^\star}(s,\mu^{\mathsf{d}}(s),b)}\mathsf{Var}_{P^0_{i,s,\mu^{\mathsf{d}}(s),b}}(\widehat{V})},\sqrt{\frac{32C_{\mathsf{r}}^\star C_{\mathsf{n}}S(A+B)\log\frac{KH}{\delta}}{K}\mathsf{Var}_{P^0_{i,s,\mu^{\mathsf{d}}(s),b}}(\widehat{V})}\right\},
\end{aligned}
\tag{71}
$$

Therefore, we have

$$\sum_{i=1}^{H} \sum_{(s,b)\in\mathcal{S}\times\mathcal{B}} d_i^{\mathsf{p},\mu^{\mathsf{d}},\nu^\star}(s,\mu^{\mathsf{d}}(s),b)B_1$$

$$\leq \sum_{i=1}^{H} \sum_{(s,b)\in\mathcal{S}\times\mathcal{B}} \sqrt{\frac{32 C_{\mathsf{r}}^\star C_{\mathsf{n}} \log\frac{KH}{\delta}}{K} d_i^{\mathsf{p},\mu^{\mathsf{d}},\nu^\star}(s,\mu^{\mathsf{d}}(s),b)\mathsf{Var}_{P^0_{i,s,\mu^{\mathsf{d}}(s),b}}\left(\widehat{V}\right)}$$

$$+ \sum_{i=1}^{H} \sum_{(s,b)\in\mathcal{S}\times\mathcal{B}} d_i^{\mathsf{p},\mu^{\mathsf{d}},\nu^\star}(s,\mu^{\mathsf{d}}(s),b)\sqrt{\frac{32 C_{\mathsf{r}}^\star C_{\mathsf{n}} S(A+B) \log\frac{KH}{\delta}}{K}\mathsf{Var}_{P^0_{i,s,\mu^{\mathsf{d}}(s),b}}\left(\widehat{V}\right)}$$

$$\leq \sqrt{\frac{32 C_{\mathsf{r}}^\star C_{\mathsf{n}} S(A+B) \log\frac{KH}{\delta}}{K}} \left( \sqrt{H \sum_{i=1}^{H} \sum_{(s,b)\in\mathcal{S}\times\mathcal{B}} d_i^{\mathsf{p},\mu^{\mathsf{d}},\nu^\star}(s,\mu^{\mathsf{d}}(s),b)\mathsf{Var}_{P^0_{i,s,\mu^{\mathsf{d}}(s),b}}\left(\widehat{V}\right)} \right.$$

$$\left. + \sqrt{\sum_{i=1}^{H} \sum_{(s,b)\in\mathcal{S}\times\mathcal{B}} d_i^{\mathsf{p},\mu^{\mathsf{d}},\nu^\star}(s,\mu^{\mathsf{d}}(s),b)\mathsf{Var}_{P^0_{i,s,\mu^{\mathsf{d}}(s),b}}\left(\widehat{V}\right)} \times \sqrt{\sum_{i=1}^{H} \sum_{(s,b)\in\mathcal{S}\times\mathcal{B}} d_i^{\mathsf{p},\mu^{\mathsf{d}},\nu^\star}(s,\mu^{\mathsf{d}}(s),b)} \right)$$

$$= \sqrt{\frac{128 C_{\mathsf{r}}^\star C_{\mathsf{n}} H S(A+B) \log\frac{KH}{\delta}}{K} \sum_{i=1}^{H} \sum_{(s,b)\in\mathcal{S}\times\mathcal{B}} d_i^{\mathsf{p},\mu^{\mathsf{d}},\nu^\star}(s,\mu^{\mathsf{d}}(s),b)\mathsf{Var}_{P^0_{i,s,\mu^{\mathsf{d}}(s),b}}\left(\widehat{V}\right)}, \tag{72}$$

where the last inequality follows from the Cauchy-Schwarz inequality.

Then, we introduce the following lemma about $\sum_{i=1}^{H} \sum_{(s,b)\in\mathcal{S}\times\mathcal{B}} d_i^{\mathsf{p},\mu^{\mathsf{d}},\nu^\star}(s,\mu^{\mathsf{d}}(s),b)\mathsf{Var}_{P_{i,s,\mu^{\mathsf{d}}(s),b}}\left(\widehat{V}\right)$, with its proof in Appendix G.4.

**Lemma D.4.** *Considering $\forall\delta\in(0,1)$, with probability at least $1-\delta$, one has: for any product policy $(\widehat{\mu},\widehat{\nu})$,*

$$\sum_{i=1}^{H} \sum_{(s,b)\in\mathcal{S}\times\mathcal{B}} d_i^{\mathsf{p},\mu^{\mathsf{d}},\nu^\star}(s,\mu^{\mathsf{d}}(s),b)\mathsf{Var}_{P^0_{i,s,a,b}}\left(\widehat{V}_{i+1}\right) \leq H\min\left\{\frac{2(H\sigma^+ - 1 + (1-\sigma^+)^H)}{(\sigma^+)^2}, H\right\}$$

$$\times \left( 4\sum_{i=1}^{H} \sum_{(s,b)\in\mathcal{S}\times\mathcal{B}} d_i^{\mathsf{p},\mu^{\mathsf{d}},\nu^\star}(s,\mu^{\mathsf{d}}(s),b)\beta_i(s,\mu^{\mathsf{d}}(s),b,\widehat{V}) + (H+3) \right). \tag{73}$$

Armed with Lemma D.4, (72) can be further bounded as

$$\sum_{i=1}^{H} \sum_{(s,b)\in\mathcal{S}\times\mathcal{B}} d_i^{\mathsf{p},\mu^{\mathsf{d}},\nu^\star}(s,\mu^{\mathsf{d}}(s),b)B_1$$

$$\leq \sqrt{\frac{128 C_{\mathsf{r}}^\star C_{\mathsf{n}} H S(A+B) \log\frac{KH}{\delta}}{K}} \sqrt{H\min\left\{\frac{2(H\sigma^+ - 1 + (1-\sigma^+)^H)}{(\sigma^+)^2}, H\right\}}$$

$$\times \sqrt{\left( 4\sum_{i=1}^{H} \sum_{(s,b)\in\mathcal{S}\times\mathcal{B}} d_i^{\mathsf{p},\mu^{\mathsf{d}},\nu^\star}(s,\mu^{\mathsf{d}}(s),b)\beta_i(s,\mu^{\mathsf{d}}(s),b,\widehat{V}) + (H+3) \right)}. \tag{74}$$

### D.2.2 Bounding $\sum_{i=1}^{H} \sum_{(s,b) \in \mathcal{S} \times \mathcal{B}} d_i^{\mathsf{p}, \mu^{\mathsf{d}}, \nu^{\star}}(s, \mu^{\mathsf{d}}(s), b) B_2$

Combining (67) with $\sum_{i=1}^{H} \sum_{(s,b) \in \mathcal{S} \times \mathcal{B}} d_i^{\mathsf{p}, \mu^{\mathsf{d}}, \nu^{\star}}(s, \mu^{\mathsf{d}}(s), b) B_2$ yields

$$
\begin{aligned}
&\sum_{i=1}^{H} \sum_{(s,b) \in \mathcal{S} \times \mathcal{B}} d_i^{\mathsf{p}, \mu^{\mathsf{d}}, \nu^{\star}}(s, \mu^{\mathsf{d}}(s), b) B_2 \\
&= \sum_{i=1}^{H} \sum_{(s,b) \in \mathcal{S} \times \mathcal{B}} d_i^{\mathsf{p}, \mu^{\mathsf{d}}, \nu^{\star}}(s, \mu^{\mathsf{d}}(s), b) \frac{16 C_{\mathsf{r}}^{\star} \left(2 C_{\mathsf{n}} + \sqrt{C_{\mathsf{n}} C_3}\right) H \log \frac{KH}{\delta}}{K \min \left\{ d_i^{\mathsf{p}, \mu^{\mathsf{d}}, \nu^{\star}}(s, \mu^{\mathsf{d}}(s), b), \frac{1}{S(A+B)} \right\}} \\
&\overset{(i)}{\leq} \frac{32 C_{\mathsf{r}}^{\star} \left(2 C_{\mathsf{n}} + \sqrt{C_{\mathsf{n}} C_3}\right) H^2 S(A+B) \log \frac{KH}{\delta}}{K},
\end{aligned}
\tag{75}
$$

where the inequality holds by the trivial fact

$$
\begin{aligned}
&\sum_{(s,b) \in \mathcal{S} \times \mathcal{B}} \frac{d_i^{\mathsf{p}, \mu^{\mathsf{d}}, \nu^{\star}}(s, \mu^{\mathsf{d}}(s), b)}{\min \left\{ d_i^{\mathsf{p}, \mu^{\mathsf{d}}, \nu^{\star}}(s, \mu^{\mathsf{d}}(s), b), \frac{1}{S(A+B)} \right\}} \\
&\leq \sum_{(s,b) \in \mathcal{S} \times \mathcal{B}} d_i^{\mathsf{p}, \mu^{\mathsf{d}}, \nu^{\star}}(s, \mu^{\mathsf{d}}(s), b) \left( \frac{1}{d_i^{\mathsf{p}, \mu^{\mathsf{d}}, \nu^{\star}}(s, \mu^{\mathsf{d}}(s), b)} + \frac{1}{1/S(A+B)} \right) \\
&= \sum_{(s,b) \in \mathcal{S} \times \mathcal{B}} 1 + S(A+B) \sum_{(s,b) \in \mathcal{S} \times \mathcal{B}} d_i^{\mathsf{p}, \mu^{\mathsf{d}}, \nu^{\star}}(s, \mu^{\mathsf{d}}(s), b) \leq 2S(A+B).
\end{aligned}
\tag{76}
$$

### D.2.3 Putting all together

Combining (74) and (75), we obtain

$$
\begin{aligned}
&\sum_{i=1}^{H} \sum_{(s,b) \in \mathcal{S} \times \mathcal{B}} d_i^{\mathsf{p}, \mu^{\mathsf{d}}, \nu^{\star}}(s, \mu^{\mathsf{d}}(s), b) \beta_i(s, \mu^{\mathsf{d}}(s), b, \widehat{V}) \\
&\leq \sqrt{\frac{128 C_{\mathsf{r}}^{\star} C_{\mathsf{n}} H^2 S(A+B) \log \frac{KH}{\delta}}{K} \min \left\{ \frac{2(H\sigma^+ - 1 + (1-\sigma^+)^H)}{(\sigma^+)^2}, H \right\}} \\
&\quad \times \sqrt{\left( \left( 4 \sum_{i=1}^{H} \sum_{(s,b) \in \mathcal{S} \times \mathcal{B}} d_i^{\mathsf{p}, \mu^{\mathsf{d}}, \nu^{\star}}(s, \mu^{\mathsf{d}}(s), b) \beta_i(s, \mu^{\mathsf{d}}(s), b, \widehat{V}) + (H+3) \right)\right.} \\
&\quad + \frac{32 C_{\mathsf{r}}^{\star} \left(2 C_{\mathsf{n}} + \sqrt{C_{\mathsf{n}} C_3}\right) H^2 S(A+B) \log \frac{KH}{\delta}}{K},
\end{aligned}
\tag{77}
$$

which can further bound as

$$\sum_{i=1}^{H} \sum_{(s,b)\in\mathcal{S}\times\mathcal{B}} d_i^{\mathsf{p},\mu^{\mathsf{d}},\nu^\star}(s,\mu^{\mathsf{d}}(s),b)\beta_i(s,\mu^{\mathsf{d}}(s),b,\widehat{V})$$

$$\leq \sqrt{\frac{128 C_{\mathsf{r}}^\star C_{\mathsf{n}} H^2 (H+3) S(A+B) \log\frac{KH}{\delta}}{K} \min\left\{\frac{2(H\sigma^+ - 1 + (1-\sigma^+)^H)}{(\sigma^+)^2}, H\right\}}$$

$$+ \frac{32 C_{\mathsf{r}}^\star \left(2C_{\mathsf{n}} + \sqrt{C_{\mathsf{n}}C_3}\right) H^2 S(A+B)\log\frac{KH}{\delta}}{K} + \sqrt{\frac{512 C_{\mathsf{r}}^\star C_{\mathsf{n}} H^2 S(A+B)\log\frac{KH}{\delta}}{K}}$$

$$\times \sqrt{\min\left\{\frac{2(H\sigma^+ - 1 + (1-\sigma^+)^H)}{(\sigma^+)^2}, H\right\} \sum_{i=1}^{H}\sum_{(s,b)\in\mathcal{S}\times\mathcal{B}} d_i^{\mathsf{p},\mu^{\mathsf{d}},\nu^\star}(s,\mu^{\mathsf{d}}(s),b)\beta_i(s,\mu^{\mathsf{d}}(s),b,\widehat{V})}$$

$$\leq \sqrt{\frac{128 C_{\mathsf{r}}^\star C_{\mathsf{n}} H^2 (H+3) S(A+B) \log\frac{KH}{\delta}}{K} \min\left\{\frac{2(H\sigma^+ - 1 + (1-\sigma^+)^H)}{(\sigma^+)^2}, H\right\}}$$

$$+ \frac{32 C_{\mathsf{r}}^\star \left(2C_{\mathsf{n}} + \sqrt{C_{\mathsf{n}}C_3}\right) H^2 S(A+B)\log\frac{KH}{\delta}}{K}$$

$$+ \frac{256 C_{\mathsf{r}}^\star C_{\mathsf{n}} H^2 S(A+B)\log\frac{KH}{\delta}}{K} \min\left\{\frac{2(H\sigma^+ - 1 + (1-\sigma^+)^H)}{(\sigma^+)^2}, H\right\}$$

$$+ \frac{1}{2}\sum_{i=1}^{H}\sum_{(s,b)\in\mathcal{S}\times\mathcal{B}} d_i^{\mathsf{p},\mu^{\mathsf{d}},\nu^\star}(s,\mu^{\mathsf{d}}(s),b)\beta_i(s,\mu^{\mathsf{d}}(s),b,\widehat{V}), \tag{78}$$

where the last relation follows from the AM-GM inequality. Rearranging the terms, it follows that

$$\sum_{i=1}^{H} \sum_{(s,b)\in\mathcal{S}\times\mathcal{B}} d_i^{\mathsf{p},\mu^{\mathsf{d}},\nu^\star}(s,\mu^{\mathsf{d}}(s),b)\beta_i(s,\mu^{\mathsf{d}}(s),b,\widehat{V})$$

$$\leq \sqrt{\frac{512 C_{\mathsf{r}}^\star C_{\mathsf{n}} H^2 (H+3) S(A+B) \log\frac{KH}{\delta}}{K} \min\left\{\frac{2(H\sigma^+ - 1 + (1-\sigma^+)^H)}{(\sigma^+)^2}, H\right\}}$$

$$+ \frac{64 C_{\mathsf{r}}^\star \left(2C_{\mathsf{n}} + \sqrt{C_{\mathsf{n}}C_3}\right) H^2 S(A+B)\log\frac{KH}{\delta}}{K}$$

$$+ \frac{512 C_{\mathsf{r}}^\star C_{\mathsf{n}} H^2 S(A+B)\log\frac{KH}{\delta}}{K} \min\left\{\frac{2(H\sigma^+ - 1 + (1-\sigma^+)^H)}{(\sigma^+)^2}, H\right\}$$

$$\leq \sqrt{\frac{512 C_{\mathsf{r}}^\star C_{\mathsf{n}} H^2 (H+3) S(A+B) \log\frac{KH}{\delta}}{K} \min\left\{\frac{2(H\sigma^+ - 1 + (1-\sigma^+)^H)}{(\sigma^+)^2}, H\right\}}$$

$$+ \frac{C_{\mathsf{r}}^\star C_2 H^2 S(A+B)\log\frac{KH}{\delta}}{K} \min\left\{\frac{2(H\sigma^+ - 1 + (1-\sigma^+)^H)}{(\sigma^+)^2}, H\right\}. \tag{79}$$

Along with the above analysis, we are ready to bound $V_1^{\star,\sigma^+}(\varrho) - V_1^{\widehat{\mu},\star,\sigma^+}(\varrho)$: There exist some sufficiently large constants $C_1, C_2, C_3 > 0$, and

$$V_1^{\star,\widehat{\nu},\sigma^+}(\varrho) - V_1^{\star,\sigma^+}(\varrho) \leq \sqrt{\frac{C_{\mathsf{r}}^\star C_1 H^3 S(A+B)\log\frac{KH}{\delta}}{K} \min\left\{\frac{2(H\sigma^+ - 1 + (1-\sigma^+)^H)}{(\sigma^+)^2}, H\right\}}$$

$$+ \frac{C_{\mathsf{r}}^\star C_2 H^2 S(A+B)\log\frac{KH}{\delta}}{K} \min\left\{\frac{2(H\sigma^+ - 1 + (1-\sigma^+)^H)}{(\sigma^+)^2}, H\right\}$$

$$\leq \sqrt{\frac{C_{\mathsf{r}}^\star C_3 H^3 S(A+B)\log\frac{KH}{\delta}}{K} \min\left\{\frac{2(H\sigma^+ - 1 + (1-\sigma^+)^H)}{(\sigma^+)^2}, H\right\}}, \tag{80}$$

where the last inequality follows from condition (22).

## D.3 Step 3: Summing up

Consequently, we obtain the upper bound of $V_1^{\star,\widehat{\nu},\sigma^+}(\varrho) - V_1^{\widehat{\mu},\widehat{\nu},\sigma^+}(\varrho)$ in (80). Similarly,

$$
\begin{aligned}
&V_1^{\star,\sigma^-}(\varrho) - V_1^{\widehat{\mu},\star,\sigma^-}(\varrho) \\
&\leq \sqrt{\frac{C_r^\star C_3 H^2 S(A+B)\log\frac{KH}{\delta}}{K} \min\left\{\frac{(H+1)(H\sigma^- - 1 + (1-\sigma^-)^H)}{(\sigma^-)^2}, H\right\}},
\end{aligned}
\tag{81}
$$

which directly leads to

$$
\begin{aligned}
\mathrm{Gap}(\widehat{\mu},\widehat{\nu}) \leq c_1 &\sqrt{\frac{C_r^\star H^2 S(A+B)\log\frac{KH}{\delta}}{K}} \\
&\times \sqrt{\min\left\{\frac{2(H\sigma^+ - 1 + (1-\sigma^+)^H)}{(\sigma^+)^2}, \frac{2(H\sigma^- - 1 + (1-\sigma^-)^H)}{(\sigma^-)^2}, H\right\}},
\end{aligned}
\tag{82}
$$

for some sufficiently large $c_1$ and

$$
K \geq HS(A+B)\log\frac{KH}{\delta}\min\left\{\frac{2(H\sigma^+ - 1 + (1-\sigma^+)^H)}{(\sigma^+)^2}, \frac{2(H\sigma^- - 1 + (1-\sigma^-)^H)}{(\sigma^-)^2}, H\right\}.
$$

**Discussion of (82).** For the term $T = \min\left(f(\sigma^+,\sigma^-), H\right)$, considering the interchangeability between $\sigma^+$ and $\sigma^-$, we define $g(\sigma^+, H) = H\sigma^+ - H(1-\sigma^+)^H - (\sigma^+)^2 H$. For $H \geq 2$, we derive the first derivative as $\frac{\partial g(\sigma^+,H)}{\partial \sigma^+} = H + H^2(1-\sigma^+)^{H-1} - 2H\sigma^+$. The second derivative is given by $\frac{\partial^2 g(\sigma^+,H)}{\partial(\sigma^+)^2} = -H^2(H-1)(1-\sigma^+)^{H-2} - 2H < 0$, indicating that $g(\sigma^+, H)$ is concave. By evaluating the first derivative at the boundaries, we find $\frac{\partial g(\sigma^+,H)}{\partial \sigma^+}|_{\sigma^+\to 0} \to H^2 + H > 0$ and $\frac{\partial g(\sigma^+,H)}{\partial \sigma^+}|_{\sigma^+=1} = -H < 0$, showing that $g(\sigma^+, H)$ first increases monotonically, reaches a maximum at some point $\sigma^\star$, and then decreases monotonically. Furthermore, since $g(\sigma^+ \to 0, H) \to -H < 0$ and $g(\sigma^+ = 1, H) = 0$, there exists $0 < \sigma^0 < 1$ such that $g(\sigma^0, H) = 0$. If $\sigma^0 \lesssim \min\{\sigma^+,\sigma^-\} \lesssim 1$, we have $T = H$. Otherwise, $T = \min\left\{\frac{(H\sigma^+ - 1 + (1-\sigma^+)^H)}{(\sigma^+)^2}, \frac{(H\sigma^- - 1 + (1-\sigma^-)^H)}{(\sigma^-)^2}\right\}$.

# E  Proof of Theorem 4.4

We focus on a simple class of RTZMGs: robust Markov decision processes (RMDPs), which are single-agent versions of RTZMGs. Recall that an RTZMG with an uncertainty set is $\mathcal{MG} = \{\mathcal{S}, \mathcal{A}, \mathcal{B}, \mathcal{U}^{\sigma^+}(P^0), \mathcal{U}^{\sigma^-}(P^0), r, H\}$. For illustration convenience, we assume $A \geq B$ and $|\mathcal{B}| = 1$, meaning the min-player's actions do not affect transitions or rewards. Thus, finding a robust NE in RTZMGs reduces to finding the max-player's optimal policy in a corresponding RMDP $\mathcal{M}_r = \{\mathcal{S}, \mathcal{A}, \mathcal{U}^{\sigma^+}(P^0), r, H\}$.

Thus, we construct the lower bound for finding the optimal policy in RTZMGs, which also implies a lower bound for finding robust NE in RTZMGs. We start by re-stating a useful property about KL divergence from Lemma 2.7 in [36].

**Lemma E.1.** $\forall p, q \in (0, 1)$, it holds that

$$
\mathsf{KL}(p \parallel q) \leq \frac{(p-q)^2}{q(1-q)}.
\tag{83}
$$

## E.1  Step 1: Constructing a family of hard Markov game instances

The hard instances developed here differ from standard MDP since we need to consider that the transition kernel can be perturbed in robust MDPs.

### E.1.1  Constructing hard robust MDP instances

We introduce an auxiliary collection $\Phi \subseteq \{0,1\}^H$ comprising $H$-dimensional vectors. Resorting to the Gilbert-Varshamov lemma [15], there also exists a set $\Phi \subseteq \{0,1\}^H$ such that:

$$\text{for any } \phi, \widetilde{\phi} \in \Phi \text{ obeying } \phi \neq \widetilde{\phi}: \quad \|\phi - \widetilde{\phi}\|_1 \geq \frac{H}{8} \quad \text{and} \quad |\Phi| \geq e^{H/8}. \tag{84}$$

Bearing this in mind, we construct a set of RMDPs as

$$\mathcal{M}(\mathcal{F}, \Phi) := \left\{ \mathcal{M}_f^{\phi} = \left( \mathcal{S}, \mathcal{A}, \mathcal{U}^{\sigma^+}(P^{f,\phi}), r, H \right) \, | \, f \in \mathcal{F} = \{0, 1, \cdots, SA - 1\}, \phi = [\phi_h]_{1 \leq h \leq H} \in \Phi \right\}, \tag{85}$$

where $\mathcal{S} = \{0, 1, \ldots, S - 1\}$, $\mathcal{A} = \{0, 1, \cdots, A - 1\}$, and $\sigma^+$ will be introduced momentarily.

In simple terms, the collection $\mathcal{M}(\mathcal{F}, \Phi)$ consists of $SA$ subsets, each containing $|\Phi|$ different RMDPs associated with some $f \in \mathcal{F}$. The state space for each RMDP $\mathcal{M}_f^{\phi} \in \mathcal{M}(\mathcal{F}, \Phi)$, denoted as $\mathcal{S}_{\text{one}}$, includes two types of states: $\mathcal{M} = \{m_i \mid i \in \mathcal{F}\}$ and $\mathcal{N} = \{n_i \mid i \in \mathcal{F}\}$. Each state in $\mathcal{M}$ and $\mathcal{N}$ has two possible actions, $\mathcal{A}_{\text{one}} = \{0, 1\}$. Thus, there are a total of $2SA$ states and $4SA$ state-action pairs.

Now, we can define the transition kernels for $\mathcal{M}(\mathcal{F}, \Phi)$. For any RMDP $\mathcal{M}_f^{\phi} \in \mathcal{M}(\mathcal{F}, \Phi)$, the transition kernel $P^{f,\phi} = \{P_h^{f,\phi}\}_{h=1}^H$ is defined as follows, for any $(s, a, s', h) \in \mathcal{S}_{\text{one}} \times \mathcal{A}_{\text{one}} \times \mathcal{S}_{\text{one}} \times [H]$,

$$P_h^{f,\phi}(s' \mid s, a) = \begin{cases} p\mathbb{1}(s' = n_f) + (1 - p)\mathbb{1}(s' = s), & \text{if } s = m_f, a = \phi_h; \\ q\mathbb{1}(s' = n_f) + (1 - q)\mathbb{1}(s' = s), & \text{if } s = m_f, a = 1 - \phi_h; \\ \mathbb{1}(s' = s), & \text{otherwise}, \end{cases} \tag{86}$$

where $p > q \geq \frac{1}{2}$.

In addition, the reward function is defined as

$$\forall (h, s, a) \in [H] \times \mathcal{S}_{\text{one}} \times \mathcal{A}_{\text{one}}: \quad r_h(s, a) = \begin{cases} 1, & \text{if } s \in \mathcal{N}; \\ 0, & \text{otherwise}. \end{cases} \tag{87}$$

### E.1.2  Uncertainty set of the transition kernels

Denote the transition kernel vector as

$$\forall (h, s, a) \in [H] \times \mathcal{S}_{\text{one}} \times \mathcal{A}_{\text{one}}: \quad P_{h,s,a}^{f,\phi} := P_h^{f,\phi}(\cdot \mid s, a) \in \Delta(\mathcal{S}). \tag{88}$$

Recall the uncertainty set defined in (1). $\mathcal{U}^{\sigma^+}(P^{f,\phi})$ represents

$$\mathcal{U}^{\sigma^+}(P^{f,\phi}) := \otimes \, \mathcal{U}^{\sigma^+}(P_{h,s,a}^{f,\phi}), \quad \mathcal{U}^{\sigma^+}(P_{h,s,a}^{f,\phi}) := \left\{ \widetilde{P}_{h,s,a}^{f,\phi} \in \Delta(\mathcal{S}) : \rho\left(\widetilde{P}_{h,s,a}^{f,\phi} - P_{h,s,a}^{f,\phi}\right) \leq \sigma^+ \right\},$$

where $\otimes$ represents the Cartesian product over $(h, s, a) \in [H] \times \mathcal{S}_{\text{one}} \times \mathcal{A}_{\text{one}}$. For the convenience of the subsequent proof, we analyze the TV distance as an uncertainty set for example, which means

$$\mathcal{U}^{\sigma^+}(P_{h,s,a}^{f,\phi}) := \left\{ \widetilde{P}_{h,s,a}^{f,\phi} \in \Delta(\mathcal{S}) : \frac{1}{2} \left\| \widetilde{P}_{h,s,a}^{f,\phi} - P_{h,s,a}^{f,\phi} \right\| \leq \sigma^+ \right\}. \tag{89}$$

For any RMDP $\mathcal{M}_f^{\phi} \in \mathcal{M}(\mathcal{F}, \Phi)$ and any $(h, s, a, s') \in [H] \times \mathcal{S}_{\text{one}} \times \mathcal{A}_{\text{one}} \times \mathcal{S}_{\text{one}}$, we define the minimum transition probability from $(s, a)$ to $s'$, determined by any perturbed transition kernel $P_{h,s,a} \in \mathcal{U}^{\sigma^+}(P_{h,s,a}^{f,\phi})$, as

$$P_h^{\text{inf},f,\phi}(s' \mid s, a) := \inf_{P_{h,s,a} \in \mathcal{U}^{\sigma^+}(P_{h,s,a}^{f,\phi})} P_h(s' \mid s, a) = \max\{P_h(s' \mid s, a) - \sigma^+, 0\}, \tag{90}$$

where the last equation inherits from the definition of $\mathcal{U}^{\sigma^+}(\cdot)$ in (89), with the remaining probability distributed to other states. We also define the transition from each $s \in \mathcal{M}$ to the corresponding state $s^{m \to n} \in \mathcal{N}$ for any $\mathcal{M}_f^{\phi}$: for all $h \in [H]$,

$$\text{for } m_f: \quad p_h^{\text{inf}} := P_h^{\text{inf},f,\phi}(n_f \mid m_f, \phi_h) = p - \sigma^+,$$
$$q_h^{\text{inf}} := P_h^{\text{inf},f,\phi}(n_f \mid m_f, 1 - \phi_h) = q - \sigma^+. \tag{91}$$

It is obvious that

$$p_1^{\text{inf}} = p_2^{\text{inf}} = \cdots p_H^{\text{inf}}, \quad q_1^{\text{inf}} = q_2^{\text{inf}} = \cdots q_H^{\text{inf}}, \tag{92}$$

which motivates us to abbreviate them consistently as $p^{\text{inf}} := p_1^{\text{inf}}$ and $q^{\text{inf}} := q_1^{\text{inf}}$ later.

### E.1.3 Robust value functions and optimal policies

We now define the robust value functions and identify the optimal policies for RMDP instances. For any RMDP $\mathcal{M}_f^\phi \in \mathcal{M}(\mathcal{F}, \Phi)$, let $\widetilde{\mu}^{\star,f,\phi} = \{\mu_h^{\star,f,\phi}\}_{h=1}^H$ represent the optimal policy, given that $\nu$ is deterministic. At each step $h$, we use $V_h^{\widetilde{\mu},\sigma^+,f,\phi}$ and $V_h^{\star,\sigma^+,f,\phi}$ to denote the robust value function of any policy $\widetilde{\mu}$ and the optimal policy $\widetilde{\mu}^{\star,f,\phi}$, respectively, under uncertainty level $\sigma^+$. The following lemma highlights key properties of robust value functions and optimal policies; the proof is deferred to Appendix H.1.

**Lemma E.2.** *Consider any $\mathcal{M}_f^\phi \in \mathcal{M}(\mathcal{F}, \Phi)$ and any policy $\widetilde{\mu}$. Defining*

$$m_h^{\widetilde{\mu},f,\phi} = p^{\text{inf}}\widetilde{\mu}_h(\phi_h \mid m_f) + q^{\text{inf}}\widetilde{\mu}_h(1 - \phi_h \mid m_f), \tag{93}$$

*it holds that*

$$\forall h \in [H]: \quad V_h^{\widetilde{\mu},\sigma^+,f,\phi}(m_f) = m_h^{\widetilde{\mu},f,\phi}V_{h+1}^{\widetilde{\mu},\sigma^+,f,\phi}(n_f) + (1 - m_h^{\widetilde{\mu},f,\phi})V_{h+1}^{\widetilde{\mu},\sigma^+,f,\phi}(m_f), \tag{94a}$$

$$\forall (s,h) \in \mathcal{N} \times [H]: \quad V_h^{\widetilde{\mu},\sigma^+,f,\phi}(s) = 1 + (1 - \sigma^+)V_{h+1}^{\widetilde{\mu},\sigma^+,f,\phi}(s) + \sigma^+ V_{h+1}^{\widetilde{\mu},\sigma^+,f,\phi}(m_f). \tag{94b}$$

*In addition, for all $h \in [H]$, the optimal policy and the optimal value function obey*

$$\widetilde{\mu}_h^{\star,f,\phi}(\phi_h \mid m_f) = \widetilde{\mu}_h^{\star,f,\phi}(\phi_h \mid n_f) = 1, \tag{95}$$

$$V_h^{\star,\sigma^+,f,\phi}(m_f) = p^{\text{inf}}V_{h+1}^{\widetilde{\mu},\sigma^+,f,\phi}(n_f) + (1 - p^{\text{inf}})V_{h+1}^{\widetilde{\mu},\sigma^+,f,\phi}(m_f). \tag{96}$$

### E.1.4 Construction of the history/batch dataset

In the nominal environment $\mathcal{M}_f^{\phi,\text{n}}$, a batch dataset is generated with $K$ independent sample trajectories with length $H$ per trajectory, according to (3) and based on the initial state distribution $\varrho^{\text{n}}$ and behavior policy $\widetilde{\mu}^{\text{n}} = \{\mu_h^{\text{n}}\}_{h=1}^H$ satisfying

$$\varrho^{\text{n}}(s) = \varrho(s) \quad \text{and} \quad \widetilde{\mu}_h^{\text{n}}(a \mid s) = \frac{1}{2}, \qquad \forall (s,a,h) \in \mathcal{S}_{\text{one}} \times \mathcal{A}_{\text{one}} \times [H]. \tag{97}$$

We define the nominal transition kernels for $\mathcal{M}_f^{\phi,\text{n}}$, where any state $m_i \in \mathcal{M}$ transitions only to the corresponding $n_i \in \mathcal{N}$ or remains at itself. For simplicity, for any $s = m_i \in \mathcal{M}$, we denote the corresponding state $n_i \in \mathcal{N}$ as $s^{m \to n}$. The basic nominal transition kernel is defined as: For all $(h,s,a) \in [H] \times \mathcal{S}_{\text{one}} \times \mathcal{A}_{\text{one}}$,

$$P_h^\star(s' \mid s,a) = \begin{cases} (p + \Delta)\mathbb{1}(s' = s^{m \to n}) + (1 - p - \Delta)\mathbb{1}(s' = s), & \text{if} \quad s \in \mathcal{M}, a = \phi_h; \\ p\mathbb{1}(s' = s^{m \to n}) + (1 - p)\mathbb{1}(s' = s), & \text{if} \quad s \in \mathcal{M}, a = 1 - \phi_h; \\ \mathbb{1}(s' = s), & \text{if} \quad s \in \mathcal{N}. \end{cases} \tag{98}$$

In other words, the transition kernel of each $\mathcal{M}_f^\phi \in \mathcal{M}(\mathcal{F}, \Phi)$ differs slightly from the basic nominal transition kernel $\mathcal{M}_f^{\phi,\text{n}}$ when $s = m_f$, making all components within $\mathcal{M}(\mathcal{F}, \Phi)$ close to each other. Specifically, $p$ and $q$ are set according to

$$0 \le p \le p + \Delta \le 1 \text{ and } 0 \le q = p - \Delta \text{ for some } p, \Delta > 0. \tag{99}$$

Without loss of generality, let $\sigma^+ \in (0, 1 - c_0]$ for some $0 < c_0 < 1$ be the uncertainty level. Taking $c_2 \le \frac{1}{4}$ and $c_1 := \frac{c_0}{2} \le \frac{1}{4}$, $p$ and $\Delta$ are set as

$$p = \begin{cases} \frac{c_2}{H}, & \text{if } \sigma^+ \le \frac{c_2}{2H} \\ \left(1 + \frac{c_1}{H}\right)\sigma^+ & \text{otherwise} \end{cases} \quad \text{and} \quad \Delta \le \begin{cases} \frac{c_2}{2H}, & \text{if } \sigma^+ \le \frac{c_2}{2H} \\ \frac{c_1}{H}\sigma^+ & \text{otherwise} \end{cases} \tag{100}$$

which establishes the fact that

$$p + \Delta \geq p \geq q = p - \Delta \geq \max\left\{\frac{c_2}{2H}, \sigma^+\right\}. \tag{101}$$

Combined with $H \geq 2$, it is easily verified that $0 \leq p + \Delta \leq 1$ as follows:

$$\text{when } \sigma^+ > \frac{c_2}{2H} : \quad \left(1 + \frac{c_1}{H}\right)\sigma^+ + \frac{c_1}{H}\sigma^+ \leq 1 - c_0 + \frac{2c_1}{H}\sigma^+ \leq 1 - \frac{c_0(H-1)}{H} < 1,$$

$$\text{when } \sigma^+ \leq \frac{c_2}{2H} : \quad \frac{3c_2}{2H} \leq 1. \tag{102}$$

In addition, let $\overline{\varrho}(s)$ represent a state distribution supported on the state subset $(m_f, n_f) \in \mathcal{M} \times \mathcal{N}$:

$$\overline{\varrho}(s) = \frac{1}{CSA}\mathbb{1}(s = m_f) + \left(1 - \frac{1}{CSA}\right)\mathbb{1}(s = n_f), \tag{103}$$

where $\mathbb{1}(\cdot)$ is the indicator function, and $C > 0$ is some constant that determines the concentrability coefficient $C_r^\star$ (as we shall detail momentarily) and obeys

$$\frac{1}{CSA} \leq \frac{1}{4}. \tag{104}$$

As it turns out, for any MDP $\mathcal{M}_\phi^f$, the occupancy distributions of the above batch dataset are the same (due to interchangeability) and admit the following simple characterization:

$$\forall (s, a) \in \mathcal{S}_{\text{one}} \times \mathcal{A}_{\text{one}}, \qquad d_1^{\mathsf{n}, P^{\phi,f}}(s, a) = \frac{1}{2}\overline{\varrho}(s), \tag{105a}$$

$$\forall (s, a, h) \in \mathcal{S}_{\text{one}} \times \mathcal{A}_{\text{one}} \times [H], \qquad \frac{\overline{\varrho}(s)}{2} \leq d_h^{\mathsf{n}, P^{\phi,f}}(s) \leq 2\overline{\varrho}(s), \quad \frac{\overline{\varrho}(s)}{4} \leq d_h^{\mathsf{n}, P^{\phi,f}}(s, a) \leq \overline{\varrho}(s). \tag{105b}$$

In addition, we choose the following initial state distribution

$$\varrho(s) = \begin{cases} \frac{1}{CSA}, & \text{if } s \in \mathcal{M} \\ 0, & \text{if } s \in \mathcal{N}. \end{cases} \tag{106}$$

With this choice of $\varrho$, the single-policy clipped concentrability coefficient $C_r^\star$ and the quantity $C$ are intimately connected:

$$C \leq C_r^\star \leq 2C. \tag{107}$$

The proofs of (105) and (107) are postponed to Appendix H.2 and H.3, respectively.

### E.2   Step 2: Establishing the minimax lower bound

Recall our goal: for any policy estimator $\widetilde{\mu}$ computed based on the empirical dataset, we plan to control the quantity

$$\max_{(f,\phi)\in\mathcal{F}\times\Phi}\left\{V_1^{\star,\sigma^+,f,\phi}(\varrho) - V_1^{\widetilde{\mu},\sigma^+,f,\phi}(\varrho)\right\} \tag{108}$$

with initial state distribution defined in (106).

#### E.2.1   Step 1: Converting the goal to estimate $(f, \phi)$

As verified in Appendix H.4, we have

$$\varepsilon \leq \begin{cases} \frac{c_2}{H}, & \text{if } \sigma^+ \leq \frac{c_2}{2H}; \\ 1, & \text{otherwise,} \end{cases} \tag{109}$$

and

$$\Delta = c_5 \begin{cases} \frac{\varepsilon}{H^2}, & \text{if } \sigma^+ \leq \frac{c_2}{2H}; \\ \frac{\sigma^+\varepsilon}{H}, & \text{otherwise,} \end{cases} \tag{110}$$

which satisfies (100) and leads to that for any policy $\widetilde{\mu}$ obeying

$$\sum_{h=1}^{H} \left\| \widetilde{\mu}_h(\cdot \mid m_f) - \widetilde{\mu}_h^{\star,f,\phi}(\cdot \mid m_f) \right\|_1 \geq \frac{H}{8}, \tag{111}$$

one has

$$V_1^{\star,\sigma^+,f,\phi}(m_f) - V_1^{\widetilde{\mu},\sigma^+,f,\phi}(m_f) > \varepsilon, \tag{112}$$

whose proof is postponed to Appendix H.4. Now, we are ready to convert the estimation of an optimal policy to estimating $(f, \phi)$. Let $\mathbb{P}_{f,\phi}$ represent the probability distribution when the RMDP is $\mathcal{M}_f^\phi$, $\forall (f, \phi) \in \mathcal{F} \times \Phi$. For any $(f, \phi) \in \mathcal{F} \times \Phi$, suppose that there exists a policy $\widetilde{\mu}$ achieving

$$\mathbb{P}_{f,\phi} \left\{ V_1^{\star,\sigma^+,f,\phi}(m_f) - V_1^{\widetilde{\mu},\sigma^+,f,\phi}(m_f) \leq \varepsilon \right\} \geq \frac{3}{4}, \tag{113}$$

which, in view of (112), indicates that

$$\mathbb{P}_{f,\phi} \left\{ \sum_{h=1}^{H} \left\| \widetilde{\mu}_h(\cdot \mid m_f) - \widetilde{\mu}_h^{\star,f,\phi}(\cdot \mid m_f) \right\|_1 < \frac{H}{8} \right\} \geq \frac{3}{4}. \tag{114}$$

Consequently, taking $\widetilde{\phi} = \arg\min_{\phi \in \Phi} \sum_{h=1}^{H} \left\| \widetilde{\mu}_h(\cdot \mid m_f) - \widetilde{\mu}_h^{\star,f,\phi}(\cdot \mid m_f) \right\|_1$, we construct the estimate of $\phi$ as $\widehat{\phi} = \widetilde{\phi}$. If $\sum_{h=1}^{H} \left\| \widetilde{\mu}_h(\cdot \mid m_f) - \widetilde{\mu}_h^{\star,f,\phi}(\cdot \mid m_f) \right\|_1 < \frac{H}{8}$ for some $\phi \in \Phi$, then for any $\phi' \in \Phi$ obeying $\phi' \neq \phi$, one has

$$\begin{aligned}
&\sum_{h=1}^{H} \left\| \widetilde{\mu}_h(\cdot \mid m_f) - \widetilde{\mu}_h^{\star,f,\phi'}(\cdot \mid m_f) \right\|_1 \\
&\geq \sum_{h=1}^{H} \left\| \widetilde{\mu}_h^{\star,f,\phi}(\cdot \mid m_f) - \widetilde{\mu}_h^{\star,f,\phi'}(\cdot \mid m_f) \right\|_1 - \sum_{h=1}^{H} \left\| \widetilde{\mu}_h(\cdot \mid m_f) - \widetilde{\mu}_h^{\star,f,\phi}(\cdot \mid m_f) \right\|_1 \\
&> \frac{H}{4} - \frac{H}{8} = \frac{H}{8}, \tag{115}
\end{aligned}$$

where the first inequality holds due to the triangle inequality, and the last inequality follows from the assumption $\sum_{h=1}^{H} \left\| \widetilde{\mu}_h(\cdot \mid m_f) - \widetilde{\mu}_h^{\star,f,\phi}(\cdot \mid m_f) \right\|_1 < \frac{H}{8}$ and the separation property of $\phi \in \Phi$; see (84). Similarly, we have $\widehat{\phi} = \phi$ if

$$\sum_{h=1}^{H} \left\| \widetilde{\mu}_h(\cdot \mid m_f) - \widetilde{\mu}_h^{\star,f,\phi}(\cdot \mid m_f) \right\|_1 < \frac{H}{8} < \sum_{h=1}^{H} \left\| \widetilde{\mu}_h(\cdot \mid m_f) - \widetilde{\mu}_h^{\star,f,\phi'}(\cdot \mid m_f) \right\|_1, \forall \phi' \in \Phi, \phi' \neq \phi, \tag{116}$$

which can be directly achieved when $\sum_{h=1}^{H} \left\| \widetilde{\mu}_h(\cdot \mid m_f) - \widetilde{\mu}_h^{\star,f,\phi}(\cdot \mid m_f) \right\|_1 < \frac{H}{8}$, and further leads to

$$\mathbb{P}_{f,\phi} \left[ \widehat{\phi} = \phi \right] \geq \mathbb{P}_{f,\phi} \left\{ \sum_{h=1}^{H} \left\| \widetilde{\mu}_h(\cdot \mid m_f) - \widetilde{\mu}_h^{\star,f,\phi}(\cdot \mid m_f) \right\|_1 < \frac{H}{8} \right\} \geq \frac{3}{4}. \tag{117}$$

### E.2.2 Step 2: Developing the probability of error in testing multiple hypotheses

Next, we address the hypothesis testing problem over $\phi \in \Phi$ and derive the information-theoretic lower bound for the probability of error. Specifically, we define the minimax probability of error as:

$$p_{\mathrm{e}} := \inf_{(\widehat{f},\widehat{\phi})} \max_{(f,\phi) \in \mathcal{F} \times \Phi} \mathbb{P}_{f,\phi}(\widehat{\phi} \neq \phi),$$

where the infimum is taken over all possible tests $\widehat{\phi}$ constructed from the available batch dataset.

Given the dataset $\mathcal{D}_0$ with $K$ independent trajectories, let $\varrho^{\mathrm{n},\phi}$ (and $\varrho_h^{\mathrm{n},\phi}(s,a)$) represent the distribution vector (and distribution) of each sample tuple $(s_h, a_h, s_h')$ at time step $h$ under the nominal

transition kernel $P^\star$ for $\mathcal{M}_f^{\phi,\mathsf{n}}$. Then, employing Fano's inequality [36, Theorem 2.2] and the additivity of KL divergence [36, Page 85], it follows that

$$
\begin{aligned}
p_{\mathrm{e}} &\geq 1 - K\frac{\max_{(\phi,\widetilde{\phi})\in\Phi,\phi\neq\widetilde{\phi}}\mathsf{KL}\big(\varrho^{\mathsf{n},\phi}\,|\,\varrho^{\mathsf{n},\widetilde{\phi}}\big) + \log 2}{\log|\Phi|} \\
&\overset{(i)}{\geq} 1 - \frac{8K}{H}\max_{(\phi,\widetilde{\phi})\in\Phi,\phi\neq\widetilde{\phi}}\mathsf{KL}\big(\varrho^{\mathsf{n},\phi}\,|\,\varrho^{\mathsf{n},\widetilde{\phi}}\big) - \frac{8\log 2}{H} \\
&\overset{(ii)}{\geq} \frac{1}{2} - \frac{8K}{H}\max_{(\phi,\widetilde{\phi})\in\Phi,\phi\neq\widetilde{\phi}}\mathsf{KL}\big(\varrho^{\mathsf{n},\phi}\,|\,\varrho^{\mathsf{n},\widetilde{\phi}}\big),
\end{aligned}
\tag{118}
$$

where (i) holds due to $|\Phi| \geq e^{H/8}$ and (ii) follows from $H \geq 16\log 2$.

Since the occupancy state distribution $d_h^{\mathsf{n}}$ is the same for any MDP $\mathcal{M}_f^\phi$, $\forall \phi \in \Phi$, we apply the chain rule of KL divergence [12, Lemma 5.2.8] and the Markov property of the independent sample trajectories to obtain:

$$
\begin{aligned}
\mathsf{KL}\big(\varrho^{\mathsf{n},\phi}\,|\,\varrho^{\mathsf{n},\widetilde{\phi}}\big) &= \sum_{h=1}^{H}\mathop{\mathbb{E}}_{s\sim d_h^{\mathsf{n}}(s)}\Big[\mathsf{KL}\big(P_h^{\star,\phi}(\cdot\,|\,s,a)\,\|\,P_h^{\star,\widetilde{\phi}}(\cdot\,|\,s,a)\big)\Big] \\
&\overset{(i)}{=} \frac{1}{2}\overline{\varrho}(m_f)\sum_{h=1}^{H}\sum_{a\in\{0,1\}}\Big[\mathsf{KL}\big(P_h^{\phi}(\cdot\,|\,m_f,a)\,\|\,P_h^{\widetilde{\phi}}(\cdot\,|\,m_f,a)\big)\Big],
\end{aligned}
\tag{119}
$$

where (i) follows from applying (105) and obtaining

$$
\begin{aligned}
&\mathop{\mathbb{E}}_{s\sim d_h^{\mathsf{n}}(s)}\Big[\mathsf{KL}\big(P_h^{\star,\phi}(\cdot\,|\,s,a)\,\|\,P_h^{\star,\widetilde{\phi}}(\cdot\,|\,s,a)\big)\Big] \\
&= \sum_s d_h^{\mathsf{n}}(s)\left\{\sum_{a,s'}\widetilde{\mu}_h^{\mathsf{n}}(a\,|\,s)P_h^{\phi_h}(s'\,|\,s,a)\log\frac{\widetilde{\mu}_h^{\mathsf{n}}(a\,|\,s)P_h^{\phi_h}(s'\,|\,s,a)}{\widetilde{\mu}_h^{\mathsf{n}}(a\,|\,s)P_h^{\widetilde{\phi}_h}(s'\,|\,s,a)}\right\} \\
&= \frac{1}{2}\overline{\varrho}(m_f)\sum_a\sum_{s'}P_h^{\phi_h}(s'\,|\,m_f,a)\log\frac{P_h^{\phi_h}(s'\,|\,m_f,a)}{P_h^{\widetilde{\phi}_h}(s'\,|\,m_f,a)} \\
&= \frac{1}{2}\overline{\varrho}(m_f)\sum_a\mathsf{KL}\big(P_h^{\phi_h}(\cdot\,|\,m_f,a)\,\|\,P_h^{\widetilde{\phi}_h}(\cdot\,|\,m_f,a)\big).
\end{aligned}
$$

Consequently, combining (118) and (119) leads to

$$
p_{\mathrm{e}} \geq \frac{1}{2} - \frac{4K}{H}\max_{(\phi,\widetilde{\phi})\in\Phi,\phi\neq\widetilde{\phi}}\left[\overline{\varrho}(m_f)\sum_{h=1}^{H}\sum_a\mathsf{KL}\big(P_h^{\phi_h}(\cdot\,|\,m_f,a)\,\|\,P_h^{\widetilde{\phi}_h}(\cdot\,|\,m_f,a)\big)\right].
\tag{120}
$$

We proceed to analyze (120) by considering different cases of the uncertainty level $\sigma^+$.

- For $0 < \sigma^+ \leq \frac{c_2}{2H}$: If $\phi_h = \widetilde{\phi}_h$, it is obvious that

$$
\sum_{a\in\{0,1\}}\mathsf{KL}\big(P_h^{\star,\phi}(\cdot\,|\,s,a)\,\|\,P_h^{\star,\widetilde{\phi}}(\cdot\,|\,s,a)\big) = 0.
\tag{121}
$$

Consider the case of $\phi_h \neq \widetilde{\phi}_h$. Without loss of generality, we suppose $\phi_h = 0$ and $\widetilde{\phi}_h = 1$, which indicates

$$
\begin{aligned}
\mathsf{KL}\big(P_h^{\star,\phi}(0\,|\,m_f,0)\,\|\,P_h^{\star,\widetilde{\phi}}(0\,|\,m_f,0)\big) &\leq \frac{(p-q)^2}{q(1-q)} \overset{(i)}{=} \frac{\Delta^2}{q(1-q)} \\
&\overset{(ii)}{=} \frac{(c_5)^2\varepsilon^2}{H^4 q(1-q)} \leq \frac{4(c_5)^2\varepsilon^2}{c_2 H^3},
\end{aligned}
\tag{122}
$$

where the first inequality exists by applying Lemma E.1, (i) follows from the definitions in (99), (ii) holds due to the definition in (110), and the last inequality arises from $q = p - \Delta \geq$

$\frac{c_2}{2H}$ (see (100)) and $1 - q \geq 1 - p \geq 1 - \frac{c_2}{H} \geq \frac{1}{2}$. Similarly, we establish the same bound for $\mathsf{KL}\big(P_h^{\star,\phi}(0 \,|\, m_f, 1) \,\|\, P_h^{\star,\widetilde{\phi}}(0 \,|\, m_f, 1)\big)$. Further incorporating (122) yields

$$\sum_{a \in \{0,1\}} \mathsf{KL}\big(P_h^{\star,\phi}(\cdot \,|\, m_f, a) \,\|\, P_h^{\star,\widetilde{\phi}}(\cdot \,|\, m_f, a)\big) \leq \frac{16(c_5)^2 \varepsilon^2}{c_2 H^3}. \tag{123}$$

- For $\frac{c_2}{2H} < \sigma^+ \leq 1 - c_0$: Following the same pipeline, it then boils down to control the main term as below:

$$\mathsf{KL}\big(P_h^{\star,\phi}(0 \,|\, m_f, 0) \,\|\, P_h^{\star,\widetilde{\phi}}(0 \,|\, m_f, 0)\big) \leq \frac{(p-q)^2}{q(1-q)} \overset{\text{(i)}}{=} \frac{\Delta^2}{q(1-q)}$$

$$\overset{\text{(ii)}}{=} \frac{(c_5)^2 \sigma^{+2} \varepsilon^2}{H^2 q(1-q)} \leq \frac{2(c_5)^2 \sigma^+ \varepsilon^2}{c_0 H^2}, \tag{124}$$

where (i) and (ii) follow from the definitions in (99) and (110). Here, the last inequality arises from

$$1 - q \geq 1 - p = 1 - (1 + \frac{c_1}{H})\sigma^+ \overset{\text{(i)}}{\geq} c_0 - \frac{c_1}{H} \overset{\text{(ii)}}{\geq} \frac{c_0}{2}$$

$$p \geq q = p - \Delta \overset{\text{(iii)}}{\geq} \sigma^+, \tag{125}$$

where (ii) holds due to the definition of $c_1 = \frac{c_0}{2}$, and (iii) follows from (101). Consequently, we arrive at

$$\sum_{a \in \{0,1\}} \mathsf{KL}\big(P_h^{\star,\phi}(\cdot \,|\, s, a) \,\|\, P_h^{\star,\widetilde{\phi}}(\cdot \,|\, s, a)\big) \leq \frac{8(c_5)^2 \sigma^+ \varepsilon^2}{c_0 H^2}. \tag{126}$$

Summing up (123) and (126), we achieve for any $(\phi, \widetilde{\phi}) \in \Phi$ with $\phi \neq \widetilde{\phi}$ and any time step $h \in [H]$

$$\sum_{a \in \{0,1\}} \mathsf{KL}\big(P_h^{\star,\phi}(\cdot \,|\, m_f, a) \,\|\, P_h^{\star,\widetilde{\phi}}(\cdot \,|\, m_f, a)\big) \leq \frac{16(c_5)^2 \varepsilon^2}{c_0 c_2 H^2} \max\{\sigma^+, 1/H\}. \tag{127}$$

Plugging (127) back to (120), under the definition in (106), we obtain

$$p_{\mathrm{e}} \geq \frac{1}{2} - \frac{4K}{H} \max_{(\phi, \widetilde{\phi}) \in \Phi, \phi \neq \widetilde{\phi}} \left[ \overline{\varrho}(m_f) \sum_{h=1}^{H} \sum_{a} \mathsf{KL}\big(P_h^{\phi_h}(\cdot \,|\, m_f, a) \,\|\, P_h^{\widetilde{\phi}_h}(\cdot \,|\, m_f, a)\big) \right]$$

$$\geq \frac{1}{2} - \frac{4K}{H} \overline{\varrho}(m_f) \sum_{h=1}^{H} \frac{16(c_5)^2 \varepsilon^2}{c_0 c_2 H^2} \max\{\sigma^+, 1/H\}$$

$$\geq \frac{1}{2} - \frac{64K(c_5)^2 \varepsilon^2}{c_0 c_2 CSAH^2} \max\{\sigma^+, 1/H\} \geq \frac{1}{4}, \tag{128}$$

as long as the sample size, $T = KH$, of the dataset is selected as

$$T \leq \frac{c_0 c_2 CSAH^3 \min\{1/\sigma^+, H\}}{256(c_5)^2 \varepsilon^2} \leq \frac{c_0 c_2 C_{\mathrm{r}}^\star SAH^3 \min\{1/\sigma^+, H\}}{256(c_5)^2 \varepsilon^2}. \tag{129}$$

### E.2.3  Step 3: Putting all together

Next, we establish (108) by contradiction. Suppose that there exists an estimator $\widetilde{\mu}$ such that

$$\max_{(f,\phi \in \mathcal{F}) \times \Phi} \mathbb{P}_{f,\phi} \left[ \left\{ V_1^{\star,\sigma^+,f,\phi}(\varrho) - V_1^{\widetilde{\mu},\sigma^+,f,\phi}(\varrho) \right\} \geq \varepsilon \right] < \frac{1}{4}. \tag{130}$$

According to (108), we would need

$$\forall w \in \mathcal{F}: \quad \max_{\phi \in \Phi} \mathbb{P}_{f,\phi} \left[ \left\{ V_1^{\star,\sigma^+,f,\phi}(m_f) - V_1^{\widetilde{\mu},\sigma^+,f,\phi}(m_f) \right\} \geq \varepsilon \right] < \frac{1}{4}. \tag{131}$$

To meet (131) for any $w \in \mathcal{F}$, we would require

$$\forall \phi \in \Phi : \mathbb{P}_{f,\phi} \left\{ V_1^{\star,\sigma^+,f,\phi}(m_f) - V_1^{\widetilde{\mu},\sigma^+,f,\phi}(m_f) < \varepsilon \right\} \geq \frac{3}{4}, \tag{132}$$

which, in view of (112), indicates that we would need

$$\forall \phi \in \Phi : \quad \mathbb{P}_{f,\phi} \left\{ \sum_{h=1}^{H} \left\| \widetilde{\mu}_h(\cdot \mid m_f) - \widetilde{\mu}_h^{\star,f,\phi}(\cdot \mid m_f) \right\|_1 < \frac{H}{8} \right\} \geq \frac{3}{4}. \tag{133}$$

On the other hand, (117) indicates

$$\forall \phi \in \Phi : \mathbb{P}_{f,\phi} \left[ \widehat{\phi} = \phi \right] \geq \frac{3}{4}. \tag{134}$$

To achieve (130) or, in other words, (133), we apply (134) to all $w \in \mathcal{F}$, which would require

$$\forall (f, \phi) \in \mathcal{F} \times \Phi : \quad \mathbb{P}_{f,\phi} \left[ (\widehat{f}, \widehat{\phi}) = (f, \phi) \right] \geq \frac{3}{4}. \tag{135}$$

This contract with (128) as long as the sample size condition in (129) is satisfied. Thus, if the sample size obeys (129), we cannot achieve an estimate $\widetilde{\mu}$ that satisfies (130), which completes the proof.

# F    Multiplayer General-sum Markov Games

In this section, we extend RTZ-VI-LCB to the setting of multi-player general-sum Markov games and present the corresponding theoretical guarantees.

## F.1    Problem formulation

A robust general-sum Markov game is a tuple $\mathcal{M}(\mathcal{S}, \{\mathcal{A}_i\}_{i=1}^m, H, \{\mathcal{U}_\rho^{\sigma_i}(P^0)\}_{i=1}^m, \{r_i\}_{i=1}^m)$ with $m$ players, where $\mathcal{S}$ denotes the state space and $H$ is the horizon length. We have $m$ different action spaces, where $\mathcal{A}_i$ is the action space for the $i^{\text{th}}$ player and $|\mathcal{A}_i| = A_i$. We let $\mathcal{A} = \mathcal{A}_1 \times \cdots \times \mathcal{A}_m$ denote the joint action space, and let $\boldsymbol{a} := (a_1, \cdots, a_m) \in \mathcal{A}$ denote the joint actions by all $m$ players. A notable deviation from standard MGs is that: for $1 \leq i \leq m$, instead of assuming a fixed transition kernel, each $i^{\text{th}}$ player anticipates that the transition kernel is allowed to be chosen arbitrarily from a prescribed uncertainty set $\mathcal{U}_\rho^{\sigma_i}(P^0)$. Here, the uncertainty set $\mathcal{U}_\rho^{\sigma_i}(P^0)$ is constructed centered on $P^0(\cdot | s, \boldsymbol{a})$, with its size and shape defined by a certain distance metric $\rho$ and a radius parameter $\sigma_i > 0$. $r_i = \{r_{h,i}\}_{h \in [H]}$ is a collection of reward functions for the $i^{\text{th}}$ player, so that $r_{h,i}(s, \boldsymbol{a})$ gives the reward received by the $i^{\text{th}}$ player if actions $\boldsymbol{a}$ are taken at state $s$ at step $h$.

The policy of the $i^{\text{th}}$ player is denoted as $\pi_i := \left\{ \pi_{h,i} : \mathcal{S} \to \Delta_{\mathcal{A}_i} \right\}_{h \in [H]}$. We denote the product policy of all players as $\pi := \pi_1 \times \cdots \times \pi_M$, and denote the policy of all players except the $i^{\text{th}}$ player as $\pi_{-i}$. We define $V_{h,i}^\pi(s)$ as the expected cumulative reward that will be received by the $i^{\text{th}}$ player if starting at state $s$ at step $h$ and all players follow policy $\pi$. For any strategy $\pi_{-i}$, there also exists a *robust best response* of the $i^{\text{th}}$ player, which is a policy $\mu^\star(\pi_{-i})$ satisfying $V_{h,i}^{\mu^\star(\pi_{-i}),\pi_{-i},\sigma_i}(s) = \sup_{\pi_i} V_{h,i}^{\pi_i,\pi_{-i},\sigma_i}(s)$ for any $(s, h) \in \mathcal{S} \times [H]$. For convenience, we denote $V_{h,i}^{\star,\pi_{-i},\sigma_i} := V_{h,i}^{\mu^\star(\pi_{-i}),\pi_{-i},\sigma_i}$. The $Q$-functions of the robust best response can be defined similarly.

Similar to the definition of behavior policy $(\mu^{\mathsf{n}}, \nu^{\mathsf{n}})$, we use the short-hand notation for the occupancy distribution w.r.t. the behavior policy $\pi^{\mathsf{n}} = (\pi_i^{\mathsf{n}}, \pi_{-i}^{\mathsf{n}})$ as: $\forall (h, s, \boldsymbol{a}) \in [H] \times \mathcal{S} \times \mathcal{A}$,

$$d_h^{\mathsf{n},P^0}(s) = d_h^{\pi^{\mathsf{n}},P^0}(s) := \mathbb{P}(s_h = s \mid s_1 \sim \varrho^{\mathsf{n}}, \pi^{\mathsf{n}}, P^0); \tag{136a}$$

$$d_h^{\mathsf{n},P^0}(s, \boldsymbol{a}) = d_h^{\pi^{\mathsf{n}},P^0}(s, \boldsymbol{a}) := \mathbb{P}(s_h = s \mid s_1 \sim \varrho^{\mathsf{n}}, \pi^{\mathsf{n}}, P^0) \, \pi^{\mathsf{n}}(\boldsymbol{a} \mid s). \tag{136b}$$

Similarly, for any product policy $\pi = (\pi_i, \pi_{-i})$, there is, $\forall (h, s, \boldsymbol{a}) \in [H] \times \mathcal{S} \times \mathcal{A}$

$$d_h^{\pi_i,\pi_{-i},P}(s) := \mathbb{P}(s_h = s \mid s_1 \sim \varrho, \pi, P); \tag{137a}$$

$$d_h^{\pi_i,\pi_{-i},P}(s, \boldsymbol{a}) := \mathbb{P}(s_h = s \mid s_1 \sim \varrho, \pi, P) \, \pi_{i,h}(a_i \mid s) \, \pi_{-i,h}(\boldsymbol{a}_{-i} \mid s). \tag{137b}$$

Therefore, the robust variant of standard solution concepts—robust NE for Robust multi-player general-sum MGs is introcuded as follows: A product policy $\pi$ is considered a *robust NE* if

$$\forall(s) \in \mathcal{S}, \quad V_1^{\pi,\sigma_i}(s) = V_h^{\star,\pi_{-i},\sigma^+}(s). \tag{138}$$

A robust NE signifies that given the product policy ($\pi$) of the opponents, no player can enhance their outcome by deviating from their current policy unilaterally when each player accounts for the worst-case scenario within their uncertainty set $\mathcal{U}_\rho^{\sigma_i}(P^0)$ for all $i = 1, 2, \cdots, m$.

Since finding exact robust equilibria can be complex and may not always be feasible, practitioners often seek approximate equilibria. In this context, a product policy $\pi \in \Delta(\mathcal{A})$ can be termed an $\varepsilon$-*robust NE* if

$$\mathrm{Gap}(\pi) := \max \left\{ \left\{ V_{i,1}^{\star,\pi_{-i},\sigma_i}(\varrho) - V_{i,1}^{\pi,\sigma_i}(\varrho) \right\}_{i=1}^m \right\} \leq \varepsilon, \tag{139}$$

where

$$V_1^{\star,\pi_{-i},\sigma_i}(\varrho) = \mathbb{E}_{s \sim \varrho} V_1^{\star,\pi_{-i},\sigma_i}(s), \qquad \text{and} \qquad V_1^{\star,\sigma_i}(\varrho) = \mathbb{E}_{s \sim \varrho} V_1^{\star,\sigma_i}(s).$$

The existence of robust NE has been established for general divergence functions used in the uncertainty set in [5].

**Goal**  With a dataset collected from the nominal environment, our objective is to find a solution among the $\varepsilon$-robust NEs for the robust multi-player general-sum MG $\mathcal{MG}_r$ w.r.t. a specified uncertainty set $\mathcal{U}_\rho^{\sigma_i}(P^0)$ around the nominal kernel, while minimizing the number of samples required under partial coverage of the state-action space.

### F.2  Multi-RTZ-VI-LCB

Here we present the Multi-RTZ-VI-LCB algorithm in Algorithm 4, which is an extension of Algorithm 2 for multi-player general-sum Markov games.

According to the empirical frequencies of state transitions, we can naturally construct an empirical estimate $\widehat{P}^0 = \{\widehat{P}_h^0\}_{h=1}^H$ of $P^0$, where

$$\widehat{P}_h^0\left(s' \mid s, \boldsymbol{a}\right) = \begin{cases} \frac{1}{N_h(s,\boldsymbol{a})} \sum_{j=1}^N \mathbb{1}\left\{\left(s_j, \boldsymbol{a}_j, s_j'\right) = (s, \boldsymbol{a}, s')\right\}, & \text{if } N_h\left(s, \boldsymbol{a}\right) > 0; \\ \frac{1}{S}, & \text{if } N_h\left(s, \boldsymbol{a}\right) = 0, \end{cases} \tag{140}$$

$$\widehat{r}_{i,h}\left(s, \boldsymbol{a}\right) = \begin{cases} r_{i,h}\left(s, \boldsymbol{a}\right), & \text{if } N_h\left(s, \boldsymbol{a}\right) > 0; \\ 0, & \text{if } N_h\left(s, \boldsymbol{a}\right) = 0, \end{cases} \tag{141}$$

for any $(i, h, s, \boldsymbol{a}, s') \in [m] \times [H] \times \mathcal{S} \times \mathcal{A} \times \mathcal{B} \times \mathcal{S}$. Besides, $N_h(s, \boldsymbol{a})$ represents the total number of sample transitions from $(s, \boldsymbol{a})$ at step $h$, and

$$N_h(s, \boldsymbol{a}) := \sum_{j=1}^N \mathbb{1}\left\{(s_j, \boldsymbol{a}_j) = (s, \boldsymbol{a})\right\}. \tag{142}$$

Before the details of Multi-RTZ-VI-LCB, we extend Algorithm 1 as Algorithm 3, which reduces statistical dependencies and produces a distributionally equivalent dataset $\mathcal{D}_0$ with independent samples. Similar to Lemma 3.1, we present the following lemma concerning the dataset $\mathcal{D}_0$, whose proof is similar to the context in Appendix C.

**Lemma F.1.** *The dataset produced by the two-stage subsampling method is distributionally identical to $\mathcal{D}_0$ with probability at least $1 - 8\delta$, where $\{N_h(s, \boldsymbol{a})\}$ are independent of the sample transitions in $\mathcal{D}^0$ and obey:* $\forall(h, s, \boldsymbol{a}) \in [H] \times \mathcal{S} \times \mathcal{A},$

$$N_h(s, \boldsymbol{a}) \geq \frac{K d_h^{\mathsf{n}}(s, \boldsymbol{a})}{8} - 5\sqrt{K d_h^{\mathsf{n}}(s, \boldsymbol{a}) \log \frac{KH}{\delta}}. \tag{143}$$

Based on Algorithm 4, we propose a model-based approach for solving robust multi-player general-sum MGs using an approximate $\widehat{P}^0$ for $P^0$, as summarized in Algorithm 4.

---

**Algorithm 3** Two-stage subsampling for Multi-RTZ-VI-LCB.

---

**input** Dataset $\mathcal{D}$, probability $\delta$.

1: **Step 1: Data Partitioning.** Split $\mathcal{D}$ into two equal-sized subsets, $\mathcal{D}^{\mathsf{m}}$ and $\mathcal{D}^{\mathsf{a}}$, each containing $K/2$ trajectories.

2: **Step 2: Defining Transition Bounds.** For step $h$ and state $s$, denote the number of transitions from $\mathcal{D}^{\mathsf{m}}$ (resp. $\mathcal{D}^{\mathsf{a}}$) as $N_h^{\mathsf{m}}(s)$ (resp. $N_h^{\mathsf{a}}(s)$). Construct the trimmed count as:

$$N_h^{\mathsf{t}}(s) := \max\left\{ N_h^{\mathsf{a}}(s) - 10\sqrt{N_h^{\mathsf{a}}(s)\log\frac{HS}{\delta}}, \; 0 \right\}.$$

3: **Step 3: Generating Subsampled Dataset.** Randomly sample transitions (quadruples of the form $(s, \boldsymbol{a}, h, s')$) from $\mathcal{D}^{\mathsf{m}}$ uniformly. For each $(s, h) \in \mathcal{S} \times [H]$, include $\min\{N_h^{\mathsf{t}}(s), N_h^{\mathsf{m}}(s)\}$ transitions in the new dataset $\mathcal{D}^{\mathsf{t}}$.

**output** Set $\mathcal{D}_0 = \mathcal{D}^{\mathsf{t}}$.

---

---

**Algorithm 4** Multi-RTZ-VI-LCB.

---

1: **Initialization**: Set uncertainty levels $\sigma_i$ for $i = 1, 2, \cdots, m$; set $\widehat{V}_{i,h}^{\sigma_i}(s) = H$ and $\widehat{Q}_{i,h}^{\sigma_i}(s, \boldsymbol{a}) = H$ for all $(i, s, \boldsymbol{a}, h) \in [m] \times \mathcal{S} \times \mathcal{A} \times [H+1]$.

2: **Compute** the empirical reward function $\widehat{r}$ using (141) and the empirical transition kernel $\widehat{P}_0$ using (140).

3: **for** $h = H, H-1, \ldots, 1$ **do**

4:     **Update** the robust Q-value estimate as

$$\widehat{Q}_{i,h}^{\sigma_i}(s, \boldsymbol{a}) = \min\left\{ \widehat{r}_{i,h}(s, \boldsymbol{a}) + \inf_{P \in \mathcal{U}^{\sigma_i}\left(\widehat{P}_{h,s,\boldsymbol{a}}^0\right)} P\widehat{V}_{i,h+1}^{\sigma_i} + \beta_{i,h}\left(s, \boldsymbol{a}, \widehat{V}_{i,h+1}^{\sigma_i}\right), \; H \right\},$$

    with $\beta_{i,h}(s, \boldsymbol{a}, V) = \min\left\{ \max\left\{ \sqrt{\frac{C_{\mathsf{n}}\log\frac{KH}{\delta}}{N_h(s,\boldsymbol{a})}\mathsf{Var}_{\widehat{P}_{h,s,\boldsymbol{a}}^0}(V)}, \frac{2C_{\mathsf{n}}H\log\frac{KH}{\delta}}{N_h(s,\boldsymbol{a})} \right\}, \; H \right\}.$

5:     **Compute** Nash policy for each $s \in \mathcal{S}$ as

$$\pi_h(s) = (\pi_{i,h}(s), \pi_{-i,h}(s)) = \mathsf{ComputNash}\left(\widehat{Q}_{i,h}^{\sigma_i}(s, \cdot)\right),$$

6:     **Update** the robust value estimate for each $s \in \mathcal{S}$ as

$$\widehat{V}_{i,h}^{\sigma_i}(s) = \mathbb{E}_{\boldsymbol{a} \sim \pi_h(s)}\left[ \widehat{Q}_{i,h}^{\sigma_i}(s, \boldsymbol{a}) \right].$$

7: **end for**

**output** The product policy $\hat{\pi}(s) = \{\pi_h(s)\}_{h=1}^{H}$ with $\pi_h(s) = \prod_{i=1}^{m} \pi_{i,h}(s)$.

---

Similar to (16), we can tackle the multi-player general-sum MGs problem as:

$$\inf_{P \in \mathcal{U}^{\sigma_i}\left(\widehat{P}_{h,s,\boldsymbol{a}}^0\right)} P\widehat{V}_{i,h+1}^{\sigma_i} = \max_{\alpha \in [\min_s \widehat{V}_{i,h+1}^{\sigma_i}, \max_s \widehat{V}_{i,h+1}^{\sigma_i}]} \left\{ \widehat{P}_{h,s,\boldsymbol{a}}^0\left[\widehat{V}_{i,h+1}^{\sigma_i}\right]_\alpha - \sigma_i\left(\alpha - \min_{s'}\left[\widehat{V}_{i,h+1}^{\sigma_i}\right]_\alpha(s')\right) \right\}.$$

(144)

where $\left[\widehat{V}_{i,h+1}^{\sigma_i}\right]_\alpha$ respectively denote the clipped versions of $\widehat{V}_{i,h+1}^{\sigma_i} \in \mathbb{R}^S$ based on some level $\alpha \geq 0$, as follows.

$$\left[\widehat{V}_{i,h+1}^{\sigma_i}\right]_\alpha(s) := \begin{cases} \widehat{V}_{i,h+1}^{\sigma_i}(s), & \text{if } \widehat{V}_{i,h+1}^{\sigma_i}(s) > \alpha; \\ \alpha. & \text{otherwise}; \end{cases}$$

(145)

### F.3 Analysis of Multi-ME-Nash-QL

In this subsection, we prove Theorem 4.5, which can separated into three steps as the proof of Theorem 4.2.

First of all, similar to Assumption 4.1, we measure the distributional discrepancy between the historical data and the target data to assess the effectiveness of the historical dataset for achieving the desired goal. We propose a novel assumption for robust multi-agent general-sum MGs as:

**Assumption F.2** (Robust multiple clipped concentrability)**.** The behavior policies of the historical dataset $\mathcal{D}$ satisfies

$$\max\left\{\left\{\sup_{(\pi_{-i},s,\boldsymbol{a},h,P)\in\Delta(\mathcal{A}_{-i})\times\mathcal{S}\times\mathcal{A}\times[H]\times\mathcal{U}^{\sigma_i}(P^0)}\frac{\min\left\{d_h^{\pi_i^\star,\pi_{-i},P}(s,\boldsymbol{a}),\frac{1}{S\sum_{i=1}^m A_i}\right\}}{d_h^{\mathsf{n},P^0}(s,\boldsymbol{a})}\right\}_{i=1}^m\right\}\leq C_{\mathrm{mr}}^\star \tag{146}$$

### F.3.1 Step 1: decoupling statistical dependency

Before bounding $\mathrm{Gap}(\widehat{\pi})$, we introduce an important lemma whose proof is similar to Lemma D.1 in Appendix G.1, quantifying the difference between $\widehat{P}$ and $P$ when projected in the direction of the value function.

**Lemma F.3.** *Instate the assumptions in Theorem 4.5. Consider any vector $V\in\mathbb{R}^S$ with $\|V\|_\infty\leq H$ for all $(i,h,s,\boldsymbol{a})\in[m]\times[H]\times\mathcal{S}\times\mathcal{A}$ satisfying $N_h(s,\boldsymbol{a})>0$. With probability at least $1-\delta$, one has*

$$\left|\inf_{P\in\mathcal{U}^{\sigma_i}(\widehat{P}_{h,s,\boldsymbol{a}}^0)}PV-\inf_{P\in\mathcal{U}^{\sigma_i}(P_{h,s,\boldsymbol{a}}^0)}PV\right|\leq C_4\sqrt{\frac{1}{N_h(s,\boldsymbol{a})}\mathsf{Var}_{\widehat{P}_{h,s,\boldsymbol{a}}^0}(V)\log\frac{KH}{\delta}}+C_4\frac{H\log\frac{KH}{\delta}}{N_h(s,\boldsymbol{a})} \tag{147}$$

*for some sufficiently large constant $C_4>0$, and*

$$\mathsf{Var}_{\widehat{P}_{h,s,\boldsymbol{a}}^0}(V)\leq 2\mathsf{Var}_{P_{h,s,\boldsymbol{a}}^0}(V)+O\left(\frac{H^2}{N_h(s,\boldsymbol{a})}\log\frac{KH}{\delta}\right). \tag{148}$$

With Lemma F.3, we can now have

$$\left|\inf_{\mathcal{P}\in\mathcal{U}^{\sigma_i}(\widehat{P}_{h,s,\boldsymbol{a}}^0)}PV-\inf_{\mathcal{P}\in\mathcal{U}^{\sigma_i}(P_{h,s,\boldsymbol{a}}^0)}PV\right|\leq\beta_h(s,\boldsymbol{a},V) \tag{149}$$

for any $(i,h,s,\boldsymbol{a})\in[m]\times[H]\times\mathcal{S}\times\mathcal{A}$ satisfying $N_h(s,\boldsymbol{a})\geq1$.

Therefore, we conclude that $\widehat{Q}_{i,h}^{\sigma_i}(s,\boldsymbol{a})$ is an optimistic estimation of $\widehat{Q}_{i,h}^{\pi,\sigma_i}(s,\boldsymbol{a})$ for any $i=1,2,\cdots,m$, which is summarized below, whose proof is similar to Lemma D.2 in Appendix G.2.

**Lemma F.4.** *With probability exceeding $1-\delta$, it holds that*

$$\widehat{Q}_{i,h}^{\sigma_i}(s,\boldsymbol{a})\geq Q_{i,h}^{\star,\widehat{\pi}_{-i},\sigma_i}(s,\boldsymbol{a})\qquad and\qquad \widehat{V}_{i,h}^{\sigma_i}(s)\geq V_{i,h}^{\star,\widehat{\pi}_{-i},\sigma_i}(s). \tag{150}$$

Besides, we introduce another key lemma highlighting the difference between robust multi-player general-sum MGs and standard multi-player general-sum MGs from the same idea of Lemma D.3, as shown below.

**Lemma F.5.** *Consider any multi-player general-sum MGs $\mathcal{MG}_{\mathsf{r}}=\left\{\mathcal{S},\{\mathcal{A}_i\}_{i=1}^m,H,\{\mathcal{U}_\rho^{\sigma_i}(P^0)\}_{i=1}^m,\{r_i\}_{i=1}^m\right\}$ and the uncertainty set $\{\mathcal{U}_\rho^{\sigma_i}(P^0)\}_{i=1}^m(\cdot)$ with TV distance. The optimistic robust value function estimate $\widehat{V}_{i,h}^{\sigma_i}$:*

$$\forall(i,h)\in[m]\times[H]:\quad\max_{s\in\mathcal{S}}\widehat{V}_{i,h}^{\sigma_i}-\min_{s\in\mathcal{S}}\widehat{V}_{i,h}^{\sigma_i}\leq\min\left\{\frac{(H+1)\left(1-(1-\sigma_i)^{H-h}\right)}{\sigma_i},H\right\}.$$

### F.3.2 Step 2: decomposing the error $\mathrm{Gap}(\widehat{\pi})$

The goal of our algorithm is to output an $\varepsilon$-robust NE policy $(\widehat{\pi})$ satisfying $\mathrm{Gap}(\widehat{\pi})$ in (139), i.e.,

$$\mathrm{Gap}(\widehat{\pi}):=\max\left\{\left\{V_{i,1}^{\star,\widehat{\pi}_{-i},\sigma_i}(\varrho)-V_{i,1}^{\widehat{\pi},\sigma_i}(\varrho)\right\}_{i=1}^m\right\}\leq\varepsilon.$$

According to the relationship in Lemma F.4, under the definition of $\mathcal{A}_{-i} := \mathcal{A}_1 \times \cdots \times \mathcal{A}_{i-1} \times \mathcal{A}_{i+1} \times \cdots \times \mathcal{A}_m$, we obtain

$$V_h^{\star,\widehat{\pi}_{-i,h},\sigma^+}(s) \leq \widehat{V}_{i,h}^{\sigma_i}(s) = \min_{\max_{\pi_{-i} \in \Delta(\mathcal{A}_{-i})}} \mathbb{E}_{\boldsymbol{a} \sim (\pi_i(s), \pi_{-i}(s))} \left[ Q_{i,h}^{\sigma_i}(s, \boldsymbol{a}) \right], \qquad (151)$$

where the first equality comes from line 8 in Algorithm 4. Therefore, there exists a deterministic policy $\pi_{-i}^{\mathsf{d}} : \mathcal{S} \leftarrow \Delta(\mathcal{A}_{-i})$ satisfying that for any $s \in \mathcal{S}$

$$\pi_{-i}^{\mathsf{d}}(s) \geq \arg \min_{\pi_{-i} \in \Delta(\mathcal{A}_i)} \mathbb{E}_{\boldsymbol{a} \sim (\pi_i(s), \pi_{-i}(s))} \left[ Q_{i,h}^{\sigma_i}(s, \boldsymbol{a}) \right]. \qquad (152)$$

Before starting, we introduce several useful notations:

- The state-action space covered by the behavior policy $\pi^{\mathsf{n}}$ in the nominal transition kernel $P^0$ is denoted as

$$\mathcal{C}^{\mathsf{n}} = \{(h, s, \boldsymbol{a}) : d_h^{\mathsf{n}}(s, \boldsymbol{a}) > 0\}. \qquad (153)$$

- For any $(i, h) \in [m] \times [H]$, the set of potential state occupancy distributions w.r.t. the policy $(\pi_i(s), \pi_{-i}^{\mathsf{b}}(s))$ and the uncertainty set $P \in \mathcal{U}^{\sigma_i}(P^0)$ f is denoted as

$$\mathcal{D}_{i,h}^{\mathsf{pi}} := \left\{ \left[ d_h^{\pi_i(s), \pi_{-i}^{\mathsf{b}}(s), P}(s) \right]_{s \in \mathcal{S}} : P \in \mathcal{U}^{\sigma_i}(P^0) \right\}; \qquad (154)$$

$$\mathcal{D}_{i,h}^{\mathsf{pai}} := \left\{ \left[ d_h^{\pi_i(s), \pi_{-i}^{\mathsf{b}}(s), P}(s, \boldsymbol{a}) \right]_{(s,\boldsymbol{a}) \in \mathcal{S} \times \mathcal{A}} : P \in \mathcal{U}^{\sigma_i}(P^0) \right\}. \qquad (155)$$

- For convenience and without ambiguity, we introduce an additional notation for $(i, h) \in [m] \times [H]$ as

$$\beta_{i,h}^{\pi_i, \pi_{-i}^{\mathsf{b}}}(s) = \mathbb{E}_{\boldsymbol{a} \sim (\pi_i(s), \pi_{-i}^{\mathsf{b}}(s))} \beta_{i,h}\left(s, \boldsymbol{a}, \widehat{V}_{i,h+1}^{\sigma_i}\right).$$

  In particular, the vector $\beta_{i,h}^{\pi_i, \pi_{-i}^{\mathsf{b}}} \in \mathbb{R}^S$ is defined with its $s$-th item given by $\beta_{i,h}^{\pi_i, \pi_{-i}^{\mathsf{b}}}(s)$.

- Similarly, we can define the notation related to rewards for $(i, h) \in [m] \times [H]$ as

$$\widehat{r}_{i,h}^{\pi_i, \pi_{-i}^{\mathsf{b}}}(s) = \mathbb{E}_{\boldsymbol{a} \sim (\pi_i(s), \pi_{-i}^{\mathsf{b}}(s))} \widehat{r}_{i,h}(s, \boldsymbol{a}).$$

To proceed with the analysis, we need to introduce a pessimistic V-estimation $\underline{V}_{i,h}^{\sigma_i}(s)$ that are defined similarly as $\widehat{V}_{i,h}^{\sigma_i}(s)$. Similar to Lemma F.4, we have $\underline{V}_{i,h}^{\sigma_i}(s) \leq V_{i,h}^{\widehat{\pi},\sigma_i}(s)$.

Therefore, according to the update rule in line 4 in Algorithm 4 and robust Bellman equality similar to (28), we derive

$$V_{i,h}^{\star,\widehat{\pi}_{-i},\sigma^+}(s) - V_{i,h}^{\widehat{\pi},\sigma^+}(s)$$

$$\leq \widehat{V}_{i,h}^{\sigma_i}(s) - \underline{V}_{i,h}^{\sigma_i}(s)(s)$$

$$\leq \mathbb{E}_{\boldsymbol{a} \sim (\pi_i(s), \pi_{-i}^{\mathsf{b}}(s))} \inf_{P \in \mathcal{U}^{\sigma_i}\left(\widehat{P}_{h,s,\boldsymbol{a}}^0\right)} P\widehat{V}_{i,h+1}^{\sigma_i} + 2\beta_{i,h}^{\pi_i,\pi_{-i}^{\mathsf{b}}}(s) - \mathbb{E}_{\boldsymbol{a} \sim (\pi_i(s), \pi_{-i}^{\mathsf{b}}(s))} \inf_{P \in \mathcal{U}^{\sigma_i}\left(P_{h,s,\boldsymbol{a}}^0\right)} P\underline{V}_{i,h+1}^{\sigma_i}$$

$$\leq \mathbb{E}_{\boldsymbol{a} \sim (\pi_i(s), \pi_{-i}^{\mathsf{b}}(s))} \left[ \inf_{P \in \mathcal{U}^{\sigma_i}\left(P_{h,s,\boldsymbol{a}}^0\right)} P\widehat{V}_{i,h+1}^{\sigma_i} - \inf_{P \in \mathcal{U}^{\sigma_i}\left(P_{h,s,\boldsymbol{a}}^0\right)} P\underline{V}_{i,h+1}^{\sigma_i} \right.$$

$$\left. + \left| \inf_{P \in \mathcal{U}^{\sigma_i}\left(P_{h,s,\boldsymbol{a}}^0\right)} P\widehat{V}_{i,h+1}^{\sigma_i} - \inf_{P \in \mathcal{U}^{\sigma_i}\left(\widehat{P}_{h,s,\boldsymbol{a}}^0\right)} P\widehat{V}_{i,h+1}^{\sigma_i} \right| \right] + 2\beta_{i,h}^{\pi_i,\pi_{-i}^{\mathsf{b}}}(s)$$

$$\overset{(i)}{\leq} \mathbb{E}_{\boldsymbol{a} \sim (\pi_i(s), \pi_{-i}^{\mathsf{b}}(s))} \left[ \inf_{P \in \mathcal{U}^{\sigma_i}\left(P_{h,s,\boldsymbol{a}}^0\right)} P\widehat{V}_{i,h+1}^{\sigma_i} - \inf_{P \in \mathcal{U}^{\sigma_i}\left(P_{h,s,\boldsymbol{a}}^0\right)} P\underline{V}_{i,h+1}^{\sigma_i} \right] + 3\beta_{i,h}^{\pi_i,\pi_{-i}^{\mathsf{b}}}(s)$$

$$\overset{(ii)}{\leq} \mathbb{E}_{\boldsymbol{a} \sim (\pi_i(s), \pi_{-i}^{\mathsf{b}}(s))} \left[ P_{i,h,s,\boldsymbol{a}}^{\inf,V}\left( \widehat{V}_{i,h+1}^{\sigma_i} - \underline{V}_{i,h+1}^{\sigma_i} \right) \right] + 3\beta_{i,h}^{\pi_i,\pi_{-i}^{\mathsf{b}}}(s). \qquad (156)$$

Here, (i) in (156) exists due to (149) in Lemma F.3 for $N_h(s, \boldsymbol{a}) > 0$ and

$$\left| \inf_{P \in \mathcal{U}^{\sigma_i}\left(P_{h,s,\boldsymbol{a}}^0\right)} P \widehat{V}_{i,h+1}^{\sigma_i} - \inf_{P \in \mathcal{U}^{\sigma_i}\left(\widehat{P}_{h,s,\boldsymbol{a}}^0\right)} P \underline{V}_{i,h+1}^{\sigma_i} \right| \leq H = \beta_{i,h}^{\pi_i, \pi_{-i}^{\mathrm{b}}}(s) \text{ for } N_h(s, \boldsymbol{a}) = 0; \quad (157)$$

and (ii) is valid under the notation

$$P_{i,h,s,\boldsymbol{a}}^{\mathrm{inf},V} := \operatorname{argmin}_{P \in \mathcal{U}^{\sigma^+}\left(P_{h,s,\boldsymbol{a}}^0\right)} P \underline{V}_{i,h+1}^{\sigma_i} \quad (158)$$

and consequently,

$$\inf_{P \in \mathcal{U}^{\sigma_i}\left(P_{h,s,\boldsymbol{a}}^0\right)} P \underline{V}_{i,h+1}^{\sigma_i} = P_{i,h,s,\boldsymbol{a}}^{\mathrm{inf},V} \underline{V}_{i,h+1}^{\sigma_i}, \text{ and } \inf_{P \in \mathcal{U}^{\sigma_i}\left(P_{h,s,\boldsymbol{a}}^0\right)} P \widehat{V}_{i,h+1}^{\sigma_i} \leq P_{i,h,s,\boldsymbol{a}}^{\mathrm{inf},V} \widehat{V}_{i,h+1}^{\sigma_i}.$$

For ease of proof, we introduce a notation as $\check{P}_{i,h,s}^{\mathrm{inf},V} := \mathbb{E}_{\boldsymbol{a} \sim (\pi_i(s), \pi_{-i}^{\mathrm{b}}(s))} P_{i,h,s,\boldsymbol{a}}^{\mathrm{inf},V}$. Furthermore, we define a sequence of matrices $\check{P}_{i,h}^{\mathrm{inf},V} \in \mathbb{R}^{S \times S}$. We can utilizing (156) recursively over the time steps $h, h+1, \cdots, H$ and derive

$$\begin{aligned}
V_{i,h}^{\star, \widehat{\pi}_{-i}, \sigma^+}(s) - V_{i,h}^{\widehat{\pi}, \sigma^+}(s) &\leq \widehat{V}_{i,h}^{\sigma_i}(s) - \underline{V}_{i,h}^{\sigma_i}(s) \\
&\leq \check{P}_{i,h}^{\mathrm{inf},V} \left( \widehat{V}_{i,h+1}^{\sigma_i} - \underline{V}_{i,h+1}^{\sigma_i} \right) + 3\beta_{i,h}^{\pi_i, \pi_{-i}^{\mathrm{b}}}(s) \\
&\leq \check{P}_{i,h}^{\mathrm{inf},V} \check{P}_{i,h+1}^{\mathrm{inf},V} \left( \widehat{V}_{i,h+2}^{\sigma_i} - \underline{V}_{i,h+2}^{\sigma_i} \right) + 3\check{P}_{i,h}^{\mathrm{inf},V} \beta_{i,h+1}^{\pi_i, \pi_{-i}^{\mathrm{b}}} + 3\beta_{i,h}^{\pi_i, \pi_{-i}^{\mathrm{b}}}(s) \\
&\leq \cdots \leq 3 \sum_{i'=h}^{H} \left( \prod_{j=h+1}^{i'-1} \check{P}_{i,j}^{\mathrm{inf},V} \right) \beta_{i,i'}^{\pi_i, \pi_{-i}^{\mathrm{b}}} + 3\beta_{i,h}^{\pi_i, \pi_{-i}^{\mathrm{b}}}(s) \\
&= 3 \sum_{i'=h}^{H} \left( \prod_{j=h}^{i'-1} \check{P}_{i,j}^{\mathrm{inf},V} \right) \beta_{i,i'}^{\pi_i, \pi_{-i}^{\mathrm{b}}}, \quad (159)
\end{aligned}$$

where we define $\left( \prod_{j=h}^{i'-1} \check{P}_{i,j}^{\mathrm{inf},V} \right) = I$ for conciseness.

For any $d_h^{\pi_i, \pi_{-i}^{\mathrm{b}}} \in \mathcal{D}_h^{\mathrm{pi}}$ (cf. (53)), taking inner product with (59) yields

$$\begin{aligned}
\left\langle d_h^{\pi_i, \pi_{-i}^{\mathrm{b}}}, V_{i,h}^{\star, \widehat{\pi}_{-i}, \sigma^+} - V_{i,h}^{\widehat{\pi}, \sigma^+} \right\rangle &\leq \left\langle d_h^{\pi_i, \pi_{-i}^{\mathrm{b}}}, 3 \sum_{i'=h}^{H} \left( \prod_{j=h}^{i'-1} \check{P}_{i,j}^{\mathrm{inf},V} \right) \beta_{i,i'}^{\pi_i, \pi_{-i}^{\mathrm{b}}} \right\rangle \\
&= 3 \sum_{i'=h}^{H} \left\langle d_{i'}^{\mathrm{p}, \pi_i, \pi_{-i}^{\mathrm{b}}}, \beta_{i,i'}^{\pi_i, \pi_{-i}^{\mathrm{b}}} \right\rangle, \quad (160)
\end{aligned}$$

where

$$d_{i'}^{\mathrm{p}, \pi_i^{\mathrm{d}}, \pi_{-i}^{\star}} := \left[ \left( d_h^{\pi_i, \pi_{-i}^{\mathrm{b}}} \right)^{\top} \left( \prod_{j=h}^{i'-1} \check{P}_{i,j}^{\mathrm{inf},V} \right) \right]^{\top} \in \mathcal{D}_{i'}^{\mathrm{pi}} \quad (161)$$

by the definition of $\mathcal{D}_{i'}^{\mathrm{pi}}$ (cf. (154)) for all $i' = h+1, \cdots, H$.

Next, we control $\langle d_{i'}^{\mathsf{p},\pi_i,\pi_{-i}^{\mathsf{b}}}, \beta_{i,i'}^{\pi_i,\pi_{-i}^{\mathsf{b}}}\rangle$ utilizing concentrability. First of all, according to the definition of penalty, we demonstrate that the pessimistic penalty satisfies

$$
\beta_{i,i'}(s,\boldsymbol{a},\hat{V}) \le \max\left\{\sqrt{\frac{C_{\mathsf{n}}\log\frac{KH}{\delta}}{N_i(s,\boldsymbol{a})}\mathsf{Var}_{\widehat{P}_{i,s,\boldsymbol{a}}^0}(\hat{V})}, \frac{2C_{\mathsf{n}}H\log\frac{KH}{\delta}}{N_i(s,\boldsymbol{a})}\right\}
$$

$$
\le \sqrt{\frac{C_{\mathsf{n}}\log\frac{KH}{\delta}}{N_i(s,\boldsymbol{a})}\mathsf{Var}_{\widehat{P}_{i,s,\boldsymbol{a}}^0}(\hat{V})} + \frac{2C_{\mathsf{n}}H\log\frac{KH}{\delta}}{N_i(s,\boldsymbol{a})}
$$

$$
\overset{\text{(i)}}{\le} \sqrt{\frac{C_{\mathsf{n}}\log\frac{KH}{\delta}}{N_i(s,\boldsymbol{a})}\left(2\mathsf{Var}_{P_{i,s,\boldsymbol{a}}^0}(\hat{V}) + \frac{C_0 H^2}{N_i(s,\boldsymbol{a})}\log\frac{KH}{\delta}\right)} + \frac{2C_{\mathsf{n}}H\log\frac{KH}{\delta}}{N_i(s,\boldsymbol{a})}
$$

$$
\overset{\text{(ii)}}{\le} \sqrt{\frac{2C_{\mathsf{n}}\log\frac{KH}{\delta}}{N_i(s,\boldsymbol{a})}\mathsf{Var}_{P_{i,s,\boldsymbol{a}}^0}(\hat{V})} + \frac{\left(2C_{\mathsf{n}} + \sqrt{C_{\mathsf{n}}C_0}\right)H\log\frac{KH}{\delta}}{N_i(s,\boldsymbol{a})} \tag{162}
$$

where (i) holds by applying (148) for some sufficiently large $C_0$ and (ii) exists follows from the Cauchy-Schwarz inequality. Therefore, combining the definition of $\beta_{i,i'}^{\pi_i,\pi_{-i}^{\mathsf{b}}}(s)$, we obtain

$$
\langle d_{i'}^{\mathsf{p},\pi_i,\pi_{-i}^{\mathsf{b}}}, \beta_{i,i'}^{\pi_i,\pi_{-i}^{\mathsf{b}}}\rangle = \sum_{s\in\mathcal{S}} d_{i'}^{\mathsf{p},\pi_i,\pi_{-i}^{\mathsf{b}}}(s)\beta_{i,i'}^{\pi_i,\pi_{-i}^{\mathsf{b}}}(s)
$$

$$
= \sum_{s\in\mathcal{S}} d_{i'}^{\mathsf{p},\pi_i,\pi_{-i}^{\mathsf{b}}}(s)\mathbb{E}_{\boldsymbol{a}\sim(\pi_i(s),\pi_{-i}^{\mathsf{b}}(s))}\beta_{i,i'}(s,\boldsymbol{a},\hat{V})
$$

$$
= \sum_{(s,\boldsymbol{a})\in\mathcal{S}\times\mathcal{A}\times\mathcal{B}} d_{i'}^{\mathsf{p},\pi_i,\pi_{-i}^{\mathsf{b}}}(s)\mathbb{1}\{a_i = \pi_i^\star(s)\}\pi_{-i}^{\mathsf{d}}(\boldsymbol{a}_{-i}|s)\beta_{i,i'}(s,\boldsymbol{a},\hat{V})
$$

$$
= \sum_{(s,a_i)\in\mathcal{S}\times\mathcal{A}} d_{i'}^{\mathsf{p},\pi_i,\pi_{-i}^{\mathsf{b}}}(s,a_i,\pi_{-i}^{\mathsf{b}}(s))\beta_{i,i'}(s,\pi_i^{\mathsf{d}}(s),\boldsymbol{a}_{-i},\hat{V}), \tag{163}
$$

where the last equation holds due to the definition in (137b). Then, we observe $d_h^{\mathsf{p},\pi_i,\pi_{-i}^{\mathsf{b}}}(s,\boldsymbol{a}) \in \mathcal{D}_h^{\mathsf{pai}}$ (cf. (155)). Thereafter, we divide the bound (163) into two cases.

**For the first case**, i.e., $s\in S$ where $\max_{P\in\mathcal{U}^{\sigma_i}(P^0)} d_{i'}^{\pi_i,\pi_{-i}^{\mathsf{b}},P}(s,a_i,\pi_{-i}^{\mathsf{b}}(s)) = 0$, it follows from the definition (cf. (154)) that for any $d_{i'}^{\mathsf{p},\pi_i,\pi_{-i}^{\mathsf{b}}}(s,a_i,\pi_{-i}^{\mathsf{b}}(s)) \in \mathcal{D}_i^{\mathsf{pai}}$, it satisfies that

$$
d_{i'}^{\mathsf{p},\pi_i,\pi_{-i}^{\mathsf{b}}}(s,a_i,\pi_{-i}^{\mathsf{b}}(s)) = 0. \tag{164}
$$

**For the second case**, i.e., $s\in S$ where $\max_{P\in\mathcal{U}^{\sigma+}(P^0)} d_{i'}^{\pi_i,\pi_{-i}^{\mathsf{b}},P}(s,a_i,\pi_{-i}^{\mathsf{b}}(s)) > 0$, by the assumption in (146)

$$
\max_{P\in\mathcal{U}^{\sigma_i}(P^0)} \frac{\min\left\{d_{i'}^{\pi_i,\pi_{-i}^{\mathsf{b}},P}(s,a_i,\pi_{-i}^{\mathsf{b}}(s)), \frac{1}{S\sum_{i=1} A_i}\right\}}{d_{i'}^{\mathsf{n}}(s,a_i,\pi_{-i}^{\mathsf{b}}(s))} \le C_{\mathsf{r}}^\star < \infty.
$$

It implies that

$$
d_{i'}^{\mathsf{n}}(s,a_i,\pi_{-i}^{\mathsf{b}}(s)) > 0 \quad\text{and}\quad \left(i',s,a_i,\pi_{-i}^{\mathsf{b}}(s)\right) \in \mathcal{C}^{\mathsf{n}}. \tag{165}
$$

Lemma F.1 tells that with probability at least $1 - 8\delta$,

$$
N_{i'}\big(s, a_i, \pi^{\mathsf{b}}_{-i}(s)\big) \geq \frac{K d^{\mathsf{n}}_{i'}\big(s, a_i, \pi^{\mathsf{b}}_{-i}(s)\big)}{8} - 5\sqrt{K d^{\mathsf{n}}_{i'}\big(s, a_i, \pi^{\mathsf{b}}_{-i}(s)\big) \log \frac{KH}{\delta}}
$$

$$
\overset{\text{(i)}}{\geq} \frac{K d^{\mathsf{n}}_{i'}\big(s, a_i, \pi^{\mathsf{b}}_{-i}(s)\big)}{16}
$$

$$
\overset{\text{(ii)}}{\geq} \frac{K \max_{P \in \mathcal{U}^{\sigma_i}(P^0)} \min\left\{ d^{\pi_i, \pi^{\mathsf{b}}_{-i}, P}_{i'}\big(s, a_i, \pi^{\mathsf{b}}_{-i}(s)\big), \frac{1}{S \sum_{i=1} A_i} \right\}}{16 C^{\star}_{\mathsf{r}}}
$$

$$
\geq \frac{K \min\left\{ d^{\mathsf{p}, \pi_i, \pi^{\mathsf{b}}_{-i}}_{i'}\big(s, a_i, \pi^{\mathsf{b}}_{-i}(s)\big), \frac{1}{S \sum_{i=1} A_i} \right\}}{16 C^{\star}_{\mathsf{r}}}, \tag{166}
$$

where (ii) comes from Assumption F.2 and (i) holds due to

$$
K d^{\mathsf{n}}_{i'}\big(s, a_i, \pi^{\mathsf{b}}_{-i}(s)\big) \geq c_0 \frac{HS \sum_{i=1} A_i}{d^{\mathsf{n}}_{\mathsf{m}}} \log \frac{KH}{\delta} f(\{\sigma_i\}^m_{i=1}, H) d^{\mathsf{n}}_{i'}\big(s, a_i, \pi^{\mathsf{b}}_{-i}(s)\big)
$$

$$
\geq c_0 HS \sum_{i=1} A_i \log \frac{KH}{\delta} f(\{\sigma_i\}^m_{i=1}, H) \geq 1600 \log \frac{KH}{\delta}, \tag{167}
$$

where $f(\{\sigma_i\}^m_{i=1}, H) = \min\left\{ \left\{ \frac{(H\sigma_i - 1 + (1-\sigma_i)^H)}{(\sigma_i)^2} \right\}^m_{i=1}, H \right\}$, the first inequality follows from condition (26), and the second inequality follows from

$$
d^{\mathsf{n}}_{\mathsf{m}} = \min_{h, s, a_i, \pi^{\mathsf{b}}_{-i}(s)} \left\{ d^{\mathsf{n}}_h(s, \pi^{\mathsf{d}}_i(s), \mathbf{a}_{-i}) : d^{\mathsf{n}}_h(s, \pi^{\mathsf{d}}_i(s), \mathbf{a}_{-i}) > 0 \right\} \leq d^{\mathsf{n}}_{i'}\big(s, a_i, \pi^{\mathsf{b}}_{-i}(s)\big). \tag{168}
$$

Combining the results in (63) and (64), we arrive at

$$
\langle d^{\mathsf{p}, \pi_i, \pi^{\mathsf{b}}_{-i}}_{i'}, \beta_{i, i'}^{\pi_i, \pi^{\mathsf{b}}_{-i}} \rangle = \sum_{(s, a_i) \in \mathcal{S} \times \mathcal{A}_i} d^{\mathsf{p}, \pi_i, \pi^{\mathsf{b}}_{-i}}_{i'}(s, a_i, \pi^{\mathsf{b}}_{-i}(s)) \beta_{i, i'}(s, a_i, \pi^{\mathsf{b}}_{-i}(s), \hat{V})
$$

$$
\leq \sum_{(s, a_i) \in \mathcal{S} \times \mathcal{A}_i} d^{\mathsf{p}, \pi_i, \pi^{\mathsf{b}}_{-i}}_{i'}(s, a_i, \pi^{\mathsf{b}}_{-i}(s)) \sqrt{\frac{2 C_{\mathsf{n}} \log \frac{KH}{\delta}}{N_i\big(s, a_i, \pi^{\mathsf{b}}_{-i}(s)\big)} \mathsf{Var}_{P^0_{i, s, a_i, \pi^{\mathsf{b}}_{-i}(s)}}\big(\hat{V}\big)}
$$

$$
+ \sum_{(s, a_i) \in \mathcal{S} \times \mathcal{A}_i} d^{\mathsf{p}, \pi_i, \pi^{\mathsf{b}}_{-i}}_{i'}(s, a_i, \pi^{\mathsf{b}}_{-i}(s)) \frac{(2 C_{\mathsf{n}} + \sqrt{C_{\mathsf{n}} C_0}) H \log \frac{KH}{\delta}}{N_i\big(s, a_i, \pi^{\mathsf{b}}_{-i}(s)\big)}
$$

$$
\leq \sum_{(s, a_i) \in \mathcal{S} \times \mathcal{A}_i} d^{\mathsf{p}, \pi_i, \pi^{\mathsf{b}}_{-i}}_{i'}(s, a_i, \pi^{\mathsf{b}}_{-i}(s)) \left( \frac{16 C^{\star}_{\mathsf{r}} (2 C_{\mathsf{n}} + \sqrt{C_{\mathsf{n}} C_0}) H \log \frac{KH}{\delta}}{K \min\left\{ d^{\mathsf{p}, \pi_i, \pi^{\mathsf{b}}_{-i}}_{i'}(s, a_i, \pi^{\mathsf{b}}_{-i}(s)), \frac{1}{S \sum_{i=1} A_i} \right\}} \right.
$$

$$
\left. + \sqrt{\frac{32 C^{\star}_{\mathsf{r}} C_{\mathsf{n}} \log \frac{KH}{\delta}}{K \min\left\{ d^{\mathsf{p}, \pi_i, \pi^{\mathsf{b}}_{-i}}_{i'}(s, a_i, \pi^{\mathsf{b}}_{-i}(s)), \frac{1}{S \sum_{i=1} A_i} \right\}} \mathsf{Var}_{P^0_{i, s, a_i, \pi^{\mathsf{b}}_{-i}(s)}}\big(\hat{V}\big)} \right). \tag{169}
$$

Similar to the proof in Appendix D.2, we are ready to bound $V^{\star, \widehat{\pi}_{-i}, \sigma^+}_{i, h}(\varrho) - V^{\widehat{\pi}, \sigma^+}_{i, h}(\varrho)$. There exists some sufficiently large constants $C_1, C_2, C_3 > 0$, and

$$
V^{\star, \widehat{\pi}_{-i}, \sigma^+}_{i, h}(\varrho) - V^{\widehat{\pi}, \sigma^+}_{i, h}(\varrho) \leq \sqrt{\frac{C^{\star}_{\mathsf{r}} C_1 H^3 S \sum_{i=1} A_i \log \frac{KH}{\delta}}{K} \min\left\{ \frac{2(H\sigma_i - 1 + (1-\sigma_i)^H)}{(\sigma_i)^2}, H \right\}}
$$

$$
+ \frac{C^{\star}_{\mathsf{r}} C_2 H^2 S \sum_{i=1} A_i \log \frac{KH}{\delta}}{K} \min\left\{ \frac{2(H\sigma_i - 1 + (1-\sigma_i)^H)}{(\sigma_i)^2}, H \right\}
$$

$$
\leq \sqrt{\frac{C^{\star}_{\mathsf{r}} C_3 H^3 S \sum_{i=1} A_i \log \frac{KH}{\delta}}{K} \min\left\{ \frac{2(H\sigma_i - 1 + (1-\sigma_i)^H)}{(\sigma_i)^2}, H \right\}}, \tag{170}
$$

where the last inequality follows from condition (26).

### F.3.3 Step 3: summing up the results

Consequently, we obtain the upper bound of $V_{i,1}^{\star,\widehat{\pi}_{-i},\sigma_i}(\varrho) - V_{i,1}^{\widehat{\pi},\sigma_i}(\varrho)$ in (170). which directly leads to

$$\mathrm{Gap}(\widehat{\pi}) \le c_1 \sqrt{\frac{C_r^\star H^2 S \sum_{i=1}^m A_i \log \frac{KH}{\delta}}{K}} \min\left\{ \left\{ \frac{2(H\sigma_i - 1 + (1-\sigma_i)^H)}{(\sigma_i)^2} \right\}_{i=1}^m, H \right\}, \quad (171)$$

for some sufficiently large $c_1$ and

$$K \ge HS \sum_{i=1} A_i \log \frac{KH}{\delta} \min\left\{ \left\{ \frac{2(H\sigma_i - 1 + (1-\sigma_i)^H)}{(\sigma_i)^2} \right\}_{i=1}^m, H \right\}.$$

## G  Proofs of lemmas for Theorem 4.2

### G.1  Proof of Lemma D.1

We prove Lemma D.1 similar to the proof of Claim 1 introduced by [43].

#### G.1.1  Proof of (46).

According to the definition in (16), for any fixed value vector $V$ independent of $\widehat{P}_{h,s,a,b}^0$, we have

$$\left| \inf_{P \in \mathcal{U}^{\sigma+}(\widehat{P}_{h,s,a,b}^0)} PV - \inf_{P \in \mathcal{U}^{\sigma+}(P_{h,s,a,b}^0)} PV \right|$$

$$= \left| \max_{\alpha \in [\min_s V(s), \max_s V(s)]} \left\{ \widehat{P}_{h,s,a,b}^0 [V]_\alpha - \sigma^+ \left( \alpha - \min_{s'} [V]_\alpha(s') \right) \right\} \right.$$

$$\left. - \max_{\alpha \in [\min_s V(s), \max_s V(s)]} \left\{ P_{h,s,a,b}^0 [V]_\alpha - \sigma^+ \left( \alpha - \min_{s'} [V]_\alpha(s') \right) \right\} \right|$$

$$\le \max_{\alpha \in [\min_s V(s), \max_s V(s)]} \left| \widehat{P}_{h,s,a,b}^0 [V]_\alpha - P_{h,s,a,b}^0 [V]_\alpha \right|$$

$$\le \max_{\alpha \in [0,H]} \left| \widehat{P}_{h,s,a,b}^0 [V]_\alpha - P_{h,s,a,b}^0 [V]_\alpha \right|, \quad (172)$$

where the last inequality exists due to the fact that the maximum operator is 1-Lipschitz. According to the definition of empirical transition kernel $\widehat{P}_{h,s,a,b}^0$, we have

$$\left( \widehat{P}_{h,s,a,b}^0 - P_{h,s,a,b}^0 \right) [V]_\alpha$$

$$= \sum_{s' \in \mathcal{S}} [V(s')]_\alpha \underbrace{\left[ \frac{\sum_{i=1}^N \mathbb{1}\{h_i = h, s_i = s, a_i = a, b_i = b, s_i' = s'\}}{N_h(s,a,b)} - P_h^0(s'\,|\,s,a,b) \right]}_{=:X_{s'}}$$

as a sum of independent random variables. Based on the relationship between $P_{h,s,a,b}^0$ and $\widehat{P}_{h,s,a,b}^0$, we verify $\mathbb{E}[X_{s'}] = 0$ and $|X_{s'}| \le H$ for all $s' \in \mathcal{S}$. With probability exceeding $1 - \delta$ and for some universal constant $C_4 > 0$, under the Bernstein inequality [37, Theorem 2.8.4], we have

$$\left( \widehat{P}_{h,s,a,b}^0 - P_{h,s,a,b}^0 \right) [V]_\alpha \le C_4 \sqrt{\frac{1}{N_h(s,a,b)} \mathsf{Var}_{P_{h,s,a,b}^0}([V]_\alpha) \log \frac{KH}{\delta}} + \frac{C_4 H \log \frac{KH}{\delta}}{N_h(s,a,b)}$$

$$\le C_4 \sqrt{\frac{1}{N_h(s,a,b)} \mathsf{Var}_{P_{h,s,a,b}^0}(V) \log \frac{KH}{\delta}} + \frac{C_4 H \log \frac{KH}{\delta}}{N_h(s,a,b)}, \quad (173)$$

where the last inequality comes from the definition of $[V]_\alpha$ in (17).

Let $\overline{V} := V - \left(P_{h,s,a,b}^0 V\right) 1$, we have

$$\text{Var}_{P_{h,s,a,b}^0}(V) = P_{h,s,a,b}^0\big(\overline{V} \circ \overline{V}\big)$$

$$= \widehat{P}_{h,s,a,b}^0\big(\overline{V} \circ \overline{V}\big) + \big(P_{h,s,a,b}^0 - \widehat{P}_{h,s,a,b}^0\big)\big(\overline{V} \circ \overline{V}\big)$$

$$= \text{Var}_{\widehat{P}_{h,s,a,b}^0}(V) + \big[\big(P_{h,s,a,b}^0 - \widehat{P}_{h,s,a,b}^0\big)V\big]^2 + \big(P_{h,s,a,b}^0 - \widehat{P}_{h,s,a,b}^0\big)\big(\overline{V} \circ \overline{V}\big),$$
(174)

where the last equation holds since

$$\widehat{P}_{h,s,a,b}^0\big(\overline{V} \circ \overline{V}\big) = \widehat{P}_{h,s,a,b}^0\left(\big[V - \big(P_{h,s,a,b}^0 V\big)1\big] \circ \big[V - \big(P_{h,s,a,b}^0 V\big)1\big]\right)$$

$$= \widehat{P}_{h,s,a,b}^0(V \circ V) - 2\big(P_{h,s,a,b}^0 V\big)\big(\widehat{P}_{h,s,a,b}^0 V\big) + \big(P_{h,s,a,b}^0 V\big)^2$$

$$= \widehat{P}_{h,s,a,b}^0\left(\big[V - \big(\widehat{P}_{h,s,a,b}^0 V\big)1\big] \circ \big[V - \big(\widehat{P}_{h,s,a,b}^0 V\big)1\big]\right) + \big(\widehat{P}_{h,s,a,b}^0 V\big)^2$$

$$\quad - 2\big(P_{h,s,a,b}^0 V\big)\big(\widehat{P}_{h,s,a,b}^0 V\big) + \big(P_{h,s,a,b}^0 V\big)^2$$

$$= \text{Var}_{\widehat{P}_{h,s,a,b}^0}(V) + \big[\big(P_{h,s,a,b}^0 - \widehat{P}_{h,s,a,b}^0\big)V\big]^2.$$

Analogous to (173), with probability exceeding $1 - \delta$,

$$\left|\big(\widehat{P}_{h,s,a,b}^0 - P_{h,s,a,b}^0\big)\big(\overline{V} \circ \overline{V}\big)\right| \le C_4 \sqrt{\frac{1}{N_h(s,a,b)}\text{Var}_{P_{h,s,a,b}^0}\big(\overline{V} \circ \overline{V}\big)\log\frac{KH}{\delta}} + \frac{C_4 H^2 \log\frac{KH}{\delta}}{N_h(s,a,b)}$$

$$\le C_4 \sqrt{\frac{H^2}{N_h(s,a,b)}\text{Var}_{P_{h,s,a,b}^0}(V)\log\frac{KH}{\delta}} + \frac{C_4 H^2 \log\frac{KH}{\delta}}{N_h(s,a,b)},$$
(175)

where the last inequation comes from the fact that

$$\text{Var}_{P_{h,s,a,b}^0}\big(\overline{V} \circ \overline{V}\big) \le P_{h,s,a,b}^0\big(\overline{V} \circ \overline{V} \circ \overline{V} \circ \overline{V}\big) \le H^2 P_{h,s,a,b}^0\big(\overline{V} \circ \overline{V}\big) = H^2 \text{Var}_{P_{h,s,a,b}^0}(V).$$

Employing (175), we further bound (174) as

$$\text{Var}_{P_{h,s,a,b}^0}(V) \le \text{Var}_{\widehat{P}_{h,s,a,b}^0}(V) + \big[\big(P_{h,s,a,b}^0 - \widehat{P}_{h,s,a,b}^0\big)V\big]^2$$

$$+ C_4 \sqrt{\frac{H^2 \log\frac{KH}{\delta}}{N_h(s,a,b)}\text{Var}_{P_{h,s,a,b}^0}(V)} + \frac{C_4 H^2 \log\frac{KH}{\delta}}{N_h(s,a,b)}$$

$$\le \text{Var}_{\widehat{P}_{h,s,a,b}^0}(V) + \big[\big(P_{h,s,a,b}^0 - \widehat{P}_{h,s,a,b}^0\big)V\big]^2 + \frac{C_4 H^2 \log\frac{KH}{\delta}}{N_h(s,a,b)}$$

$$+ \frac{1}{2}\text{Var}_{P_{h,s,a,b}^0}(V) + \frac{C_4^2 H^2 \log\frac{KH}{\delta}}{2N_h(s,a,b)},$$

where the last relation holds due to the AM-GM inequality. Therefore, we obtain

$$\text{Var}_{P_{h,s,a,b}^0}(V) \le 2\text{Var}_{\widehat{P}_{h,s,a,b}^0}(V) + 2\big[\big(P_{h,s,a,b}^0 - \widehat{P}_{h,s,a,b}^0\big)V\big]^2 + \frac{\big(C_4^2 + 2C_4\big)H^2 \log\frac{KH}{\delta}}{N_h(s,a,b)}. \quad (176)$$

Combining (176) and (173), we derive

$$\left|\big(\widehat{P}_{h,s,a,b}^0 - P_{h,s,a,b}^0\big)V\right| \le \sqrt{\frac{2C_4^2}{N_h(s,a,b)}\text{Var}_{\widehat{P}_{h,s,a,b}^0}(V)\log\frac{KH}{\delta}} + \frac{\sqrt{C_4^2\big(C_4^2 + 2C_4\big)H\log\frac{KH}{\delta}}}{N_h(s,a,b)}$$

$$+ \sqrt{\frac{2C_4^2}{N_h(s,a,b)}\log\frac{KH}{\delta}}\left|\big(\widehat{P}_{h,s,a,b}^0 - P_{h,s,a,b}^0\big)V\right| + \frac{C_4 H \log\frac{KH}{\delta}}{N_h(s,a,b)}.$$
(177)

Next, we consider two cases, i.e., $N_h(s, a, b) \leq \frac{1}{8C_4^2} \log \frac{KH}{\delta}$ and $N_h(s, a, b) > \frac{1}{8C_4^2} \log \frac{KH}{\delta}$. In the first case of $N_h(s, a, b) \leq \frac{1}{8C_4^2} \log \frac{KH}{\delta}$, (46) is valid since

$$\left| \inf_{P \in \mathcal{U}^{\sigma+}(\widehat{P}_{h,s,a,b}^0)} PV - \inf_{P \in \mathcal{U}^{\sigma+}(P_{h,s,a,b}^0)} PV \right| \leq \max_{\alpha \in [\min_s V(s), \max_s V(s)]} \left| (\widehat{P}_{h,s,a,b}^0 - P_{h,s,a,b}^0) V \right|$$

$$\leq 2H = O\left( \frac{H \log \frac{KH}{\delta}}{N_h(s, a, b)} \right). \tag{178}$$

In the second case of $N_h(s, a, b) > \frac{1}{8C_4^2} \log \frac{KH}{\delta}$, it follows from (177) that

$$\left| (\widehat{P}_{h,s,a,b}^0 - P_{h,s,a,b}^0) V \right| \leq \frac{1}{2} \left| (\widehat{P}_{h,s,a,b}^0 - P_{h,s,a,b}^0) V \right| + \sqrt{\frac{2C_4^2}{N_h(s, a, b)} \mathsf{Var}_{\widehat{P}_{h,s,a,b}^0}(V) \log \frac{KH}{\delta}}$$

$$+ \frac{C_4 + \sqrt{C_4^2(C_4^2 + 2C_4)}}{N_h(s, a, b)} H \log \frac{KH}{\delta},$$

which, by arranging the terms, leads to

$$\left| (\widehat{P}_{h,s,a,b}^0 - P_{h,s,a,b}^0) V \right| \leq \sqrt{\frac{8C_4^2}{N_h(s, a, b)} \mathsf{Var}_{\widehat{P}_{h,s,a,b}^0}(V) \log \frac{KH}{\delta}}$$

$$+ 2H \frac{C_4 + \sqrt{C_4^2(C_4^2 + 2C_4)}}{N_h(s, a, b)} \log \frac{KH}{\delta}. \tag{179}$$

Combining (172) and (179), we obtain

$$\left| \inf_{P \in \mathcal{U}^{\sigma+}(\widehat{P}_{h,s,a,b}^0)} PV - \inf_{P \in \mathcal{U}^{\sigma+}(P_{h,s,a,b}^0)} PV \right| \leq \sqrt{\frac{8C_4^2}{N_h(s, a, b)} \mathsf{Var}_{\widehat{P}_{h,s,a,b}^0}(V) \log \frac{KH}{\delta}}$$

$$+ 2H \frac{C_4 + \sqrt{C_4^2(C_4^2 + 2C_4)}}{N_h(s, a, b)} \log \frac{KH}{\delta}. \tag{180}$$

Joining these two cases, we conclude the proof of (46).

### G.1.2 Proof of (47).

We consider two cases, i.e., $N_h(s, a, b) < 16C_4^2 \log \frac{KH}{\delta}$ and $N_h(s, a, b) \geq 16C_4^2 \log \frac{KH}{\delta}$. In the first case of $N_h(s, a, b) < 16C_4^2 \log \frac{KH}{\delta}$, (47) is valid since

$$\mathsf{Var}_{\widehat{P}_{h,s,a,b}^0}(V) \leq H^2 = O\left( \frac{H^2 \log \frac{KH}{\delta}}{N_h(s, a, b)} \right).$$

In the second case of $N_h(s, a, b) \geq 16C_4^2 \log \frac{KH}{\delta}$, we have

$$\mathsf{Var}_{\widehat{P}_{h,s,a,b}^0}(V) \overset{(i)}{=} \mathsf{Var}_{P_{h,s,a,b}^0}(V) - \left[ (P_{h,s,a,b}^0 - \widehat{P}_{h,s,a,b}^0) V \right]^2 - (P_{h,s,a,b}^0 - \widehat{P}_{h,s,a,b}^0)(\overline{V} \circ \overline{V})$$

$$\overset{(ii)}{\leq} \mathsf{Var}_{P_{h,s,a,b}^0}(V) + C_4 \sqrt{\frac{H^2}{N_h(s, a, b)} \mathsf{Var}_{P_{h,s,a,b}^0}(V) \log \frac{KH}{\delta}} + \frac{C_4 H^2 \log \frac{KH}{\delta}}{N_h(s, a, b)}$$

$$\overset{(iii)}{\leq} 2\mathsf{Var}_{P_{h,s,a,b}^0}(V) + \frac{(C_4^2/4 + C_4) H^2 \log \frac{KH}{\delta}}{N_h(s, a, b)}$$

$$= 2\mathsf{Var}_{P_{h,s,a,b}^0}(V) + O\left( \frac{H^2 \log \frac{KH}{\delta}}{N_h(s, a, b)} \right),$$

where (i) comes from (174), (ii) holds due to (175), and (iii) is based on the AM-GM inequality. Combining the two cases, (47) holds and Lemma D.1 is proved.

## G.2 Proof of Lemma D.2

Assuming that $\widehat{Q}_h^+(s,a,b) \geq Q_h^{\star,\widehat{\nu},\sigma^+}(s,a,b)$, then we can obtain $\widehat{V}_h^+(s) \geq V_h^{\star,\nu,\sigma^+}(s)$, since

$$\widehat{V}_h^+(s) = \mathbb{E}_{a\sim\mu_h^+(s),b\sim\nu_h^+(s)}\left[\widehat{Q}_h^+(s,a,b)\right]$$
$$\overset{(i)}{\geq} \mathbb{E}_{a\sim\mu^\star(s),b\sim\widehat{\nu}(s)}\left[\widehat{Q}_h^+(s,a,b)\right] \geq \mathbb{E}_{a\sim\mu^\star(s),b\sim\widehat{\nu}(s)}\left[Q_h^{\star,\widehat{\nu},\sigma^+}(s,a,b)\right] = V_h^{\star,\widehat{\nu},\sigma^+}(s),$$

where (i) holds due to the fact that $\widehat{\nu} = \nu_h^+$ and $(\mu_h^+,\nu_h^+)$ is the Nash equilibrium of $\widehat{Q}_h^+(s,a,b)$. Hence, we need to verify

$$\widehat{Q}_h^+(s,a,b) \geq Q_h^{\star,\widehat{\nu},\sigma^+}(s,a,b), \tag{181}$$

which can be proved by mathematical induction. Specifically, (181) holds when $h = H+1$ under the trivial fact $\widehat{Q}_{H+1}^+(s,a,b) = Q_{H+1}^{\star,\sigma^+}(s,a,b) = 0$. Suppose that (181) holds for all $(s,a,b) \in \mathcal{S} \times \mathcal{A} \times \mathcal{B}$ at some time step $h \in [H]$. According to the update rule in line 4 in Algorithm 2, (181) exists if $\widehat{Q}_h^+(s,a,b) = H$ because $\widehat{Q}_h^+(s,a,b) = H \geq Q_h^{\star,\widehat{\nu},\sigma^+}(s,a,b)$. Besides, in the case of $N_h(s,a,b) = 0$, we have $\beta_h\left(s,a,b,\widehat{V}_{h+1}^+\right) = H$, leading to $\widehat{Q}_h^+(s,a,b) = H \geq Q_h^{\star,\widehat{\nu},\sigma^+}(s,a,b)$. Otherwise, for $N_h(s,a,b) > 0$, $\widehat{Q}_h^+(s,a,b)$ is updated as

$$\widehat{Q}_h^+(s,a,b) = \widehat{r}(s,a,b) + \inf_{P \in \mathcal{U}^{\sigma+}\left(\widehat{P}_{h,s,a,b}^0\right)} P\widehat{V}_{h+1}^+ + \beta_h\left(s,a,b,\widehat{V}_{h+1}^+\right)$$

$$\geq \widehat{r}(s,a,b) + \inf_{P \in \mathcal{U}^{\sigma+}\left(P_{h,s,a,b}^0\right)} P\widehat{V}_{h+1}^+ + \beta_h\left(s,a,b,\widehat{V}_{h+1}^+\right)$$

$$- \left|\inf_{P \in \mathcal{U}^{\sigma+}\left(\widehat{P}_{h,s,a,b}^0\right)} P\widehat{V}_{h+1}^+ - \inf_{P \in \mathcal{U}^{\sigma+}\left(P_{h,s,a,b}^0\right)} P\widehat{V}_{h+1}^+\right|$$

$$\geq \widehat{r}(s,a,b) + \inf_{P \in \mathcal{U}^{\sigma+}\left(P_{h,s,a,b}^0\right)} P\widehat{V}_{h+1}^+ + 0$$

$$\geq \widehat{r}(s,a,b) + \inf_{P \in \mathcal{U}^{\sigma+}\left(P_{h,s,a,b}^0\right)} P\widehat{V}_{h+1}^{\star,\widehat{\nu},\sigma^+} + 0 = Q_h^{\star,\widehat{\nu},\sigma^+}(s,a,b), \tag{182}$$

where the second inequality holds due to (48) in Lemma D.1 and the last equality comes from the empirical robust Bellman equation (30). Together with the case of $h = H+1$, we complete prove Lemma D.2.

## G.3 Proof of Lemma D.3

Following the proof by Lemma 3 in [34], we bound $\min_{s\in\mathcal{S}} \widehat{V}_h^+(s)$ and $\max_{s\in\mathcal{S}} \widehat{V}_h^+(s)$. Specifically,

$$\min_{s\in\mathcal{S}} \widehat{V}_h^+(s) = \min_{s\in\mathcal{S}} \mathbb{E}_{(a,b)\sim\mu_h^+\times\nu_h^+}\left[\widehat{Q}_h^+(s,a,b)\right]$$

$$= \min_{s\in\mathcal{S}} \mathbb{E}_{(a,b)\sim\mu_h^+\times\nu_h^+}\left[\widehat{r}_h(s,a,b) + \inf_{P \in \mathcal{U}^{\sigma+}\left(\widehat{P}_{h,s,a,b}^0\right)} P\widehat{V}_{h+1}^+ + \beta_h\left(s,a,b,\widehat{V}_{h+1}^+\right)\right]$$

$$\geq 0 + \min_{s\in\mathcal{S}} \widehat{V}_{h+1}^+(s) + 0, \tag{183}$$

where the second equality is valid due to the update rule in line 4 in Algorithm 2. Similarly,

$$\max_{s\in\mathcal{S}} \widehat{V}_h^+ = \max_{s\in\mathcal{S}} \mathbb{E}_{(a,b)\sim\mu_h^+\times\nu_h^+}\left[\widehat{Q}_h^+(s,a,b)\right]$$

$$= \max_{s\in\mathcal{S}} \mathbb{E}_{(a,b)\sim\mu_h^+\times\nu_h^+}\left[\widehat{r}_h(s,a,b) + \inf_{P \in \mathcal{U}^{\sigma+}\left(\widehat{P}_{h,s,a,b}^0\right)} P\widehat{V}_{h+1}^+ + \beta_h\left(s,a,b,\widehat{V}_{h+1}^+\right)\right]$$

$$\leq 1 + \max_{(s,a,b)\in\mathcal{S}\times\mathcal{A}\times\mathcal{B}} \inf_{P \in \mathcal{U}^{\sigma+}\left(\widehat{P}_{h,s,a,b}\right)} P\widehat{V}_{h+1}^+ + H. \tag{184}$$

For any $h \in [H]$, there exists at least one state $s_h^\star$ that satisfies $\widehat{V}_h^+(s_h^\star) = \min_{s \in \mathcal{S}} \widehat{V}_h^+(s)$. Furthermore, for any accessible uncertainty set $\sigma^+ > 0$ and $(s, a, b) \in \mathcal{S} \times \mathcal{A} \times \mathcal{B}$, we define an auxiliary vector $\widehat{P}'_{h,s,a,b} \in \mathbb{R}^S$ by reducing the values of several elements of $\widehat{P}^0_{h,s,a,b}$ strictly, namely,

$$0 \leq \widehat{P}'_{h,s,a,b} \leq \widehat{P}^0_{h,s,a,b} \quad \text{and} \quad \sum_{s' \in \mathcal{S}} \widehat{P}^0_{h,s,a,b}(s') - \widehat{P}'_{h,s,a,b}(s') = \left\| \widehat{P}'_{h,s,a,b} - \widehat{P}^0_{h,s,a,b} \right\|_1 = \sigma^+.$$

(185)

Let $l_{s_h^\star}$ represent an $S$-dimensional standard basis under $s_h^\star$. We can derive that

$$\frac{1}{2} \left\| \widehat{P}'_{h,s,a,b} + \sigma^+ \left[ l_{s_h^\star} \right]^\top - \widehat{P}^0_{h,s,a,b} \right\|_1 \leq \frac{1}{2} \left\| \widehat{P}'_{h,s,a,b} - \widehat{P}^0_{h,s,a,b} \right\|_1 + \frac{1}{2} \left\| \sigma^+ \left[ l_{s_h^\star} \right]^\top \right\|_1 \leq \sigma^+, \quad (186)$$

where the first inequality is valid since the 'distance' function (e.g., TV distance) satisfies the triangle inequality.

Therefore, we can conclude that $\widehat{P}'_{h,s,a,b} + \sigma^+ \left[ l_{s_h^\star} \right]^\top \in \mathcal{U}^{\sigma^+}(\widehat{P}^0_{h,s,a,b})$ and $\widehat{P}'_{h,s,a,b} + \sigma^+ \left[ l_{s_h^\star} \right]^\top$ is a distribution vector based on (186), leading to

$$\inf_{P \in \mathcal{U}^{\sigma^+}(\widehat{P}^0_{h,s,a,b})} P \widehat{V}_{h+1}^+ \leq \left( \widehat{P}'_{h,s,a,b} + \sigma^+ \left[ l_{s_{i,h}^\star} \right]^\top \right) \widehat{V}_{h+1}^+$$

$$\leq \left\| \widehat{P}'_{h,s,a,b} \right\|_1 \left\| \widehat{V}_{h+1}^+ \right\|_\infty + \sigma^+ \widehat{V}_{h+1}^+(s_{h+1}^\star)$$

$$\leq \left( 1 - \sigma^+ \right) \max_{s \in \mathcal{S}} \widehat{V}_{h+1}^+(s) + \sigma^+ \min_{s \in \mathcal{S}} \widehat{V}_{h+1}^+(s), \quad (187)$$

where the last inequality holds since

$$\left\| P'_{h,s,a,b} \right\|_1 = \sum_{s'} P'_{h,s,a,b}(s') = -\sum_{s'} \left( P^0_{h,s,a,b}(s') - P'_{h,s,a,b}(s') \right) + \sum_{s'} P^0_{h,s,a,b}(s') = 1 - \sigma^+.$$

(188)

Putting (187) and (184) together shows

$$\max_{s \in \mathcal{S}} \widehat{V}_h^+(s) \leq 1 + \max_{(s,a,b) \in \mathcal{S} \times \mathcal{A} \times \mathcal{B}} \inf_{P \in \mathcal{U}^{\sigma^+}(P^0_{h,s,a,b})} P \widehat{V}_{h+1}^+ + H$$

$$\leq H + 1 + \left( 1 - \sigma^+ \right) \max_{s \in \mathcal{S}} \widehat{V}_{h+1}^+(s) + \sigma^+ \min_{s \in \mathcal{S}} \widehat{V}_{h+1}^+(s). \quad (189)$$

By substituting (189) it (183), it readily follows that

$$\max_{s \in \mathcal{S}} \widehat{V}_h^+ - \min_{s \in \mathcal{S}} \widehat{V}_h^+ \leq H + 1 + \left( 1 - \sigma^+ \right) \max_{s \in \mathcal{S}} \widehat{V}_{h+1}^+(s) + \sigma^+ \min_{s \in \mathcal{S}} \widehat{V}_{h+1}^+(s) - \min_{s \in \mathcal{S}} V_{h+1}^+(s)$$

$$= H + 1 + (1 - \sigma^+) \left( \max_{s \in \mathcal{S}} \widehat{V}_{h+1}^+(s) - \min_{s \in \mathcal{S}} \widehat{V}_{h+1}^+(s) \right)$$

$$\leq H + 1 + (1 - \sigma^+) \left[ H + 1 + (1 - \sigma^+) \left( \max_{s \in \mathcal{S}} \widehat{V}_{h+2}^+(s) - \min_{s \in \mathcal{S}} \widehat{V}_{h+2}^+(s) \right) \right]$$

$$\leq \cdots \leq \frac{(H+1) \left( 1 - (1 - \sigma^+)^{H-h} \right)}{\sigma^+}. \quad (190)$$

which, combined with $\max_{s \in \mathcal{S}} \widehat{V}_h^+(s) - \min_{s \in \mathcal{S}} \widehat{V}_h^+(s) \leq H$, conclude this proof.

### G.4 Proof of Lemma D.4

First, we introduce auxiliary values and reward functions to control $\sum_{i=1}^H \sum_{(s,b) \in \mathcal{S} \times \mathcal{B}} d_i^{\mathsf{p}, \mu^{\mathsf{d}}, \nu^\star}(s, \mu^{\mathsf{d}}(s), b) \mathsf{Var}_{P^0_{i,s,\mu^{\mathsf{d}}(s),b}} \left( \widehat{V} \right)$ as below: for any time step $i$

- $\widehat{V}_i^{\mathsf{m}} := \min_{s \in \mathcal{S}} \widehat{V}_i^+(s)$: the minimum value of all the entries in vector $\widehat{V}_i^+$.
- $\widehat{V}_i' := \widehat{V}_i^+ - \widehat{V}_i^{\mathsf{m}} \mathbf{1}$: truncated value function.
- $\widehat{r}_i^{\mu^{\mathsf{d}}, \nu^\star}(s) = \mathbb{E}_{(a,b) \sim (\mu^{\mathsf{d}}(s), \nu^\star(s))} \widehat{r}_i(s, a, b)$: average reward function.

- $\widehat{r}_i^{\mathsf{m}} = r_i^{\mu^{\mathsf{d}},\nu^\star} + \left(\widehat{V}_{i+1}^{\mathsf{m}} - \widehat{V}_i^{\mathsf{m}}\right)\mathbf{1}$: truncated reward function.

Applying the robust Bellman's consistency equation in (30) gives

$$
\begin{aligned}
\widehat{V}_i' = \widehat{V}_i^+ - \widehat{V}_i^{\mathsf{m}}\mathbf{1} \overset{(i)}{\le}\; & \widehat{r}_i^{\mu^{\mathsf{d}},\nu^\star} + \widetilde{P}_i^{\inf,\widehat{V}}\widehat{V}_{i+1}^+ + 2\beta_i^{\mu^{\mathsf{d}},\nu^\star} - \widehat{V}_i^{\mathsf{m}}\mathbf{1} \\
=\; & \widehat{r}_i^{\mu^{\mathsf{d}},\nu^\star} + \widetilde{P}_i^{\inf,\widehat{V}}\widehat{V}_{i+1}^+ + \left(\widehat{V}_{i+1}^{\mathsf{m}}\mathbf{1} - \widehat{V}_i^{\mathsf{m}}\mathbf{1}\right) - \widehat{V}_{i+1}^{\mathsf{m}}\mathbf{1} + 2\beta_i^{\mu^{\mathsf{d}},\nu^\star} \\
=\; & \widehat{r}_i^{\mathsf{m}} + \widetilde{P}_i^{\inf,\widehat{V}}\widehat{V}_{i+1}^+ - \widehat{V}_{i+1}^{\mathsf{m}}\mathbf{1} + 2\beta_i^{\mu^{\mathsf{d}},\nu^\star} \\
=\; & \widehat{r}_i^{\mathsf{m}} + \widetilde{P}_i^{\inf,\widehat{V}}\widehat{V}_{i+1}' + 2\beta_i^{\mu^{\mathsf{d}},\nu^\star},
\end{aligned}
\tag{191}
$$

where (i) follows from the fact that

$$
\begin{aligned}
\widehat{V}_i^+(s) \le\; & \widehat{r}_i^{\mu^{\mathsf{d}},\nu^\star}(s) + \mathbb{E}_{(a,b)\sim(\mu^{\mathsf{d}}(s),\nu^\star(s))} \inf_{P\in\mathcal{U}^{\sigma+}\left(\widehat{P}_{i,s,a,b}^0\right)} P\widehat{V}_{i+1}^+ + \beta_i^{\mu^{\mathsf{d}},\nu^\star}(s) \\
\overset{(i)}{\le}\; & \widehat{r}_i^{\mu^{\mathsf{d}},\nu^\star}(s) + \mathbb{E}_{(a,b)\sim(\mu^{\mathsf{d}}(s),\nu^\star(s))}\left[\inf_{P\in\mathcal{U}^{\sigma+}\left(P_{i,s,a,b}^0\right)} P\widehat{V}_{i+1}^+ \right.\\
& \left. + \left|\inf_{P\in\mathcal{U}^{\sigma+}\left(\widehat{P}_{i,s,a,b}^0\right)} P\widehat{V}_{i+1}^+ - \inf_{P\in\mathcal{U}^{\sigma+}\left(P_{i,s,a,b}^0\right)} P\widehat{V}_{i+1}^+\right|\right] + \beta_i^{\mu^{\mathsf{d}},\nu^\star}(s) \\
\overset{(ii)}{\le}\; & \widehat{r}_i^{\mu^{\mathsf{d}},\nu^\star}(s) + \mathbb{E}_{(a,b)\sim(\mu^{\mathsf{d}}(s),\nu^\star(s))}\left[P_{i,s,a,b}^{\inf,\widehat{V}}\widehat{V}_{i+1}^+\right] + 2\beta_i^{\mu^{\mathsf{d}},\nu^\star}(s) \\
\overset{(iii)}{=}\; & \widehat{r}_i^{\mu^{\mathsf{d}},\nu^\star}(s) + \widetilde{P}_{i,s}^{\inf,\widehat{V}}\widehat{V}_{i+1}^+ + 2\beta_i^{\mu^{\mathsf{d}},\nu^\star}(s),
\end{aligned}
\tag{192}
$$

(ii) is valid under the notation

$$
P_{i,s,a,b}^{\inf,\widehat{V}} := \operatorname{argmin}_{P\in\mathcal{U}^{\sigma+}\left(P_{i,s,a,b}^0\right)} P\widehat{V}_{i+1}^+,
\tag{193}
$$

and (iii) holds under the notation as $\widetilde{P}_{i,s}^{\inf,\widehat{V}} := \mathbb{E}_{(a,b)\sim(\mu^{\mathsf{d}}(s),\nu^\star(s))} P_{i,s,a,b}^{\inf,\widehat{V}}$ and the sequence as $\widetilde{P}_i^{\inf,V} \in \mathbb{R}^{S\times S}$. Besides, (i) in (192) exists due to (48) in Lemma D.1 for $N_i(s,a,b) > 0$ and

$$
\left|\inf_{P\in\mathcal{U}^{\sigma+}\left(P_{i,s,a,b}^0\right)} P\widehat{V}_{i+1}^+ - \inf_{P\in\mathcal{U}^{\sigma+}\left(\widehat{P}_{i,s,a,b}^0\right)} P\widehat{V}_{i+1}^+\right| \le H = \beta_i^{\mu^{\mathsf{d}},\nu^\star}(s) \text{ for } N_i(s,a,b) = 0.
\tag{194}
$$

Then, we have

$$
\begin{aligned}
& \mathbb{E}_{(a,b)\sim(\mu^{\mathsf{d}}(s),\nu^\star(s))}\mathsf{Var}_{P_{i,s,a,b}^{\inf,V}}\left(\widehat{V}_{i+1}^+\right) \\
\overset{(i)}{=}\; & \mathbb{E}_{(a,b)\sim(\mu^{\mathsf{d}}(s),\nu^\star(s))}\mathsf{Var}_{P_{i,s,a,b}^{\inf,V}}\left(\widehat{V}_{i+1}'\right) \\
=\; & \mathbb{E}_{(a,b)\sim(\mu^{\mathsf{d}}(s),\nu^\star(s))}\left[P_{i,s,a,b}^{\inf,V}\left(\widehat{V}_{i+1}'\circ\widehat{V}_{i+1}'\right) - (P_{i,s,a,b}^{\inf,V}\widehat{V}_{i+1}')\circ(P_{i,s,a,b}^{\inf,V}\widehat{V}_{i+1}')\right] \\
\le\; & \mathbb{E}_{(a,b)\sim(\mu^{\mathsf{d}}(s),\nu^\star(s))}\left[P_{i,s,a,b}^{\inf,V}\left(\widehat{V}_{i+1}'\circ\widehat{V}_{i+1}'\right) - (P_{i,s,a,b}^{\inf,\widehat{V}}\widehat{V}_{i+1}')\circ(P_{i,s,a,b}^{\inf,\widehat{V}}\widehat{V}_{i+1}')\right] \\
\overset{(ii)}{\le}\; & \widetilde{P}_{i,s}^{\inf,V}\left(\widehat{V}_{i+1}'\circ\widehat{V}_{i+1}'\right) - (\widetilde{P}_{i,s}^{\inf,\widehat{V}}\widehat{V}_{i+1}')\circ(\widetilde{P}_{i,s}^{\inf,\widehat{V}}\widehat{V}_{i+1}') \\
=\; & \widetilde{P}_{i,s}^{\inf,V}\left(\widehat{V}_{i+1}'\circ\widehat{V}_{i+1}'\right) - \widehat{V}_i'(s)\circ\widehat{V}_i'(s) + \widehat{V}_i'(s)\circ\widehat{V}_i'(s) - (\widetilde{P}_{i,s}^{\inf,\widehat{V}}\widehat{V}_{i+1}')\circ(\widetilde{P}_{i,s}^{\inf,\widehat{V}}\widehat{V}_{i+1}') \\
=\; & \widetilde{P}_{i,s}^{\inf,V}\left(\widehat{V}_{i+1}'\circ\widehat{V}_{i+1}'\right) - \widehat{V}_i'(s)\circ\widehat{V}_i'(s) + \left(\widehat{V}_i'(s) - (\widetilde{P}_{i,s}^{\inf,\widehat{V}}\widehat{V}_{i+1}')\right)\circ\left(\widehat{V}_i'(s) + (\widetilde{P}_{i,s}^{\inf,\widehat{V}}\widehat{V}_{i+1}')\right) \\
\overset{(iii)}{\le}\; & \widetilde{P}_{i,s}^{\inf,V}\left(\widehat{V}_{i+1}'\circ\widehat{V}_{i+1}'\right) - \widehat{V}_i'(s)\circ\widehat{V}_i'(s) + \left(\widehat{r}_i^{\mathsf{m}}(s) + 2\beta_h^{\mu^{\mathsf{d}},\nu^\star}(s)\right)\circ\left(\widehat{V}_i'(s) + (\widetilde{P}_{i,s}^{\inf,\widehat{V}}\widehat{V}_{i+1}')\right) \\
\overset{(iv)}{\le}\; & \widetilde{P}_{i,s}^{\inf,V}\left(\widehat{V}_{i+1}'\circ\widehat{V}_{i+1}'\right) - \widehat{V}_i'(s)\circ\widehat{V}_i'(s) + \left(\left\|\widehat{V}_i'\right\|_\infty + \left\|\widehat{V}_{i+1}'\right\|_\infty\right)\left(2\beta_i^{\mu^{\mathsf{d}},\nu^\star}(s) + 1\right),
\end{aligned}
\tag{195}
$$

where (i) follows from that $\mathrm{Var}_{P_{i,s}^{\mathrm{inf},V}}(V - b\mathbf{1}) = \mathrm{Var}_{P_{i,s}^{\mathrm{inf},V}}(V)$ for any value vector $V \in \mathbb{R}^S$ and scalar $b$, (ii) holds since

$$\mathbb{E}_{(a,b)\sim(\mu^{\mathsf{d}}(s),\nu^{\star}(s))}\left[\left(P_{i,s,a,b}^{\mathrm{inf},\widehat{V}}\widehat{V}_{i+1}'\right)\circ\left(P_{i,s,a,b}^{\mathrm{inf},\widehat{V}}\widehat{V}_{i+1}'\right)\right]$$
$$\geq\mathbb{E}_{(a,b)\sim(\mu^{\mathsf{d}}(s),\nu^{\star}(s))}\left[\left(P_{i,s,a,b}^{\mathrm{inf},\widehat{V}}\widehat{V}_{i+1}'\right)\right]\circ\mathbb{E}_{(a,b)\sim(\mu^{\mathsf{d}}(s),\nu^{\star}(s))}\left[\left(P_{i,s,a,b}^{\mathrm{inf},\widehat{V}}\widehat{V}_{i+1}'\right)\right],$$

(iii) comes from (191), and (iv) arises from $\widehat{r}_i^{\mathsf{m}} \leq r_i \leq 1$ due to $\widehat{V}_{i+1}^{\mathsf{m}} - \widehat{V}_i^{\mathsf{m}} \leq 0$ by definition. Consequently, combining (62), we arrive at

$$\sum_{(s,b)\in\mathcal{S}\times\mathcal{B}} d_i^{\mathsf{p},\mu^{\mathsf{d}},\nu^{\star}}(s,\mu^{\mathsf{d}}(s),b)\mathsf{Var}_{P_{i,s,a,b}^{\mathrm{inf},V}}\left(\widehat{V}_{i+1}^+\right)$$

$$=\sum_{s\in\mathcal{S}} d_i^{\mathsf{p},\mu^{\mathsf{d}},\nu^{\star}}(s)\mathbb{E}_{(a,b)\sim(\mu^{\mathsf{d}}(s),\nu^{\star}(s))}\mathsf{Var}_{P_{i,s,a,b}^{\mathrm{inf},V}}\left(\widehat{V}_{i+1}^+\right)$$

$$\leq\sum_{s\in\mathcal{S}} d_i^{\mathsf{p},\mu^{\mathsf{d}},\nu^{\star}}(s)\left(\widetilde{P}_{i,s}^{\mathrm{inf},V}\left(\widehat{V}_{i+1}'\circ\widehat{V}_{i+1}'\right)-\widehat{V}_i'(s)\circ\widehat{V}_i'(s)\right.$$
$$\left.+\left(\left\|\widehat{V}_i'\right\|_\infty+\left\|\widehat{V}_{i+1}'\right\|_\infty\right)\left(2\beta_i^{\mu^{\mathsf{d}},\nu^{\star}}(s)+1\right)\right)$$

$$\leq\sum_{s\in\mathcal{S}} d_i^{\mathsf{p},\mu^{\mathsf{d}},\nu^{\star}}(s)\left(\widetilde{P}_{i,s}^{\mathrm{inf},V}\left(\widehat{V}_{i+1}'\circ\widehat{V}_{i+1}'\right)-\widehat{V}_i'(s)\circ\widehat{V}_i'(s)\right)+\left(\left\|\widehat{V}_i'\right\|_\infty+\left\|\widehat{V}_{i+1}'\right\|_\infty\right)$$
$$+2\left(\left\|\widehat{V}_i'\right\|_\infty+\left\|\widehat{V}_{i+1}'\right\|_\infty\right)\sum_{s\in\mathcal{S}} d_i^{\mathsf{p},\mu^{\mathsf{d}},\nu^{\star}}(s)\beta_i^{\mu^{\mathsf{d}},\nu^{\star}}(s)$$

$$=\sum_{s\in\mathcal{S}}\left(d_{i+1}^{\mathsf{p},\mu^{\mathsf{d}},\nu^{\star}}(s)\left(\widehat{V}_{i+1}'(s)\circ\widehat{V}_{i+1}'(s)\right)-d_i^{\mathsf{p},\mu^{\mathsf{d}},\nu^{\star}}(s)\widehat{V}_i'(s)\circ\widehat{V}_i'(s)\right)+\left(\left\|\widehat{V}_i'\right\|_\infty+\left\|\widehat{V}_{i+1}'\right\|_\infty\right)$$
$$+2\left(\left\|\widehat{V}_i'\right\|_\infty+\left\|\widehat{V}_{i+1}'\right\|_\infty\right)\sum_{s\in\mathcal{S}} d_i^{\mathsf{p},\mu^{\mathsf{d}},\nu^{\star}}(s)\beta_i^{\mu^{\mathsf{d}},\nu^{\star}}(s)$$

$$=\sum_{s\in\mathcal{S}}\left(d_{i+1}^{\mathsf{p},\mu^{\mathsf{d}},\nu^{\star}}(s)\left(\widehat{V}_{i+1}'(s)\circ\widehat{V}_{i+1}'(s)\right)-d_i^{\mathsf{p},\mu^{\mathsf{d}},\nu^{\star}}(s)\widehat{V}_i'(s)\circ\widehat{V}_i'(s)\right)+\left(\left\|\widehat{V}_i'\right\|_\infty+\left\|\widehat{V}_{i+1}'\right\|_\infty\right)$$
$$+2\left(\left\|\widehat{V}_i'\right\|_\infty+\left\|\widehat{V}_{i+1}'\right\|_\infty\right)\sum_{(s,b)\in\mathcal{S}\times\mathcal{B}} d_i^{\mathsf{p},\mu^{\mathsf{d}},\nu^{\star}}(s,\mu^{\mathsf{d}}(s),b)\beta_i(s,\mu^{\mathsf{d}}(s),b,\widehat{V}). \tag{196}$$

Besides, under TV distance, we have

$$\left|\mathsf{Var}_{P_{i,s,a,b}^0}\left(\widehat{V}_{i+1}^+\right)-\mathsf{Var}_{P_{i,s,a,b}^{\mathrm{inf},V}}\left(\widehat{V}_{i+1}^+\right)\right|=\left|\mathsf{Var}_{P_{i,s,a,b}^0}\left(\widehat{V}_{i+1}'\right)-\mathsf{Var}_{P_{i,s,a,b}^{\mathrm{inf},V}}\left(\widehat{V}_{i+1}'\right)\right|$$
$$\leq\left\|P_{i,s,a,b}^0-P_{i,s,a,b}^{\mathrm{inf},V}\right\|_1\left\|\widehat{V}_{i+1}'\right\|_\infty^2$$
$$\leq\sigma^+\left\|\widehat{V}_{i+1}'\right\|_\infty^2\leq(H+1)\left\|\widehat{V}_{i+1}'\right\|_\infty, \tag{197}$$

where the last inequality comes from Lemma D.3.

Therefore, we derive

$$\sum_{i=1}^{H} \sum_{(s,b)\in\mathcal{S}\times\mathcal{B}} d_i^{\mathsf{p},\mu^{\mathsf{d}},\nu^{\star}}(s,\mu^{\mathsf{d}}(s),b)\mathsf{Var}_{P_{i,s,a,b}^0}\big(\widehat{V}_{i+1}^+\big)$$

$$\leq \sum_{i=1}^{H} \sum_{(s,b)\in\mathcal{S}\times\mathcal{B}} d_i^{\mathsf{p},\mu^{\mathsf{d}},\nu^{\star}}(s,\mu^{\mathsf{d}}(s),b)\mathsf{Var}_{P_{i,s,a,b}^{\inf,V}}\big(\widehat{V}_{i+1}^+\big)$$

$$+ \sum_{i=1}^{H} \sum_{(s,b)\in\mathcal{S}\times\mathcal{B}} d_i^{\mathsf{p},\mu^{\mathsf{d}},\nu^{\star}}(s,\mu^{\mathsf{d}}(s),b)\left|\mathsf{Var}_{P_{i,s,a,b}^0}\big(\widehat{V}_{i+1}^+\big) - \mathsf{Var}_{P_{i,s,a,b}^{\inf,V}}\big(\widehat{V}_{i+1}^+\big)\right|$$

$$\leq \sum_{i=1}^{H} 2\left(\big\|\widehat{V}_i'\big\|_\infty + \big\|\widehat{V}_{i+1}'\big\|_\infty\right)\sum_{s\in\mathcal{S}} d_i^{\mathsf{p},\mu^{\mathsf{d}},\nu^{\star}}(s)\beta_i^{\mu^{\mathsf{d}},\nu^{\star}}(s) + \sum_{i=1}^{H}\left(\big\|\widehat{V}_i'\big\|_\infty + (H+2)\big\|\widehat{V}_{i+1}'\big\|_\infty\right)$$

$$+ \sum_{s\in\mathcal{S}} d_{H+1}^{\mathsf{p},\mu^{\mathsf{d}},\nu^{\star}}(s)\widehat{V}_{H+1}'(s)\circ\widehat{V}_{H+1}'(s)$$

$$\leq 4\sum_{i=1}^{H}\min\left\{\frac{(H+1)\left(1-(1-\sigma^+)^{H-i}\right)}{\sigma^+},H\right\}\sum_{(s,b)\in\mathcal{S}\times\mathcal{B}} d_i^{\mathsf{p},\mu^{\mathsf{d}},\nu^{\star}}(s,\mu^{\mathsf{d}}(s),b)\beta_i(s,\mu^{\mathsf{d}}(s),b,\widehat{V})$$

$$+ (H+3)\sum_{i=1}^{H}\min\left\{\frac{(H+1)\left(1-(1-\sigma^+)^{H-i}\right)}{\sigma^+},H\right\}$$

$$\overset{(i)}{\leq} 4\sum_{i=1}^{H}\min\left\{\frac{(H+1)\left(1-(1-\sigma^+)^{H-i}\right)}{\sigma^+},H\right\}\sum_{i=1}^{H}\sum_{(s,b)\in\mathcal{S}\times\mathcal{B}} d_i^{\mathsf{p},\mu^{\mathsf{d}},\nu^{\star}}(s,\mu^{\mathsf{d}}(s),b)\beta_i(s,\mu^{\mathsf{d}}(s),b,\widehat{V})$$

$$+ (H+3)\sum_{i=1}^{H}\min\left\{\frac{(H+1)\left(1-(1-\sigma^+)^{H-i}\right)}{\sigma^+},H\right\}$$

$$\overset{(ii)}{\leq} 4H\min\left\{\frac{2(H\sigma^+-1+(1-\sigma^+)^H)}{(\sigma^+)^2},H\right\}\sum_{i=1}^{H}\sum_{(s,b)\in\mathcal{S}\times\mathcal{B}} d_i^{\mathsf{p},\mu^{\mathsf{d}},\nu^{\star}}(s,\mu^{\mathsf{d}}(s),b)\beta_i(s,\mu^{\mathsf{d}}(s),b,\widehat{V})$$

$$+ (H+3)H\min\left\{\frac{2(H\sigma^+-1+(1-\sigma^+)^H)}{(\sigma^+)^2},H\right\}, \tag{198}$$

where (i) comes from Cauchy-Schwarz inequality and the (ii) holds since

$$\sum_{i=1}^{H}\frac{(H+1)\left(1-(1-\sigma^+)^{H-i}\right)}{\sigma^+} = \frac{H(H+1)}{\sigma^+} - \sum_{i=0}^{H-1}\frac{(H+1)(1-\sigma^+)^i}{\sigma^+}$$

$$= \frac{H(H+1)}{\sigma^+} - \frac{(H+1)(1-(1-\sigma^+)^H)}{(\sigma^+)^2}$$

$$= \frac{(H+1)(H\sigma^+-1+(1-\sigma^+)^H)}{(\sigma^+)^2}$$

$$\leq \frac{2H(H\sigma^+-1+(1-\sigma^+)^H)}{(\sigma^+)^2}.$$

## H Proof for Theorem 4.4

To establish Theorem 4.4, we present key auxiliary facts, each addressing critical components of the proof and providing essential theoretical underpinnings.

### H.1 Proof of Lemma E.2

Since all RMDPs in $\mathcal{M}(\mathcal{F},\Phi)$ are constructed similarly for each $w\in\mathcal{F}$ and $\phi\in\Phi$, we will focus on a specific RMDP $\mathcal{M}_f^\phi\in\mathcal{M}(\mathcal{F},\Phi)$, with the results applicable to all other RMDPs in $\mathcal{M}(\mathcal{F},\Phi)$.

### H.1.1 Ordering the robust value function over different states

Before proceeding, we introduce several facts and notations that will be useful throughout this section. First, for any $\mathcal{M}_f^\phi$ and any policy $\widetilde{\mu}$, we observe the following at the final step $H+1$:

$$\forall s \in \mathcal{M} \cup \mathcal{N}: \quad V_{H+1}^{\widetilde{\mu},\sigma^+,f,\phi}(s) = 0. \tag{199}$$

Then for step $H$, we can easily verify that

$$\forall s \in \mathcal{N}: \quad V_H^{\widetilde{\mu},\sigma^+,f,\phi}(s) = \mathbb{E}_{a \sim \widetilde{\mu}_H(\cdot \mid s)} \left[ r_H(s,a) + \inf_{\mathcal{P} \in \mathcal{U}^{\sigma^+}(P_{H,s,a}^{f,\phi})} \mathcal{P} V_{H+1}^{\widetilde{\mu},\sigma^+,f,\phi} \right] = 1 \tag{200a}$$

$$\forall s \in \mathcal{M}: \quad V_H^{\widetilde{\mu},\sigma^+,f,\phi}(s) = \mathbb{E}_{a \sim \widetilde{\mu}_H(\cdot \mid s)} \left[ r_H(s,a) + \inf_{\mathcal{P} \in \mathcal{U}^{\sigma^+}(P_{H,s,a}^{f,\phi})} \mathcal{P} V_{H+1}^{\widetilde{\mu},\sigma^+,f,\phi} \right] = 0, \tag{200b}$$

which holds by (199) and the definition of the reward function (see (87)). The above fact directly indicates that

$$\forall (s,s') \in \mathcal{M} \times \mathcal{N}: \quad \min_{\widetilde{s} \in \mathcal{S}} V_H^{\widetilde{\mu},\sigma^+,f,\phi}(\widetilde{s}) = V_H^{\widetilde{\mu},\sigma^+,f,\phi}(m_f) \le V_H^{\widetilde{\mu},\sigma^+,f,\phi}(s) < V_H^{\widetilde{\mu},\sigma^+,f,\phi}(s'), \tag{201a}$$

$$\forall (s,s') \in \mathcal{N} \times \mathcal{N}: \quad V_H^{\widetilde{\mu},\sigma^+,f,\phi}(s) = V_H^{\widetilde{\mu},\sigma^+,f,\phi}(s'). \tag{201b}$$

Then we introduce a claim which we will prove by induction in a moment as below:

$$\forall (h,s,s') \in [H] \times \mathcal{M} \times \mathcal{N}: \quad V_h^{\widetilde{\mu},\sigma^+,f,\phi}(m_f) \le V_h^{\widetilde{\mu},\sigma^+,f,\phi}(s) < V_h^{\widetilde{\mu},\sigma^+,f,\phi}(s') \tag{202a}$$

$$\forall (s,s') \in \mathcal{N} \times \mathcal{N}: \quad V_h^{\widetilde{\mu},\sigma^+,f,\phi}(s) = V_h^{\widetilde{\mu},\sigma^+,f,\phi}(s'). \tag{202b}$$

Note that the base case when the time step is $H+1$ is verified in (201). Assume that the following fact at time step $h+1$ holds

$$\forall (s,s') \in \mathcal{M} \times \mathcal{N}: \quad \min_{\widetilde{s} \in \mathcal{S}} V_{h+1}^{\widetilde{\mu},\sigma^+,f,\phi}(\widetilde{s}) = V_{h+1}^{\widetilde{\mu},\sigma^+,f,\phi}(m_f) \le V_{h+1}^{\widetilde{\mu},\sigma^+,f,\phi}(s) < V_{h+1}^{\widetilde{\mu},\sigma^+,f,\phi}(s'), \tag{203a}$$

$$\forall (s,s') \in \mathcal{N} \times \mathcal{N}: \quad V_{h+1}^{\widetilde{\mu},\sigma^+,f,\phi}(s) = V_{h+1}^{\widetilde{\mu},\sigma^+,f,\phi}(s'). \tag{203b}$$

The rest of the proof focuses on proving the same property for time step $h$. For RMDP $\mathcal{M}_f^\phi \in \mathcal{M}(\mathcal{F}, \Phi)$ and any policy $\widetilde{\mu}$, we characterize the robust value function of different states separately:

- *For state $s \in \mathcal{N}$:* we observe that for any $s \in \mathcal{N}$,

$$
\begin{aligned}
V_h^{\widetilde{\mu},\sigma^+,f,\phi}(s) &= \mathbb{E}_{a \sim \widetilde{\mu}_h(\cdot \mid s)} \left[ r_h(s,a) + \inf_{P \in \mathcal{U}^{\sigma^+}(P_{h,s,a}^{f,\phi})} P V_{h+1}^{\widetilde{\mu},\sigma^+,f,\phi} \right] \\
&\stackrel{(i)}{=} 1 + \mathbb{E}_{a \sim \widetilde{\mu}_h(\cdot \mid s)} \left[ P_h^{\inf,f,\phi}(s \mid s,a) V_{h+1}^{\widetilde{\mu},\sigma^+,f,\phi}(s) \right] + \sigma^+ V_{h+1}^{\widetilde{\mu},\sigma^+,f,\phi}(m_f) \\
&= 1 + (1 - \sigma^+) V_{h+1}^{\widetilde{\mu},\sigma^+,f,\phi}(s) + \sigma^+ V_{h+1}^{\widetilde{\mu},\sigma^+,f,\phi}(m_f),
\end{aligned} \tag{204}
$$

  where (i) holds by $r_h(s,a) = 1$ for all $s \in \mathcal{N}$ (see (87)), the fact that $\min_{\widetilde{s} \in \mathcal{S}} V_{h+1}^{\widetilde{\mu},\sigma^+,f,\phi}(\widetilde{s}) = V_{h+1}^{\widetilde{\mu},\sigma^+,f,\phi}(m_f)$ induced by the induction assumption (cf. (203)) and the definition of $P_h^{\inf,f,\phi}(s \mid s,a)$ in (90), and the last equality follows from $P^{f,\phi}(s \mid s,a) = 1$ for all $(s,a) \in \mathcal{N} \times \mathcal{A}_{\text{one}}$. Resorting to the induction assumption in (203), we have

$$\forall (s,s') \in \mathcal{N} \times \mathcal{N}: \quad V_h^{\widetilde{\mu},\sigma^+,f,\phi}(s) = V_h^{\widetilde{\mu},\sigma^+,f,\phi}(s'). \tag{205}$$

- *For state $m_f$:* first, the robust value function at state $m_f$ obeys

$$V_h^{\widetilde{\mu},\sigma^+,f,\phi}(m_f) = \mathbb{E}_{a\sim\widetilde{\mu}_h(\cdot\,|\,m_f)}\left[r_h(m_f,a) + \inf_{P\in\mathcal{U}^{\sigma^+}(P_{h,m_f,a}^{f,\phi})} PV_{h+1}^{\widetilde{\mu},\sigma^+,f,\phi}\right]$$

$$\overset{(i)}{=} 0 + \widetilde{\mu}_h(\phi_h\,|\,m_f)\inf_{P\in\mathcal{U}^{\sigma^+}(P_{h,m_f,\phi_h}^{f,\phi})} PV_{h+1}^{\widetilde{\mu},\sigma^+,f,\phi}$$

$$+ \widetilde{\mu}_h(1-\phi_h\,|\,m_f)\inf_{P\in\mathcal{U}^{\sigma^+}(P_{h,m_f,1-\phi_h}^{f,\phi})} PV_{h+1}^{\widetilde{\mu},\sigma^+,f,\phi}$$

$$\overset{(ii)}{=} \widetilde{\mu}_h(\phi_h\,|\,m_f)\Big[p^{\inf}V_{h+1}^{\widetilde{\mu},\sigma^+,f,\phi}(n_f) + \big(1-p^{\inf}\big)V_{h+1}^{\widetilde{\mu},\sigma^+,f,\phi}(m_f)\Big]$$

$$+ \widetilde{\mu}_h(1-\phi_h\,|\,m_f)\Big[q^{\inf}V_{h+1}^{\widetilde{\mu},\sigma^+,f,\phi}(n_f) + \big(1-q^{\inf}\big)V_{h+1}^{\widetilde{\mu},\sigma^+,f,\phi}(m_f)\Big]$$

$$\overset{(iii)}{=} m_h^{\widetilde{\mu},f,\phi}V_{h+1}^{\widetilde{\mu},\sigma^+,f,\phi}(n_f) + (1-m_h^{\widetilde{\mu},f,\phi})V_{h+1}^{\widetilde{\mu},\sigma^+,f,\phi}(m_f) \tag{206}$$

$$\leq (1-\sigma^+)V_{h+1}^{\widetilde{\mu},\sigma^+,f,\phi}(n_f) + \sigma^+ V_{h+1}^{\widetilde{\mu},\sigma^+,f,\phi}(m_f). \tag{207}$$

where (i) uses the definition of the robust value function and the reward function in (87), (ii) uses the induction assumption in (203) so that the minimum is attained by picking the choice specified in (91) to absorb probability mass to state $m_f$, and (iii) holds by plugging in the definition (93) of $m_h^{\widetilde{\mu},f,\phi}$. Finally, the last inequality follows from the fact that function $f(m) := mV_{h+1}^{\widetilde{\mu},\sigma^+,f,\phi}(n_f) + (1-m)V_{h+1}^{\widetilde{\mu},\sigma^+,f,\phi}(m_f)$ is monotonically increasing with $m$ since $V_{h+1}^{\widetilde{\mu},\sigma^+,f,\phi}(n_f) > V_{h+1}^{\widetilde{\mu},\sigma^+,f,\phi}(m_f)$ (see the induction assumption (203)), and the fact $m_h^{\widetilde{\mu},f,\phi} \leq 1-\sigma^+$.

Combining the above results with (205), we confirm the claim in (202).

### H.1.2 Deriving the optimal policy and optimal robust value function

We shall characterize the optimal policy and corresponding optimal robust value function for different states separately:

- *For states in $\mathcal{M}$:* Recall (206)

$$V_h^{\widetilde{\mu},\sigma^+,f,\phi}(m_f) = m_h^{\widetilde{\mu},f,\phi}V_{h+1}^{\widetilde{\mu},\sigma^+,f,\phi}(n_f) + (1-m_h^{\widetilde{\mu},f,\phi})V_{h+1}^{\widetilde{\mu},\sigma^+,f,\phi}(m_f) \tag{208}$$

and the fact $V_{h+1}^{\widetilde{\mu},\sigma^+,f,\phi}(n_f) > V_{h+1}^{\widetilde{\mu},\sigma^+,f,\phi}(m_f)$ in (202). We observe that (208) is monotonicity increasing w.r.t. $m_h^{\widetilde{\mu},f,\phi}$, and $m_h^{\widetilde{\mu},f,\phi}$ is also increasing in $\widetilde{\mu}_h(\phi_h\,|\,m_f)$ (refer to the fact $p^{\inf} \geq q^{\inf}$ since $p \geq q$; see (99) and (91)). Consequently, the optimal policy and optimal robust value function in state $m_f$ thus obey

$$\forall h \in [H]: \quad \widetilde{\mu}_h^{\star,f,\phi}(\phi_h\,|\,m_f) = 1,$$

$$V_h^{\star,\sigma^+,f,\phi}(m_f) = p^{\inf}V_{h+1}^{\star,\sigma^+,f,\phi}(n_f) + \big(1-p^{\inf}\big)V_{h+1}^{\star,\sigma^+,f,\phi}(m_f). \tag{209}$$

- *For states $s \in \mathcal{N}$:* Recall the transitions in (98) and (86). Considering that the action does not influence the state transition for all states $s \in \mathcal{N}$, without loss of generality, we choose the robust optimal policy obeying

$$\forall s \in \mathcal{N}: \quad \widetilde{\mu}_h^{\star,f,\phi}(\phi_h\,|\,s) = 1. \tag{210}$$

### H.2 Proof of (105)

With the initial state distribution and behavior policy defined in (97), we have for any MDP $\mathcal{M}_\phi^f$,

$$d_1^{\mathsf{n},P^{\phi,f}}(s) = \varrho^{\mathsf{n}}(s) = \overline{\varrho}(s),$$

which leads to

$$\forall (m_f, a) \in \mathcal{M} \times \mathcal{A}_{\text{one}}: \quad d_1^{\mathsf{n}, P^{\phi, f}}(m_f, a) = \frac{1}{2}\overline{\varrho}(m_f). \tag{211}$$

Along with $d_1^{\mathsf{n}, P^{\phi, f}}(n_f, a) = \frac{1}{2}\overline{\varrho}(n_f) = 0$, (105a) is proved.

In view of (98), the state occupancy distribution at any step $h = 2, 3, \cdots, H$ obeys

$$
\begin{aligned}
d_h^{\mathsf{n}, P^{\phi, f}}(m_f) &\geq \mathbb{P}\{s_h = s' \mid s_{h-1} = m_f; \widetilde{\mu}^{\mathsf{n}}\} \\
&\geq d_{h-1}^{\mathsf{n}, P^{\phi, f}}(m_f)\left[\widetilde{\mu}_{h-1}^{\mathsf{n}}(\phi_{h-1} \mid m_f)(1 - p - \Delta) + \widetilde{\mu}_{h-1}^{\mathsf{n}}(1 - \phi_{h-1} \mid m_f)(1 - p)\right] \\
&\geq d_{h-1}^{\mathsf{n}, P^{\phi, f}}(m_f)(1 - p - \Delta) \geq \cdots \geq d_1^{\mathsf{n}, P^{\phi}}(m_f)\prod_{j=0}^{h-1}(1 - p - \Delta) \\
&\geq d_1^{\mathsf{n}, P^{\phi}}(m_f)\left(1 - p - \Delta\right)^H > \frac{\overline{\varrho}(m_f)}{2},
\end{aligned}
\tag{212}
$$

where the last line makes use of the properties $p$ and $\Delta$ in (101) and

$$\left(1 - p - \Delta\right)^H \geq \left(1 - \frac{c_2}{2H}\right)^H \geq \left(1 - \frac{1}{2H}\right)^H \geq \frac{1}{2},$$

provided that $0 < c_2 < 1$. In addition, as state $n_f$ is an absorbing state and state $m_f$ will only transfer to itself or state $n_f$ at each time step, we directly achieve that

$$d_h^{\mathsf{n}, P^{\phi, f}}(m_f) \leq d_{h-1}^{\mathsf{n}, P^{\phi, f}}(m_f) \leq \cdots \leq d_1^{\mathsf{n}, P^{\phi, f}}(m_f) \leq \overline{\varrho}(m_f). \tag{213}$$

For state $n_f$, as it is absorbing, we directly have

$$d_h^{\mathsf{n}, P^{\phi, f}}(n_f) = \mathbb{P}\{s_h = n_f \mid s_{h-1} = n_f; \widetilde{\mu}^{\mathsf{n}}\} \geq d_{h-1}^{\mathsf{n}, P^{\phi, f}}(n_f) \geq \cdots \geq d_1^{\mathsf{n}, P^{\phi, f}}(n_f) = \overline{\varrho}(n_f). \tag{214}$$

According to the assumption in (104), it is easily verified that

$$d_h^{\mathsf{n}, P^{\phi, f}}(n_f) \leq 1 \leq 2\overline{\varrho}(n_f). \tag{215}$$

Finally, combining (212), (213), (214), (215), the definitions of $P_h^\star(\cdot \mid s, a)$ in (98), and the Markov property, we arrive at for any $(h, s) \in [H] \times \mathcal{S}$,

$$\frac{\overline{\varrho}(s)}{2} \leq d_h^{\mathsf{n}, P^{\phi, f}}(s) \leq 2\overline{\varrho}(s), \tag{216}$$

which directly leads to

$$\frac{\overline{\varrho}(s)}{4} \leq d_h^{\mathsf{n}, P^{\phi, f}}(s, a) = \widetilde{\mu}_1^{\mathsf{n}}(a \mid s) d_h^{\mathsf{n}, P^{\phi, f}}(s) \leq \overline{\varrho}(s). \tag{217}$$

### H.3 Proof of (107)

Examining the definition of $C_{\mathsf{r}}^\star$ in (20), we make the following observations.

- For $h = 1$, we have

$$
\begin{aligned}
&\max_{(s, a, P) \in \mathcal{S}_{\text{one}} \times \mathcal{A}_{\text{one}} \times \mathcal{U}^\sigma(P^\phi)} \frac{\min\left\{d_1^{\star, P}(s, a), \frac{1}{4SA}\right\}}{d_1^{\mathsf{n}, P^{\phi, f}}(s, a)} \\
&\overset{\text{(i)}}{=} \max_{(s, P) \in \mathcal{M} \times \mathcal{U}^\sigma(P^\phi)} \frac{\min\left\{d_1^{\star, P}(s, \phi_1), \frac{1}{4SA}\right\}}{d_1^{\mathsf{n}, P^{\phi, f}}(s, \phi_1)} \\
&\overset{\text{(ii)}}{=} \max_{(s, P) \in \mathcal{M} \times \mathcal{U}^\sigma(P^\phi)} \frac{1}{4SA d_1^{\mathsf{n}, P^{\phi, f}}(s, \phi_1)} \\
&\overset{\text{(iii)}}{=} \max_{s \in \mathcal{M}} \frac{1}{2SA\overline{\varrho}(s)} = C,
\end{aligned}
\tag{218}
$$

where (i) holds by $d_1^{\star, P}(s) = \rho(s) = 0$ for all $s \in \mathcal{N}$ (see (106)) and $\widetilde{\mu}_h^{\star, \phi}(\phi_h \mid s) = 1$ for all $(s, h) \in \mathcal{M} \times [H]$ (see (95)), (ii) follows from the fact that $d_1^{\star, P}(s, \phi_1) = 1$ for all $s \in \mathcal{M}$, (iii) is verified in (105), and the last equality arises from the definition in (103).

- Similarly, for $h = 2, 3, \cdots, H$, we arrive at

$$\max_{(s,a,P) \in \mathcal{S}_{\mathrm{one}} \times \mathcal{A}_{\mathrm{one}} \times \mathcal{U}^\sigma(P^\phi)} \frac{\min\left\{d_h^{\star,P}(s,a), \frac{1}{4SA}\right\}}{d_h^{\mathsf{n},P^{\phi,f}}(s,a)}$$

$$\overset{\mathrm{(i)}}{=} \max_{(s,P) \in \mathcal{S} \times \mathcal{U}^\sigma(P^\phi)} \frac{\min\left\{d_h^{\star,P}(s,\phi_h), \frac{1}{4SA}\right\}}{d_h^{\mathsf{n},P^{\phi,f}}(s,\phi_h)}$$

$$\leq \max_{(s,P) \in \mathcal{M} \times \mathcal{U}^\sigma(P^\phi)} \frac{1}{4SA d_h^{\mathsf{n},P^{\phi,f}}(s,\phi_h)}$$

$$\overset{\mathrm{(ii)}}{\leq} \max_{s \in \mathcal{M}} \frac{1}{2SA\overline{\varrho}(s)} = 2C, \tag{219}$$

where (i) holds by the optimal policy in (95) and the trivial fact that $d_h^{\star,P}(s) = 0$ for all $s \in \mathcal{N}$ (see (106) and (98)), (ii) arises from (105), and the last equality comes from (103).

Combining the above cases, we complete the proof by

$$\frac{C}{2} \leq C_{\mathrm{r}}^\star = \max_{(h,s,a,P) \in [H] \times \mathcal{S}_{\mathrm{one}} \times \mathcal{A}_{\mathrm{one}} \times \mathcal{U}^\sigma(P^\phi)} \frac{\min\left\{d_h^{\star,P}(s,a), \frac{1}{4SA}\right\}}{d_h^{\mathsf{n},P^{\phi,f}}(s,a)} \leq C.$$

### H.4 Proof of (112)

Recalling (94a) and (96), we first consider a more general form

$$V_h^{\star,\sigma^+,f,\phi}(m_f) - V_h^{\widetilde{\mu},\sigma^+,f,\phi}(m_f)$$

$$= p^{\mathrm{inf}} V_{h+1}^{\star,\sigma^+,f,\phi}(n_f) + (1 - p^{\mathrm{inf}}) V_{h+1}^{\star,\sigma^+,f,\phi}(m_f)$$

$$\quad - \left( m_h^{\widetilde{\mu},f,\phi} V_{h+1}^{\widetilde{\mu},\sigma^+,f,\phi}(n_f) + \left[ 1 - m_h^{\widetilde{\mu},f,\phi} \right] V_{h+1}^{\widetilde{\mu},\sigma^+,f,\phi}(m_f) \right)$$

$$= \left( p^{\mathrm{inf}} - m_h^{\widetilde{\mu},f,\phi} \right) V_{h+1}^{\star,\sigma^+,f,\phi}(n_f) + m_h^{\widetilde{\mu},f,\phi} \left( V_{h+1}^{\star,\sigma^+,f,\phi}(n_f) - V_{h+1}^{\widetilde{\mu},\sigma^+,f,\phi}(n_f) \right)$$

$$\quad + (1 - p^{\mathrm{inf}}) \left( V_{h+1}^{\star,\sigma^+,f,\phi}(m_f) - V_{h+1}^{\widetilde{\mu},\sigma^+,f,\phi}(m_f) \right) - \left( p^{\mathrm{inf}} - m_h^{\widetilde{\mu},f,\phi} \right) V_{h+1}^{\widetilde{\mu},\sigma^+,f,\phi}(m_f)$$

$$= m_h^{\widetilde{\mu},f,\phi} \left( V_{h+1}^{\star,\sigma^+,f,\phi}(n_f) - V_{h+1}^{\widetilde{\mu},\sigma^+,f,\phi}(n_f) \right) + (1 - p^{\mathrm{inf}}) \left( V_{h+1}^{\star,\sigma^+,f,\phi}(m_f) - V_{h+1}^{\widetilde{\mu},\sigma^+,f,\phi}(m_f) \right)$$

$$\quad + \left( p^{\mathrm{inf}} - m_h^{\widetilde{\mu},f,\phi} \right) \left( V_{h+1}^{\star,\sigma^+,f,\phi}(n_f) - V_{h+1}^{\star,\sigma^+,f,\phi}(m_f) \right)$$

$$\geq (1 - p^{\mathrm{inf}}) \left( V_{h+1}^{\star,\sigma^+,f,\phi}(m_f) - V_{h+1}^{\widetilde{\mu},\sigma^+,f,\phi}(m_f) \right)$$

$$\quad + \left( p^{\mathrm{inf}} - m_h^{\widetilde{\mu},f,\phi} \right) \left( V_{h+1}^{\star,\sigma^+,f,\phi}(n_f) - V_{h+1}^{\star,\sigma^+,f,\phi}(m_f) \right)$$

$$\geq (1 - p^{\mathrm{inf}}) \left( V_{h+1}^{\star,\sigma^+,f,\phi}(m_f) - V_{h+1}^{\widetilde{\mu},\sigma^+,f,\phi}(m_f) \right)$$

$$\quad + \frac{1}{2}(p - q) \left\| \widetilde{\mu}_h^{\star,f,\phi}(\cdot \mid m_f) - \widetilde{\mu}_h(\cdot \mid m_f) \right\|_1 \left( V_{h+1}^{\star,\sigma^+,f,\phi}(n_f) - V_{h+1}^{\star,\sigma^+,f,\phi}(m_f) \right), \tag{220}$$

where the last inequality holds since

$$p^{\mathrm{inf}} - m_h^{\widetilde{\mu},f,\phi} = \left( p^{\mathrm{inf}} - q^{\mathrm{inf}} \right) \left( 1 - \widetilde{\mu}_h(\phi_h \mid m_f) \right)$$

$$= (p - q) \left( 1 - \widetilde{\mu}_h(\phi_h \mid m_f) \right)$$

$$= \frac{1}{2}(p - q) \left( 1 - \widetilde{\mu}_h(\phi_h \mid m_f) + \widetilde{\mu}_h(1 - \phi_h \mid m_f) \right)$$

$$= \frac{1}{2}(p - q) \left\| \widetilde{\mu}_h^{\star,f,\phi}(\cdot \mid m_f) - \widetilde{\mu}_h(\cdot \mid m_f) \right\|_1, \tag{221}$$

with the first equality holding by (93) and the second existing by (91).

To further control (220), we have

$$
\begin{aligned}
V_h^{\star,\sigma^+,f,\phi}(n_f) - V_h^{\star,\sigma^+,f,\phi}(m_f) &\overset{(i)}{=} 1 + (1-\sigma^+)V_{h+1}^{\star,\sigma^+,f,\phi}(n_f) + \sigma^+ V_{h+1}^{\star,\sigma^+,f,\phi}(m_f) \\
&\quad - \left( p^{\mathrm{inf}} V_{h+1}^{\star,\sigma^+,f,\phi}(n_f) + (1-p^{\mathrm{inf}})V_{h+1}^{\star,\sigma^+,f,\phi}(m_f) \right) \\
&= 1 + (1 - p^{\mathrm{inf}} - \sigma^+)\left( V_{h+1}^{\star,\sigma^+,f,\phi}(n_f) - V_{h+1}^{\star,\sigma^+,f,\phi}(m_f) \right) \\
&\overset{(ii)}{=} 1 + (1-p)\left( V_{h+1}^{\star,\sigma^+,f,\phi}(n_f) - V_{h+1}^{\star,\sigma^+,f,\phi}(m_f) \right) \\
&= \cdots = \sum_{j=0}^{H-h} (1-p)^j,
\end{aligned}
\tag{222}
$$

where (i) follows from Lemma E.2 and (ii) holds by (91). Then, we consider two cases w.r.t. the uncertainty level $\sigma^+$ to control (222), respectively:

- *When $0 < \sigma^+ \le \frac{c_2}{2H}$:* Recall $p = \frac{c_2}{H}$ if $\sigma^+ \le \frac{c_2}{2H}$. In this case, applying (222), we have

$$
\begin{aligned}
&V_h^{\star,\sigma^+,f,\phi}(n_f) - V_h^{\star,\sigma^+,f,\phi}(m_f) \\
&= \sum_{j=0}^{H-h}(1-p)^j \ge \sum_{j=0}^{H-h}\left(1 - \frac{c_2}{H}\right)^j = \frac{1 - \left(1 - \frac{c_2}{H}\right)^{H-h+1}}{c_2/H} \ge \frac{2c_2(H-h+1)}{3}.
\end{aligned}
\tag{223}
$$

Here, the final inequality holds by observing

$$
\left(1 - \frac{c_2}{H}\right)^{H-h+1} \le \exp\left(-\frac{c_2(H-h+1)}{H}\right) \le 1 - \frac{2c_2(H-h+1)}{3H},
\tag{224}
$$

where the first inequality holds by noticing $c_2 < \frac{1}{2}$ and then $1 - x \le \exp(-x)$, and the last inequality holds by $\exp(-x) \le 1 - \frac{2x}{3}$ for any $0 \le x \le \frac{1}{2}$.

Plugging above fact in (223) back to (220), we arrive at

$$
\begin{aligned}
&V_h^{\star,\sigma^+,f,\phi}(m_f) - V_h^{\widetilde{\mu},\sigma^+,f,\phi}(m_f) \\
&\ge (1-p^{\mathrm{inf}})\left( V_{h+1}^{\star,\sigma^+,f,\phi}(m_f) - V_{h+1}^{\widetilde{\mu},\sigma^+,f,\phi}(m_f) \right) \\
&\quad + \frac{1}{2}(p-q)\left\| \widetilde{\mu}_h^{\star,f,\phi}(\cdot \mid m_f) - \widetilde{\mu}_h(\cdot \mid m_f) \right\|_1 \frac{2c_2(H-h+1)}{3}.
\end{aligned}
\tag{225}
$$

Then, invoking the assumption

$$
\sum_{h=1}^{H}\left\| \widetilde{\mu}_h(\cdot \mid m_f) - \widetilde{\mu}_h^{\star,f,\phi}(\cdot \mid m_f) \right\|_1 \ge \frac{H}{8}
\tag{226}
$$

in (111) and applying (225) recursively for $h = 1, 2, \cdots, H$ yields

$$
\begin{aligned}
&V_1^{\star,\sigma^+,f,\phi}(m_f) - V_1^{\widetilde{\mu},\sigma^+,f,\phi}(m_f) \\
&\ge \frac{c_2}{3}\sum_{h=1}^{H}(1-p^{\mathrm{inf}})^{h-1}(p-q)(H-h+1)\left\| \widetilde{\mu}_h^{\star,f,\phi}(\cdot \mid m_f) - \widetilde{\mu}_h(\cdot \mid m_f) \right\|_1 \\
&\overset{(i)}{\ge} \frac{c_2}{3}\sum_{h=1}^{H}(1-\frac{c_2}{H})^{h-1}(p-q)(H-h+1)\left\| \widetilde{\mu}_h^{\star,f,\phi}(\cdot \mid m_f) - \widetilde{\mu}_h(\cdot \mid m_f) \right\|_1 \\
&\overset{(ii)}{\ge} \frac{c_2}{6}\sum_{h=1}^{H}(p-q)(H-h+1)\left\| \widetilde{\mu}_h^{\star,f,\phi}(\cdot \mid m_f) - \widetilde{\mu}_h(\cdot \mid m_f) \right\|_1 \\
&\overset{(iii)}{=} \frac{c_2\Delta}{6}\sum_{h=1}^{H} h\left\| \widetilde{\mu}_{H-h+1}^{\star,f,\phi}(\cdot \mid m_f) - \widetilde{\mu}_{H-h+1}(\cdot \mid m_f) \right\|_1 \\
&\overset{(iv)}{\ge} \frac{c_2\Delta}{6}\sum_{h=1}^{\lfloor H/16 \rfloor} 2h \ge \frac{c_2\Delta}{6}\lfloor H/16 \rfloor (\lfloor H/16 \rfloor + 1),
\end{aligned}
\tag{227}
$$

where (i) follows from $1 - p^{\mathrm{inf}} \geq 1 - p = 1 - \frac{c_2}{H}$, and (ii) holds by

$$\forall h \in [H]: \quad (1 - \frac{c_2}{H})^{h-1} \geq (1 - \frac{c_2}{H})^H \geq \frac{1}{2}b \tag{228}$$

as long as $c_2 \leq \frac{1}{2}$. Here, (iii) arises from the definition of $p, q$ in (99); (iv) can be verified by the fact that for any series $0 \leq m_1, m_2, \cdots, m_H \leq m_{\max}$ that obeys $\sum_{h=1}^{H} m_h \geq y$, one has

$$\sum_{h=1}^{H} m_h h \geq \sum_{h=1}^{\lfloor m_{\max}/n \rfloor} m_{\max} h, \tag{229}$$

and taking $m_h = \left\| \widetilde{\mu}_{H-h+1}(\cdot \mid m_f) - \widetilde{\mu}_{H-h+1}^{\star, f, \phi}(\cdot \mid m_f) \right\|_1 \leq 2 = m_{\max}$ and $n = \frac{H}{8}$. Consequently, observed from (227), the following inequality holds

$$V_1^{\star, \sigma^+, f, \phi}(m_f) - V_1^{\widetilde{\mu}, \sigma^+, f, \phi}(m_f) \geq \frac{c_2 \Delta}{6} \lfloor H/16 \rfloor \left( \lfloor H/16 \rfloor + 1 \right) \geq c_3 \Delta H^2 > \varepsilon \tag{230}$$

for some small enough constant $c_3$ and letting $\Delta = \frac{\varepsilon}{c_3 H^2}$.

- *When $\frac{c_2}{2H} < \sigma^+ \leq 1 - c_0$:* Similarly, recalling $p = \left(1 + \frac{c_1}{H}\right)\sigma^+$ if $\sigma^+ > \frac{c_2}{2H}$ and invoking (222) gives

$$
\begin{aligned}
V_h^{\star, \sigma^+, f, \phi}(n_f) - V_h^{\star, \sigma^+, f, \phi}(m_f) &= \sum_{j=0}^{H-h} (1 - p)^j = \sum_{j=0}^{H-h} \left(1 - \left(1 + \frac{c_1}{H}\right)\sigma^+\right)^j \\
&\geq \frac{1 - \left(1 - (1 + \frac{c_1}{H})\sigma^+\right)^{H-h+1}}{(1 + \frac{c_1}{H})\sigma^+} \\
&\geq \frac{c_2(H - h + 1)}{3\sigma^+ H},
\end{aligned} \tag{231}
$$

where the final inequality holds by observing

$$
\begin{aligned}
\left(1 - \left(1 + \frac{c_1}{H}\right)\sigma^+\right)^{H-h+1} &\leq \exp\left(-\left(1 + \frac{c_1}{H}\right)\sigma^+(H - h + 1)\right) \\
&\overset{(i)}{\leq} \exp\left(-\frac{c_2}{2H}\left(1 + \frac{c_1}{H}\right)(H - h + 1)\right) \\
&\leq 1 - \left(1 + \frac{c_1}{H}\right)\frac{c_2(H - h + 1)}{3H}. 
\end{aligned} \tag{232}
$$

Here, (i) holds by observing $\frac{c_2}{2H} < \sigma^+$, and the last inequality holds by $\left(1 + \frac{c_1}{H}\right) \leq 2$, $c_2 \leq \frac{1}{2}$, and the fact $\exp(-x) \leq 1 - \frac{2x}{3}$ for any $0 \leq x \leq \frac{1}{2}$.

Plugging (231) into (220) gives

$$
\begin{aligned}
&V_h^{\star, \sigma^+, f, \phi}(m_f) - V_h^{\widetilde{\mu}, \sigma^+, f, \phi}(m_f) \\
&\geq (1 - p^{\mathrm{inf}}) \left(V_{h+1}^{\star, \sigma^+, f, \phi}(m_f) - V_{h+1}^{\widetilde{\mu}, \sigma^+, f, \phi}(m_f)\right) \\
&\quad + \frac{1}{2}(p - q)\left\| \widetilde{\mu}_h^{\star, f, \phi}(\cdot \mid m_f) - \widetilde{\mu}_h(\cdot \mid m_f) \right\|_1 \frac{c_2(H - h + 1)}{3\sigma^+ H}.
\end{aligned} \tag{233}
$$

Following the steps to achieve (227), applying (233) recursively for $h = 1, 2, \cdots, H$ gives

$$
\begin{aligned}
&V_1^{\star, \sigma^+, f, \phi}(m_f) - V_1^{\widetilde{\mu}, \sigma^+, f, \phi}(m_f) \\
&\geq \sum_{h=1}^{H} (1 - p^{\mathrm{inf}})^{h-1}(p - q)\frac{c_2(H - h + 1)}{6\sigma^+ H}\left\| \widetilde{\mu}_h^{\star, f, \phi}(\cdot \mid m_f) - \widetilde{\mu}_h(\cdot \mid m_f) \right\|_1 \\
&\overset{(i)}{=} \frac{c_2(p - q)}{6\sigma^+ H} \sum_{h=1}^{H} (1 - \frac{c_1}{H})^{h-1}(H - h + 1)\left\| \widetilde{\mu}_h^{\star, f, \phi}(\cdot \mid m_f) - \widetilde{\mu}_h(\cdot \mid m_f) \right\|_1 \\
&\overset{(ii)}{\geq} \frac{c_2 \Delta}{12\sigma^+ H} \lfloor H/16 \rfloor \left( \lfloor H/16 \rfloor + 1 \right),
\end{aligned} \tag{234}
$$

where (i) follows from $1 - p^{\text{inf}} = 1 - (p - \sigma^+) = 1 - \frac{c_1}{H}\sigma^+$, and (ii) holds by letting $c_1 \le \frac{1}{2}$ and following the same routine of (227). Consequently, (234) yields

$$V_1^{\star,\sigma^+,f,\phi}(m_f) - V_1^{\widetilde{\mu},\sigma^+,f,\phi}(m_f) \ge \frac{c_2\Delta}{12\sigma^+ H}\lfloor H/16\rfloor\left(\lfloor H/16\rfloor + 1\right) \ge \frac{c_4\Delta H}{\sigma^+} > \varepsilon, \tag{235}$$

which holds for some small enough constant $c_4$ and letting $\Delta = \frac{\sigma^+ \varepsilon}{c_4 H}$.

