# OpenReview forum: "Sample-Efficient Tabular Self-Play for Offline Robust Reinforcement Learning"
_NeurIPS.cc/2025/Conference — NeurIPS 2025 poster_

### Official Review · Reviewer_A8VL · 2025-06-23

**Clarity:** 3
**Significance:** 3
**Originality:** 4
**Rating:** 5
**Confidence:** 2

**Summary:**

This paper tackles the challenging problem of offline robust reinforcement learning in two-player zero-sum Markov games (RTZMGs), with a focus on tabular environments under partial coverage and environmental uncertainty. The authors introduce RTZ-VI-LCB, a novel model-based algorithm combining robust value iteration with lower confidence bounds. It incorporates a Bernstein-style penalty and a two-stage subsampling procedure to handle statistical dependencies in offline datasets. The paper provides theoretical guarantees by deriving both upper and lower bounds on sample complexity, showing that RTZ-VI-LCB achieves optimal sample efficiency with respect to state and action sizes. The method is further generalized to the multi-agent setting.

**Questions:**

1. The concentrability coefficient C_{r}^* plays a central role in your bounds. Is there a principled way to estimate or bound C_{r}^* from offline data in practice, especially when the robust optimal policies are unknown?

2. Solving robust zero-sum matrix games is generally PPAD-hard. In your implementation of RTZ-VI-LCB, how do you compute the robust NE at each state and timestep efficiently? Are there computational relaxations or approximations used?

3. Your theoretical contribution is strong, but the empirical validation is limited to synthetic tabular environments. Do you envision obstacles in extending RTZ-VI-LCB to high-dimensional? What are the main challenges in applying your framework to more realistic offline multi-agent domains?

4. The proposed Bernstein-style penalty and two-stage subsampling are key components to deal with partial coverage and statistical dependence. Could you provide insights into how much each of these components contributes to the algorithm's performance and stability?

**Ethical Concerns:**

["NO or VERY MINOR ethics concerns only"]

**Final Justification:**

The authors have provided clear answers to my questions, addressing both theoretical aspects, namely, the infeasibility of estimating  $C_{\mathrm{r}}^\star$​ from offline data, and implementation details such as robust NE computation and the roles of the Bernstein penalty and subsampling. Based on these clarifications, I maintain my initial score of 5 (Accept).

**Limitations:**

Yes.

**Paper Formatting Concerns:**

Nothing to report.

**Quality:**

4

**Strengths And Weaknesses:**

Strengths
- The paper presents a significant theoretical contribution by establishing optimal sample complexity bounds for robust Nash equilibrium computation in offline RTZMGs. The use of robust unilateral clipped concentrability (Definition 4.1) is novel and tighter than previous assumptions (e.g., in P2M2PO).
- The method is carefully constructed with a dual formulation for value computation, a tailored penalty function, and proofs of optimality across uncertainty regimes.
- First optimal bound in robust MARL under offline constraints

Weaknesses
- While the paper is positioned as theoretical, it includes a small experimental section that aims to support the theoretical results. However, the experiments are limited to synthetic, low-dimensional settings with no real-world applicability or comparison to existing baselines like P2M2PO. As such, the empirical section offers limited additional insight and does not strongly validate the practical robustness of the method.
- The paper could benefit from a short discussion on how to move toward larger and practical domains.

---

> ### Author Rebuttal · Authors · 2025-07-31
>
> **Answer to W1**
>
> We thank the reviewer for this valuable feedback. To strengthen the empirical validation, we have added a new comparison against $\mathrm{P}^2\mathrm{M}^2\mathrm{PO}$, a recent baseline designed for RTZMGs. Specifically, we report the performance gap $\mathrm{Gap}({\mu}, {\nu})$ with respect to dataset size across both methods. To demonstrate our optimal sample complexity scaling in the action space, we use a setting with $S = 50$, $A = B = 3$, and $H = 100$, which differs from the configuration in Appendix A (where $AB = A + B$). The new results are presented in the following table, which will be included in the revised manuscript:
>
> | $\log(KH)$            | $6$      | $7$      | $8$      | $9$      |
> |-----------------------|----------|----------|----------|----------|
> | **RTZ-VI-LCB**        | **3.1699** | **2.1498** | **0.2324** | **0.0819** |
> | $\mathrm{P}^2\mathrm{M}^2\mathrm{PO}$ | 3.5970   | 2.2503   | 0.2404   | 0.0890   |
>
> These results highlight the superiority of our method across varying dataset sizes and validate the theoretical claims under controlled experimental conditions. While we focus on tabular settings in this work, our algorithm can serve as a theoretical foundation for scalable extensions based on linear or kernel-based function approximation.
>
> **Answer to W2**
>
> Thank you for this helpful suggestion. Our current study focuses on tabular settings, which provide a clean foundation for precise theoretical analysis and algorithmic development. To scale to larger and more practical domains, straightforward approaches are function approximation methods, particularly linear or kernel-based methods, as explored in prior works, e.g., Refs. [1,2]. These approaches enable generalization over large or continuous state-action spaces while preserving a degree of theoretical tractability. We will include a brief discussion of this extension path in Section 5 of the revised version, stating that: "*While RTZ-VI-LCB is developed in the tabular setting, it offers a solid theoretical foundation for robust offline multi-agent learning. Generalizing its core principles, such as data-driven penalization, to high-dimensional settings via function approximation (e.g., linear or kernel-based methods [1,2]) represents a promising direction toward practical deployment.*"
>
> **Answer to Q1**
>
> Thank you for this insightful question. In general, estimating $C_{\mathrm{r}}^\star$ from purely offline data is information-theoretically impossible. Consider, for example, the degenerate robust two-player Markov game with $\sigma^+=\sigma^- = 0$ and $B=1$, which reduces to the standard single-agent RL setting.  For an MDP with horizon $H=2$, whose first step has a singleton state space $\mathcal{S}_ 1=\\{0\\}$ and action space $\mathcal{A}_ 1=\\{0,1\\}$, and whose second step has state space $\mathcal{S}_ 2=\\{0,1\\}$ and singleton action space $\mathcal{A}_ 2=\\{0\\}$, [24] has shown it is impossible to determine the sample size required or provide a valid certificate. Therefore, estimating $C_ {\mathrm{r}}^\star$ is also generally infeasible in RTZMGs.
>
> Fortunately, our algorithm does not rely on prior knowledge of $C_{\mathrm{r}}^\star$. It can be executed with any offline dataset and will succeed once the underlying task becomes statistically feasible. Although we cannot determine in advance whether a given dataset is sufficient, RTZ-VI-LCB remains practically useful by avoiding reliance on unverifiable assumptions.
>
> We will incorporate this discussion near Definition 4.1 in the revised manuscript, adding:
> "*Estimating the concentrability coefficient $C_{\mathrm{r}}^\star$ from offline data is generally information-theoretically impossible, as demonstrated even in the single-agent case by the example construction in Section 3.4 of [24]. Fortunately, this does not affect the execution of our algorithm.*"
>
> **Answer to Q2**
>
> Thank you for this thoughtful question. In this work, we do not explicitly address the computational complexity of solving the robust Nash equilibrium (NE). That is, our algorithm assumes access to arbitrary robust NE solver and focuses on the sample complexity of the overall procedure. In addition, our framework is compatible with any computational relaxations or approximation methods for solving RTZMGs, such as Monte-Carlo sampling (Ref. [3]).  In our experiments, we use the Python package *nashpy* for NE computation.
>
> We will add the following remark to the manuscript to clarify this implementation detail: *"In our experiments, the robust NE at each state and timestep is computed using standard NE solvers, i.e., the Python package nashpy. Our algorithm is compatible with any exact or approximate NE solver, including computational relaxations or sampling-based methods. If an approximate solver is used, the corresponding approximation error will be introduced into our theoretical analysis as an additional term."*
>
> **Answer to Q3**
>
> Thank you for highlighting this important point. Our current work focuses on multi-agent tabular settings (see Theorem 4.4 and Appendix F) to facilitate precise theoretical analysis. To extend RTZ-VI-LCB to high-dimensional cases, a natural direction is to incorporate function approximation, particularly using linear function classes or kernel-based representations. Key challenges in this setting include ensuring concentration and coverage conditions in high-dimensional spaces. As responded to Weakness 2, we will include a discussion of this extension path in Section 5 of the revised version.
>
> **Answer to Q4**
>
> Thank you for the insightful question.
> The Bernstein-style penalty acts as an adaptive, variance-aware term. In particular, it significantly improves the algorithm’s performance when the actual variance is much smaller than the trivial bound of $H^2$. By leveraging a tighter, data-dependent penalty, the algorithm can achieve improved performance.
>
> In contrast, the two-stage subsampling is primarily introduced for theoretical analysis, specifically to handle within-episode temporal dependence. In practice, we have observed that this subsampling step is not strictly required for the empirical performance. However, how to rigorously remove this step while maintaining theoretical guarantees remains an open problem for future research.
>
> We will clarify these insights in Section 1.1 and discuss the practical and theoretical roles of each component in more detail.
>
> **Reference**
>
> [1] Nika, Andi, et al. "Corruption-robust offline two-player zero-sum Markov games." International Conference on Artificial Intelligence and Statistics. PMLR, 2024.
>
> [2] Lim, Shiau Hong, and Arnaud Autef. "Kernel-based reinforcement learning in robust Markov decision processes." International conference on machine learning. PMLR, 2019.
>
> [3] Ponsen, Marc, Steven De Jong, and Marc Lanctot. "Computing approximate Nash equilibria and robust best-responses using sampling." Journal of Artificial Intelligence Research 42, 575-605, 2011.

---

> > ### Comment · Reviewer_A8VL · 2025-08-04
> >
> > Thank you to the authors for the responses, which have clarified my concerns. I have no further questions.

---

### Official Review · Reviewer_V9XV · 2025-07-01

**Clarity:** 3
**Significance:** 3
**Originality:** 2
**Rating:** 4
**Confidence:** 4

**Summary:**

This paper addresses robust two-player zero-sum Markov games (TZMGs) by introducing the RTZ-VI-LCB algorithm, under the offline setting.

**Questions:**

(1). Can you explain the major difference of your method compared to [a]+offline LCB approach for robust RL?
(2). Can you connect your result together with the generative model setting? Namely, if the dataset is generated based on a uniform distribution, what will the $C^*_r$ term become, and what is the resulting result? And how is that result compared to [34]?

**Ethical Concerns:**

["NO or VERY MINOR ethics concerns only"]

**Final Justification:**

Authors have addressed most of my concerns.

**Limitations:**

Yes.

**Quality:**

3

**Strengths And Weaknesses:**

S: (1). Robust Markov game is a very interesting topic with growing research interests.
(2). The paper provides a rigorous theoretical framework, including upper and lower sample complexity bounds. The upper bound is near-optimal.

W: (1). The major weakness is on the limited technical novelty of the work. The algorithm is an direct extension of [a], together with standard LCB technique for offline RL.
(2). Some references on offline single-agent robust RL are missing, e.g., [b-e]. Also a work [f] on robust games:

[a] Model-Based Reinforcement Learning for Offline Zero-Sum Markov Games
Y Yan, G Li, Y Chen, J Fan - arXiv preprint arXiv:2206.04044, 2022

[b] Distributionally robust off-dynamics reinforcement learning: Provable efficiency with linear function approximation
Z Liu, P Xu
International Conference on Artificial Intelligence and Statistics, 2719-2727

[c] Minimax optimal and computationally efficient algorithms for distributionally robust offline reinforcement learning
Z Liu, P Xu
arXiv preprint arXiv:2403.09621

[d] A unified principle of pessimism for offline reinforcement learning under model mismatch
Y Wang, Z Sun, S Zou
Advances in Neural Information Processing Systems 37, 9281-9328

[e]Distributionally robust model-based offline reinforcement learning with near-optimal sample complexity
L Shi, Y Chi - Journal of Machine Learning Research, 2024

[f]Breaking the Curse of Multiagency in Robust Multi-Agent Reinforcement Learning
L Shi, J Gai, E Mazumdar, Y Chi, A Wierman - arXiv preprint arXiv:2409.20067, 2024

---

> ### Author Rebuttal · Authors · 2025-07-31
>
> **Answer to W1 and Q1**
>
> We sincerely appreciate the reviewer’s insightful comment regarding the novelty of our work. The study by Yan et al. [a] investigates offline Markov games (MGs) using the LCB approach. Our work builds upon this line by explicitly incorporating environmental uncertainty, resulting in a substantially stronger and more challenging formulation. In the presence of uncertainty, we can no longer solve the value function in its primal form and need to operate instead in the dual space. To address this, we introduce a novel Bernstein-style penalty that ensures the empirical variance estimate (i.e., the plug-in estimate) closely matches the true variance. Through these innovations, we establish a minimax optimal sample complexity that significantly improves over prior results in its dependence on the state and action spaces under uncertainty. While the best existing bounds scale as
> $
> O\left( \frac{C_ \mathrm{r} H^5 S^2 A B}{\varepsilon^2} \right),
> $
> our result achieves
> $
> O\left( \frac{C^\star_\mathrm{r} H^4 S(A+B)}{\varepsilon^2} \cdot f(\sigma^+, \sigma^{-}, H) \right)
> $ with $C^\star_\mathrm{r}\le C_\mathrm{r}$ and $f(\sigma^+, \sigma^{-}, H)\le H$,
> which notably reduces the dependence on both the state and action dimensions.
> We will revise the manuscript to include a discussion clarifying the novelty, stating: "*RTZ-VI-LCB extends the offline MGs algorithm proposed in [a] by explicitly incorporating environmental uncertainty and solving the dual problem.*"
>
> **Answer to W2**
>
> Thank you for pointing this out. We apologize for the omission and will revise Section 1.2 (Related Work) to incorporate the missing references more comprehensively. Specifically, in the paragraph on Single-agent robust offline RL, we will include the following sentence as "*Several works [b, c] study offline robust RL with linear function approximation, while others consider various uncertainty metrics such as total variation, $\chi^2$-divergence, and KL-divergence [d], or operate under more general settings without requiring strong coverage assumptions [e].*" In the paragraph on Robustness in MARL, we will add "*Optimal dependence on the action space size $\{A, B\}$ has been achieved in recent work from an online learning perspective [f], but the offline robust formulation in multi-agent settings remains largely underexplored.*"
>
> **Answer to Q2**
>
> Thank you for this insightful and inspirational question. Under a generative model with uniform sampling, the visitation distribution becomes explicit; that is, $d_h^{\mathsf{n}, P^0}(s, a, b) = \frac{1}{SAB}$. In this case, it is sufficient to choose $C_{\mathrm{r}}^\star = \min\\{A, B\\}$, which can be readily verified to satisfy Eq. (20). This choice leads to the sample complexity
> $$
> \widetilde{\mathcal{O}}\left( \frac{H^{4} SAB}{\varepsilon^{2}} \cdot f(\sigma^+, \sigma^{-}, H) \right),
> $$
> which matches the bound established in [34] and confirms the efficiency of our approach in the generative model setting.
> We will add a corresponding remark after Theorem 4.2 to clarify this point.

---

> > ### Comment · Reviewer_V9XV · 2025-08-01
> >
> > I appreciate the authors’ detailed response.
> >
> > (1) I am still not fully convinced by the theoretical justification. When the authors write:
> >         ``While the best existing bounds scale as $O\left( \frac{C_\mathrm{r} H^5 S^2 A B}{\varepsilon^2} \right)$'',
> >     is this based on a specific prior work, or is it derived directly by extending single-agent results? Clarifying this would help contextualize the improvement.
> >
> > (2) Also, regarding the statement: ``we can no longer solve the value function in its primal form and need to operate instead in the dual space. To address this, we introduce a novel Bernstein-style penalty that ensures the empirical variance estimate (i.e., the plug-in estimate) closely matches the true variance'',
> >     this seems to follow a standard approach in robust RL, where most works already operate in the dual space and use similar techniques for variance control. Could the authors clarify what is novel in their formulation?
> >
> > (3) The reduction from $AB$ to $A + B$ in the complexity is interesting. However, I would appreciate a more precise explanation of the key driver behind this improvement. Many prior works in robust multi-agent learning still suffer from the curse of joint action spaces. What structural property or assumption enables this result in your case?
> >
> > (4) A related concern: why does the multi-agent setting appear to achieve better complexity than the single-agent one (if we treat the joint policy as a single agent over action space $AB$)? Is this improvement due to the nature of equilibrium computation (e.g., computing a NE rather than an optimal policy), or does it reflect a deeper structural advantage?
> >
> > (5) I am slightly confused by the uniform-data case. If the dataset is uniformly collected, does the sample complexity again depend on $AB$, rather than $A + B$? If so, does this suggest that the improvement is primarily due to the problem setting (i.e., the form of the offline dataset or associated concentrability terms), rather than a fundamentally new algorithmic insight?

---

> > > ### Author Response · Authors · 2025-08-02
> > >
> > > Thanks for your time your reply your efforts for reviewing this paper.
> > >
> > > **Answer to Q1**
> > >
> > > We apologize for the missing reference. The bound
> > > $O\left( \frac{C_\mathrm{r} H^5 S^2 A B}{\varepsilon^2} \right)$
> > > was achieved by the prior work P$^2$M$^2$PO [5].
> > >
> > > **Answer to Q2**
> > >
> > > Indeed, our algorithm is based on existing ideas like the Bernstein-type penalty and dual formulation. But how to apply these techniques to RTZMGs and achieve optimal complexity is still underdeveloped. For example, to address the temporal correlation arising from offline data, we make use of a two-stage subsampling scheme, which is the first one applied in the robust setting. Furthermore, we incorporate a Bernstein-style penalty, which is sensitive to the actual variance and is often much smaller than the worst-case bound of $H^2$. Unlike prior works that rely on generic plug-in estimates, our penalty incorporates game-specific structure and improved sample efficiency. Besides, our analysis achieves a sample complexity reduction from $AB$ to $A+B$; see our "Answer to Q3" for further details.
> > >
> > > Thus, our algorithm achieves the optimal sample complexity in terms of $(S, A, B)$ and the best on $H$ for offline RTZMGs. Specifically, the state-of-the-art P$^2$M$^2$PO [5] incurs a complexity of $O\left( \frac{C_\mathrm{r} H^5 S^2 A B}{\varepsilon^2} \right)$, while our method improves this to $O\left( \frac{C_\mathrm{r}^\star H^4 S (A + B)}{\varepsilon^2} \right)$, with $C^\star_\mathrm{r} \le C_\mathrm{r}$ and $f(\sigma^+, \sigma^{-}, H)\le H$.
> > > We will add this discussion to our revised manuscript.
> > >
> > > **Answer to Q3**
> > >
> > > To establish the sample complexity upper bound that scales with $A+B$ rather than $AB$, our proof introduces three key analytical components. First, we employ a leave-one-out analysis to effectively decouple complex statistical dependencies across iterations, which, to the best of our knowledge, is applied for the first time in the context of RTZMGs. Second, we develop a self-bounding argument, where a target quantity is upper bounded by a contraction of itself plus a set of well-controlled error terms, enabling a sharp bound on the duality gap. Finally, we introduce an auxiliary policy $\mu^{\mathsf{d}}(s)$ (see Eq. (51), Appendix D.2). This auxiliary policy allows us to effectively decouple the two agents' actions. As a result, the key error term can be bounded via Eq. (61) as
> > > $$2\sum_{i=h}^H \left\langle d_i^{\sf{p}, \mu^{\sf{d}},\nu^\star}, \beta_i^{\mu^{\sf{d}},\nu^\star} \right\rangle,$$
> > > which scales linearly with the number of individual actions $A + B$ rather than $AB$. Together, these techniques yield a tight and optimal sample complexity bound on $\\{S, A, B\\}$ for offline RTZMGs, improving upon the state-of-the-art [5]. We will add a corresponding remark after Theorem 4.2 to clarify this point.
> > >
> > > **Answer to Q4**
> > >
> > > We clarify that the multi-agent robust setting does not achieve better sample complexity than the single-agent case. The single-agent case can attain a rate of $O\left(\frac{C^\star H^5S}{P_{\min}^\star\sigma^2\varepsilon^2}\right)$ [1]. On the contrary, the multi-agent robust setting is fundamentally more challenging due to the nature of equilibrium computation under adversarial uncertainty.
> > >
> > > In RTZMGs, one must compute a robust Nash equilibrium over the product space, which involves simultaneous optimization of both agents' policies against the worst-case dynamics. In contrast, the single-agent robust setting involves optimizing a policy over a single action space, without needing to account for an adaptive opponent. This typically results in a lower sample complexity. Moreover, the multi-agent robust setting requires stronger assumptions on the state-action coverage to be tractable.
> > >
> > > Thus, the multi-agent robust setting is generally more complex than its single-agent counterpart in terms of both computation and sample efficiency.
> > >
> > > [1] Distributionally robust model-based offline reinforcement learning with near-optimal sample complexity
> > >
> > > **Answer to Q5**
> > >
> > > We would like to clarify that the claimed improvement in our paper is **not in comparison with uniform setting**. Specifically, our contribution lies in improving the sample complexity of the **offline case** over prior works under the **same problem setting**. In addition, the uniform sampling is not the focus of this paper, although our result can be extended to this case. The $AB$ dependency under the uniform setting arises naturally from its sampling mechanism, as we take the same number of samples for all action pairs simultaneously. We will add the following discussion to make this point more clear: "*Our contribution lies in improving the sample complexity of the offline case over prior works. Although other sampling regimes are not the focus of this paper, it can be checked easily that applying our result directly to uniform setting can achieve the same bound ${O}\left( \frac{H^{4} SAB}{\varepsilon^{2}} \cdot f(\sigma^+, \sigma^{-}, H) \right)$.*"

---

> ### Comment · Reviewer_V9XV · 2025-08-02
>
> I appreciate authors' detailed and quick response. I have no more questions and have raised my score.

---

> > ### Author Response · Authors · 2025-08-03
> >
> > We sincerely thank the reviewer for their thoughtful comments and for reconsidering the evaluation of our work. We appreciate the constructive feedback which helped us improve the paper.

---

### Official Review · Reviewer_S3RM · 2025-07-03

**Clarity:** 3
**Significance:** 4
**Originality:** 3
**Rating:** 5
**Confidence:** 3

**Summary:**

The authors of this work develop a new lower bound on the sample complexity of robust MARL. They also present an offline algorithm nearly whose sample complexity is near optimal based on the lower bound.

**Questions:**

It is known that computing robust policies and even approximating them is NP-hard in many regimes. What is the time complexity of your algorithm and do the authors foresee a potential tradeoff between time complexity and sample complexity for robust settings? Also, do the authors believe their results could potentially extend to uncertainty sets beyond (s,a,b) rectangularity? In particular, other works have been able to handle more complex uncertainty sets through a relationship with regularized Markov games.

**Ethical Concerns:**

["NO or VERY MINOR ethics concerns only"]

**Final Justification:**

Overall, I feel very highly of the paper. The robustness literature for standard RL has been very rich, but the literature for MARL has yet to catch up. Much of the robust MARL literature focuses on the planning setting, with only a few results on the sample complexity. This work nicely fills this gap by not only giving improved results, but also giving optimal results. The lower-bound provided not only elevates their proposed algorithm, but also gives new insights on what is possible in the robust MARL regime, which is a significant theoretical discovery. The main downsides of this work are the focus on (s,a,b)-rectangularity, use of NE concept, and high time complexity, but I would argue these plague many of these game-theoretic approaches, so making progress on the orthogonal issue of sample complexity is welcomed and moves the field nicely forward until the planning issue is resolved.

**Limitations:**

Yes

**Quality:**

3

**Strengths And Weaknesses:**

The brand new lower bound is very significant to the robust MARL literature, and the near-optimal attaining algorithm presents new ideas that push the field forward. The primary downside to this model is the adherence to the (s,a,b)-rectangular model of uncertainty, and the potential computational difficulties that typically arise in such works. Despite these downsides, the results presented are still significant.

---

> ### Author Rebuttal · Authors · 2025-07-31
>
> **Answer to Question about time complexity and the tradeoff between time complexity and sample complexity**
>
> We thank the reviewer for this important question. As the reviewer mentioned, computing Nash Equilibria (NE) for two-player general-sum games is PPAD-hard. Since our algorithm solves an NE at each horizon and state, its overall time complexity is also PPAD-hard.
>
> Regarding the tradeoff, there does not exist a direct tradeoff between time complexity and sample complexity, as the main computational bottleneck is NE computation. A potential aspect of tradeoff could lie in the number of NE computations required by the algorithm. In this regard, the number of NE computations in our algorithm is $HS$ (with $H$ as the horizon and $S$ as the number of states), which is optimal. Therefore, our work primarily focuses on improving sample efficiency, as the time complexity per NE computation is dictated by the underlying hardness of NE-solving itself.
>
> **Answer to uncertainty sets beyond $(s,a,b)$-rectangularity**
>
> We thank the reviewer for this valuable suggestion. Extending our framework to uncertainty sets beyond $(s,a,b)$-rectangularity is indeed an important direction. In particular, KL-divergence-based uncertainty sets offer a principled connection between hard robustness and soft entropy regularization. To extend our algorithm to handle KL divergence, two key aspects must be addressed.
>
> Algorithmically, RTZ-VI-LCB can be generalized by replacing its robust Bellman operator with an entropy-regularized (soft) Bellman operator and solving for a \emph{regularized} Nash equilibrium. Additionally, the data-driven penalty term would need to be modified. Instead of relying on the empirical variance $\mathsf{Var}_ {\widehat{P}^0_ {h,s,a,b}}(\widehat{V})$, the penalty should incorporate the minimal nonzero transition probability $
> \hat{P}_ {\min, h}(s, a, b) = \min_ {s'} \\{ \hat{P}_ h^0(s' \mid s, a, b) : \hat{P}_ h^0(s' \mid s, a, b) > 0 \\}$.
>
> As for theoretical analysis, this generalization would require a different complexity characterization. In particular, the sample complexity would depend on the smallest positive transition probability $P_ {\min}^\star = \min_ {h, s, a, b, s'} \\{ P_ h^0(s' \mid s, \mu_h^\star(s), \nu_h^\star(s)) > 0 \\}$ under the optimal robust policy, as well as a burn-in cost condition on the smallest positive state transition probability $P_ {\min}^{\sf{n}} = \min_ {h, s, a, b, s'} \\{ P_ h^0(s' \mid s, a, b) > 0 : d_h^{\mathsf{n}, P^0}(s, a, b) > 0 \\}$ of the behavior policy.
>
> We will include this discussion in Section 5 to clarify how our algorithm can be extended to more general uncertainty structures, including KL-divergence-based sets.

---

> > ### Comment · Reviewer_S3RM · 2025-08-04
> > **Rebuttal Response**
> >
> > I appreciate the authors' detailed response. Since the NE computation forces your algorithmic approach to be PPAD-hard, I have a few follow-up questions. First, do you believe the PPAD-hardness significantly limits the applicability of your approach, as even if the sample complexity is low, actually performing the whole process could take significant time (or perhaps all known algorithms suffer from this problem, but yours is still faster)? Moreover, have the authors thought of using approximation algorithms for the NE computation, or even to relax the goal to CCE (even robust versions are poly-time solvable) computation, to ensure the whole process is efficient?

---

> ### Author Response · Authors · 2025-08-05
>
> We would like to thank you for the time and effort you dedicated to the review.
>
> **Answer to Q1**
>
> Thank you for this intriguing question.
> In our setting, the number of NE computations required under the robust NE policy is $HS$. All algorithms in the same setting need to calculate at least $HS$ number of NE computations since there are $HS$ different states.
> Hence, as the reviewer pointed out, our approach achieves optimality in both sample complexity and computational complexity to find an $\varepsilon$-optimal robust NE policy.
>
> **Answer to Q2**
>
> We thank the reviewer for this valuable suggestion. In this work, we do not explicitly address the computational complexity of solving the robust NE. In our implementation, we employ the Python package *nashpy* for NE computation.
>
> Generally, our framework is compatible with a wide range of computational relaxations and approximation schemes. For instance, NE computation can be approximated using Monte Carlo sampling–based methods (Ref. [1]).
> In addition, another way to address such a computation issue is to consider CE (Ref. [2]) or CCE (Ref. [3]) instead of NE, both of which admit polynomial-time solutions even in the robust setting. In this case, our algorithm can be directly adapted by substituting $\mathsf{ComputNash}(\cdot)$ in Algorithm 2 with the corresponding CE or CCE computation.
>
> We will incorporate a discussion on NE approximations and relaxations into the revised manuscript to clarify the computational aspects.
>
> **Reference**
>
> [1] Ponsen, Marc, Steven De Jong, and Marc Lanctot. "Computing approximate Nash equilibria and robust best-responses using sampling." Journal of Artificial Intelligence Research, 42, 575-605, 2011.
>
> [2] Chen, Ziyi, Shaocong Ma, and Yi Zhou. "Finding correlated equilibrium of constrained markov game: A primal-dual approach." Advances in Neural Information Processing Systems, 35, 25560-25572, 2022.
>
> [3] Chen, Zixiang, Dongruo Zhou, and Quanquan Gu. "Almost optimal algorithms for two-player zero-sum linear mixture markov games." International Conference on Algorithmic Learning Theory, PMLR, 2022.

---

> > ### Comment · Reviewer_S3RM · 2025-08-09
> > **Response Acknowledgement**
> >
> > Thanks! I have no more questions and have edited my final justification appropriately.

---

### Decision · Program_Chairs · 2025-09-17

**Decision:**

Accept (poster)

**Comment:**

This paper gives a near-optimal sample-complexity algorithm for offline robust two-player zero-sum Markov games, along with a matching lower bound. The theory is clean and addresses an important gap in robust MARL, where offline data and uncertainty interact in subtle ways. The presentation is clear, the assumptions are stated upfront, and the proof techniques look reusable by others in the area.